# Bregman Meets Lévy: Stochastic Mirror Descent with Heavy-Tailed Noise in Continuous and Discrete time

Pierre-Louis Cauvin [1]    Panayotis Mertikopoulos [1]

## Abstract

We study the robustness of stochastic mirror descent (SMD) under heavy-tailed noise, focusing on whether the method retains its convergence guarantees when run with infinite-variance stochastic gradient input. To address this question in a principled manner, we begin by introducing a continuous-time model of SMD as a stochastic differential equation (SDE) driven by a centered Lévy noise process with finite $p$-th order moments, $1 < p \leq 2$. This scheme—which we call the *Lévy mirror flow* (LMF)—arises naturally as the scaling limit of SMD in the presence of heavy-tailed noise. In particular, when $p < 2$—the *heavy noise* regime—the trajectories of LMF generically exhibit jump discontinuities of arbitrary magnitude which, if frequent enough, lead to infinite variance. Nonetheless, despite this highly singular behavior, we show that LMF attains $\varepsilon$-optimality within $\mathcal{O}(\varepsilon^{-p/(p-1)})$ time in the convex case, and within $\tilde{\mathcal{O}}(\varepsilon^{-1/(p-1)})$ time for (relatively) strongly convex objectives. These guarantees provide a transparent characterization of the impact of frequent long jumps on the convergence of the process, and percolate to a series of matching discrete-time guarantees for several variants of SMD under heavy-tailed noise.

## 1 Introduction

**Background and motivation.** Stochastic mirror descent (SMD) and its variants comprise one of the most widely studied classes of first-order methods in stochastic optimization. Originally conceived by Nemirovski [76] and Nemirovski & Yudin [77] as a general scheme for solving constrained convex minimization problems, SMD has since grown into a mature algorithmic template with applications in min-max problems and variational inequalities [78, 80, 81], online optimization and bandits [1, 7, 22, 53], games and equilibrium problems [24, 42, 74], sampling [26, 37], optimal transport [88, 118], and many other fields that are central to machine learning.

At a high level, mirror descent methods proceed by show replacing Euclidean projections with a non-Euclidean "prox-type" step—or, dually to this, by aggregating gradient steps in an unconstrained, dual space, and performing a "mirror step" at each iteration to generate a new query point and obtain a new gradient sample. This mechanism links the geometry of the underlying problem with the prox-step defining the method and which, if chosen appropriately, can lead to superior, almost-dimension-free convergence guarantees.

A major challenge in this setting occurs when the gradient samples that are fed into the algorithm have infinite variance, thus jeopardizing the method's convergence—which usually presumes tame, light-tailed noise. This is a particularly pernicious issue as, even in 1-dimensional quadratic problems, running "vanilla" stochastic gradient descent (SGD) with heavy-tailed noise could lead to divergence in cases where the method is otherwise convergent [32, 39, 117].

Due to the massive—and massively complex—structure of deep neural nets, this "heavy noise" regime is ubiquitous in contemporary machine learning models and architectures, cf. [35, 36, 60, 101, 109, 117] and references therein. Especially in the context of deep learning, the heavy-tailed behavior of gradient noise was first highlighted by Şimşekli et al. [103], who argued empirically that gradient minibatches in deep nets follow a heavy-tailed $\alpha$-stable noise distribution, even in relatively small datasets—such as MNIST and CIFAR10. Even though these findings were later challenged by Xie et al. [112], subsequent work—both empirical and theoretical—has reaffirmed the heavy-tailed profile of the noise in a wide range of training regimes, from adaptive, momentum-based methods, to recurrent networks [117], deep convolutional architectures [10, 120], large language models [2], and reinforcement learning [33]. Several theoretical works further attribute these phenomena to multiplicative noise effects leading to the convergence

---

[1]Univ. Grenoble Alpes, CNRS, Inria, Grenoble INP, LIG, 38000 Grenoble, France. Correspondence to: Pierre-Louis Cauvin <pierre-louis.cauvin@univ-grenoble-alpes.fr>.

*Proceedings of the 43rd International Conference on Machine Learning*, Seoul, South Korea. PMLR 306, 2026. Copyright 2026 by the author(s).

of the iterates toward a heavy-tailed stationary distribution [29, 35, 36, 87].

**Our contributions in the context of related work.** Our aim in this paper is precisely this: *to examine the robustness of stochastic mirror descent methods under heavy noise.*

Building on a long-standing trend in optimization, we begin with a continuous-time viewpoint and we introduce the *Lévy mirror flow* (LMF), a stochastic differential equation (SDE) formulation of SMD driven by a centered Lévy noise process. In contrast to existing SDE models of mirror descent driven by *Brownian* noise [26, 37, 50, 56, 69–72, 91, 118], the introduction of Lévy noise changes the landscape dramatically. If the SDE is driven by an Itô, Brownian-like martingale, the solutions of the SDE are inherently "diffusive": (*i*) they exhibit tame, light-tailed increments; (*ii*) their accumulation leads to a finite-variance process; and (*iii*) the resulting sample paths are continuous (a.s.). By contrast, under Lévy noise, LMF trajectories are "purely discontinuous": (*i*) their increments are no longer Gaussian but heavy–tailed; (*ii*) they may—*and do!*—have infinite variance; and (*iii*) their sample paths may—*and do!*—exhibit jump discontinuities of arbitrary magnitude.

This leads to a highly challenging optimization landscape, especially because Itô's formula—the workhorse of stochastic calculus—cannot be applied in this context. A key technical contribution of our paper—which, to the best of our knowledge, is new in the stochastic analysis literature—is a *weak Itô formula* for convex functions that are not $C^2$ (so the standard Itô formula does not apply).

Thanks to this new technical tool, we analyze several variants of the method, with both constant and variable learning rate, and we establish the following series of results:

1. In convex problems, time-averaged LMF orbits attain an $\varepsilon$-approximate solution within $\tilde{\mathcal{O}}(\varepsilon^{-p/(p-1)})$ time.

2. In strongly convex problems, LMF orbits converge at a geometric rate to an "uncertainty ball" whose radius $\delta_\eta$ scales with the method's learning rate and the intensity of the noise, and the fraction of the time spent $\delta$-far from the problem's global minimum scales as $\mathcal{O}(\delta_\eta^2/\delta^2)$.

3. If the objective function is strongly convex relative to the method's Bregman function, LMF orbits get within $\delta$ of the problem's minimum in $\tilde{\mathcal{O}}(\delta^{-2p/(p-1)})$ time (in terms of both "first-passage" and "last-iterate" guarantees).

Subsequently, we turn to the bona fide, discrete-time setting, and we study different variants of SMD under heavy-tailed noise. To connect with existing results, we first establish an $\tilde{\mathcal{O}}(\varepsilon^{-p/(p-1)})$ ergodic convergence guarantee for relatively smooth convex functions, echoing the bounds of Nemirovski & Yudin [77] and Vural et al. [108] for SMD under a "uniform smoothness" assumption. We then specialize

to functions that are also strongly convex relatively to the method's generating Bregman function: here, we establish the convergence of the method's trajectories in mean square, and as in the continuous-time setting, we show that (*i*) the algorithm's orbits converge at a geometric rate to an "uncertainty ball" whose size scales with the intensity of the noise; and (*ii*) the algorithm attains an $\varepsilon$-optimal point in $\tilde{\mathcal{O}}(\varepsilon^{-1/(p-1)})$ iterations. This is similar to the convergence rate estimates obtained by Liu [57] for stochastic gradient descent and dual averaging with unbounded variance (for a more detailed comparison, see Appendix A), but we are not otherwise aware of any results in the literature comparable to the derived concentration and first-passage estimates.

Importantly, our discrete-time results mirror our continuous-time findings, both qualitatively and quantitatively. In fact, in all cases, the discrete-time bounds decompose into a term stemming from the continuous-time analysis, plus a "discretization" term. This indicates that the proposed Lévy model is a faithful proxy for studying SMD in the heavy-noise regime, thus opening up an important direction for further research in the heavy-noise regime.

## 2 Setup and preliminaries

**2.1. Notation and terminology.** We begin by collecting some basic definitions from convex analysis and optimization. This is intended only to set terminology and notation; for a detailed treatment, see [16, 19, 21, 95].

Throughout our paper, $\mathcal{V}$ will denote a real $n$-dimensional normed space with norm $\|\cdot\|$ (typically an $L^p$-norm). The dual space of $\mathcal{V}$ will be denoted by $\mathcal{Y} \equiv \mathcal{V}^*$, and we will write $\|y\|_* := \sup\{\langle y, x\rangle : \|x\| \leq 1\}$ for the *dual norm* of $y$ in $\mathcal{Y}$, where $\langle y, x\rangle$ denotes the *dual pairing* between $y \in \mathcal{Y}$ and $x \in \mathcal{V}$. Following standard conventions in the field [95, 96], if $f\colon \mathcal{V} \to \mathbb{R} \cup \{\infty\}$ is an extended-real-valued function on $\mathcal{V}$, we will write $\operatorname{dom} f := \{x \in \mathcal{V} : f(x) < \infty\}$ for the *effective domain* of $f$, and $\partial f(x) := \{y \in \mathcal{Y} : f(x') \geq f(x) + \langle y, x' - x\rangle$ for all $x' \in \mathcal{V}\}$ for the *subdifferential* of $f$ at $x$. Any $g \in \partial f(x)$ will be called a *subgradient* of $f$ at $x$, and we will write $\operatorname{dom} \partial f := \{x \in \operatorname{dom} f : \partial f(x) \neq \varnothing\}$ for the *domain of subdifferentiability* of $f$. By standard results [95, Theorem 26.1], we have $\operatorname{ri} \operatorname{dom} f \subseteq \operatorname{dom} \partial f \subseteq \operatorname{dom} f$.

Finally, we will use the "soft-oh" notation $\tilde{\mathcal{O}}(g(n))$ as a shorthand for $\mathcal{O}(g(n) \log^k n)$ for some $k \geq 0$, and the notation $\bar{\mathcal{O}}(g(n))$ as a shorthand for $\mathcal{O}(g(n)^{1+o(1)})$.

**2.2. Problem setup.** In the rest of our paper, we focus on convex minimization problems of the form

$$\begin{aligned} \text{minimize} \quad & f(x) \\ \text{subject to} \quad & x \in \mathcal{X} \end{aligned} \tag{Opt}$$

where $\mathcal{X}$ is a compact convex subset of $\mathcal{V}$ and $f\colon \mathcal{V} \to \mathbb{R} \cup \{\infty\}$ is a convex function satisfying the following standing assumptions:

(a) *Continuity:* $f$ is finite and continuous on $\mathcal{X}$.

(b) *Smoothness:* The subdifferential $\partial f$ of $f$ admits a *continuous selection*, i.e., a continuous mapping $\nabla f$ such that $\nabla f(x) \in \partial f(x)$ for all $x \in \mathrm{dom}\, \partial f$.

*Remark* 1. In a slight abuse of terminology, we will refer to $\nabla f(x)$ as the *gradient* of $f$ at $x$. This *does not* imply—or presume—that $\partial f(x)$ is a singleton; instead, this assumption is intended to capture cases where $\mathrm{dom}\, f$ lies in a lower-dimensional subspace of $\mathcal{V}$—like the probability simplex—but $f$ is smooth along any curve in $\mathcal{X}$. Formally, it means that the directional derivative $f'(x; z) = d/dt|_{t=0^+} f(x + tz)$ of $f$ at $x$ is equal to $\langle \nabla f(x), z \rangle$ for all vectors of the form $z = x' - x$, $x \in \mathrm{dom}\, \partial f$, $x' \in \mathcal{X}$. ◆

We will specialize these assumptions later as needed. For now, we only stress that the domain of subdifferentiability of $f$ need not extend to the boundary of $\mathcal{X}$; in particular, the derivatives of $f$ may be unbounded over $\mathcal{X}$, even though $\mathcal{X}$ is assumed compact. In this regard, the formulation (Opt) is sufficiently flexible to account for a wide range of optimization objectives that arise often in machine learning, signal processing and data science, but which may fail to be differentiable on the boundary of their domain—from Poisson problems to quantum state tomography and beyond [3, 4, 11, 14, 18, 64].

**2.3. Stochastic gradients.** To solve (Opt), we will assume that the optimizer has access to a black-box oracle that returns a stochastic estimate of the gradient of $f$ at the point of interest. Formally, when queried at a $x \in \mathrm{dom}\, \partial f$, this mechanism returns a *stochastic gradient* of the form

$$\mathsf{G}(x; \omega) = \nabla f(x) + \mathsf{U}(x; \omega) \tag{SG}$$

where $(i)$ $\omega$ is a random seed drawn from some (complete) probability space $(\Omega, \mathcal{F}, \mathbb{P})$; and $(ii)$ $\mathsf{U}(x; \omega)$ denotes an umbrella error term that captures all sources of randomness and uncertainty in the oracle.

In practice, $\mathsf{G}$ is queried repeatedly at a sequence of points $x_t \in \mathcal{X}$, $t = 1, 2, \ldots$, with a sequence of seeds $\omega_t$ drawn i.i.d. from $\Omega$ according to $\mathbb{P}$. In so doing, we obtain the sequence of stochastic gradients

$$g_t = \mathsf{G}(x_t; \omega_t) = \nabla f(x_t) + U_t \tag{1}$$

where $U_t$ is a martingale difference sequence measuring the gradient noise at step $t$. For book-keeping purposes, we denote by $\mathcal{F}_t$ the history (adapted filtration) of $x_t$; in particular, $x_t$ is $\mathcal{F}_t$-measurable, while $\omega_t$, $g_t$, and $U_t$ are not.

With all this in hand, our standing assumptions for the gradient noise term $U_t = \mathsf{U}(x_t; \omega_t)$ will be as follows:

(a) *Unbiasedness:* $\mathbb{E}[\mathsf{U}(x; \omega)] = 0$ for all $x \in \mathrm{dom}\, \partial f$.

(b) *Bounded $p$-th central moments:* $\mathbb{E}[\|\mathsf{U}(x; \omega)\|_*^p] \leq \sigma^p$ for some $p \in (1, 2]$ and all $x \in \mathrm{dom}\, \partial f$.

Importantly, for $p < 2$, this assumption allows for heavy-tailed gradient input with possibly infinite variance. For concreteness, we will refer to the case $p < 2$ as the *heavy-noise* regime, and to the case $p = 2$ as the *tame regime*.

As we noted in the introduction, the heavy-noise regime is particularly relevant for applications to deep learning [10], attention models [2], and reinforcement learning [33], where empirical studies have shown that stochastic gradients are typically heavy-tailed. The assumption we employ above is the standard surrogate for heavy-noise models in the literature, cf. [77, 117] and references therein.

**2.4. Mirror descent and its friends.** In the rest of our paper, we will focus on the seminal *mirror descent* (MD) class of algorithms introduced by Nemirovski [76] and Nemirovski & Yudin [77] as a general scheme for solving constrained optimization problems of the form (Opt). This classical family of first-order methods replaces gradient steps by a "mirror step", that is, a non-Euclidean projection step which is tailored to the geometry of the problem under study. The main ingredients of this template are as follows:

**Definition 1.** Let $h\colon \mathcal{V} \to \mathbb{R} \cup \{\infty\}$ be a convex function with $\mathrm{dom}\, h = \mathcal{X}$, and let $\mathcal{X}_h \equiv \mathrm{dom}\, \partial h$. We say that $h$ is a *Bregman regularizer* on $\mathcal{X}$ if:

(a) The subdifferential $\partial h$ of $h$ admits a continuous selection $\nabla h(x) \in \partial h(x)$ for all $x \in \mathcal{X}_h$.

(b) $h$ is *strongly convex* relative to $\|\cdot\|$ on $\mathcal{X}$, viz.

$$h(x') \geq h(x) + \langle \nabla h(x), x' - x \rangle + \frac{K}{2} \|x' - x\|^2 \tag{2}$$

for some $K > 0$ and all $x \in \mathcal{X}_h$, $x' \in \mathcal{X}$.

We also define the *Bregman divergence* of $h$ as

$$D(q, x) = h(q) - h(x) - \langle \nabla h(x), q - x \rangle \tag{3}$$

and the associated *prox-mapping* as

$$P_x(w) = \arg\min_{u \in \mathcal{X}} \{\langle w, x - u \rangle + D(u, x)\} \tag{4}$$

for all $q \in \mathcal{X}$, $x \in \mathcal{X}_h$, and all $w \in \mathcal{Y}$. Finally, we define the *mirror map* induced by $h$ as

$$Q(y) = \arg\max_{x \in \mathcal{X}} \{\langle y, x \rangle - h(x)\} \tag{5}$$

for all $y \in \mathcal{Y}$. ◆

With all this in hand, the *stochastic mirror descent* (SMD) algorithm unfolds iteratively as

$$x_{t+1} = P_{x_t}(-\gamma_t g_t) \tag{SMD}$$

where $g_t$, $t = 1, 2, \ldots$, is a sequence of stochastic gradients of the form (1), and $\gamma_t > 0$ is a variable step-size sequence, typically of the form $\gamma_t \propto 1/t^{\ell_\gamma}$ for some $\ell_\gamma \in [0, 1]$.

Despite the method's remarkable and well-documented success—see e.g., [12, 21, 52] and references therein—(SMD) exhibits the somewhat strange feature that "new gradient information enters the algorithm with decreasing weights". This observation was first articulated in this way by Nesterov [81], who used it as the starting point for the variant *stochastic dual averaging* (SDA) scheme

$$\begin{aligned} y_{t+1} &= y_t - g_t \\ x_{t+1} &= Q(\eta_{t+1} y_{t+1}) \end{aligned} \quad \text{(SDA)}$$

where $\eta_t > 0$ is a variable nonincreasing *learning rate* sequence that *post-multiplies* the aggregation of gradient steps $y_{t+1} \leftarrow y_t - g_t$ in the dual—hence the name, see [111].

More generally, Nesterov [81] considered the template

$$\begin{aligned} y_{t+1} &= y_t - \gamma_t g_t \\ x_{t+1} &= Q(\eta_{t+1} y_{t+1}) \end{aligned} \quad \text{(SDA}_{\gamma, \eta})$$

which includes both a *pre-* and a *post-*multiplier of gradient steps. When run with constant learning rate $\eta_t \leftarrow 1$ instead of a constant step-size, this template specializes to the iterative method known as *stochastic mirror descent with lazy updates*—or *lazy mirror descent* [21, 100, 121]—namely

$$\begin{aligned} y_{t+1} &= y_t - \gamma_t g_t \\ x_{t+1} &= Q(y_{t+1}) \end{aligned} \quad \text{(LMD)}$$

Importantly, when $h$ is *steep*—in the sense that $\mathcal{X}_h = \text{ri } \mathcal{X}$, so $\nabla h$ blows up at the boundary of $\mathcal{X}$—the methods (SMD) and (LMD) are equivalent; for completeness, we state and prove this fact formally in Appendix B.

*Remark* 2. At a high level, dual averaging provides a more flexible algorithmic template than mirror descent, all the while avoiding the inverse recency bias. Discounting more recent gradient feedback in favor of older, more stale input could prove catastrophic in the presence of heavy-tailed noise, so we will focus primarily on (SDA) in the sequel. ✦

## 3   Analysis and results in continuous time

In this section, we introduce and analyze a general class of stochastic dynamics modeling the heavy noise regime in continuous time. Specifically, we consider a stochastic differential equation (SDE) driven by *Lévy noise*, a stochastic process with independent stationary increments and typically discontinuous paths. Beyond its intrinsic interest, this continuous-time formulation serves as a principled stepping stone for understanding the convergence of (SDA) under heavy noise, as it allows us to focus on the interplay between infinite-variance perturbations and unbounded jumps.

To streamline our presentation, the proofs of all results in this section are collected in Appendices C and D.

**3.1. The Lévy mirror flow.**   In the absence of noise, a natural continuous-time model for mirror descent is given by the *mirror flow* (MF) of $f$, defined here via the dynamics

$$\dot{Y}(t) = -\nabla f(X(t)) \quad \text{with} \quad X(t) = Q(\eta(t) Y(t)) \quad \text{(MF)}$$

where $\eta > 0$ is a nonincreasing, differentiable learning rate function, as per (SDA). Following [50, 70, 91], a standard way to account for noisy gradient observations in (MF) is to consider the associated *stochastic mirror flow*

$$\begin{aligned} dY(t) &= -\nabla f(X(t))\, dt + dM(t) \\ X(t) &= Q(\eta(t) Y(t)) \end{aligned} \quad \text{(SMF)}$$

where $M(t) \in \mathcal{Y}$, $t \geq 0$, denotes a continuous square-integrable martingale. As such, (SMF) can be seen as a rigorous formulation of the informal Langevin dynamics

$$\dot{Y}(t) = -\nabla f(X(t)) + \text{"noise"} \quad (6)$$

with $M(t)$ playing the role of a catch-all, "white noise" term intended to capture all sources of randomness and uncertainty in the gradient signal $\nabla f(X(t))$.

Owing to its simplicity and flexibility, (SMF) has been studied extensively in the literature—see e.g., [48, 50, 70, 91] for applications to optimization, [26, 37, 56, 118] for sampling, and [20, 23, 47, 49, 61, 62, 66, 67, 69, 71, 73] for variational inequalities, game theory, and beyond. At the same time however, the driving noise process $M(t)$ of (SMF) is inherently "tame":(*i*) its increments are Gaussian, and hence, light-tailed; (*ii*) their accumulation leads to a finite-variance process ($\mathbb{E}[\|M(t)\|_*^2] < \infty$ for all $t$); and (*iii*) as a result, the sample paths of (SMF) remain (uniformly) continuous.

To overcome these limitations of (SMF), we will instead consider the *Lévy mirror flow*

$$\begin{aligned} dY(t) &= -\nabla f(X(t))\, dt + dL(t) \\ X(t) &= Q(\eta(t) Y(t)) \end{aligned} \quad \text{(LMF)}$$

where, echoing our assumptions for (SG), the process $L(t)$ is a $\mathcal{Y}$-valued centered Lévy process with $\mathbb{E}[\|L(t)\|_*^p] < \infty$ for some $p \in (1, 2]$. For completeness, we provide all necessary background on Lévy processes in Appendix C. For the moment, we only note that, for $p < 2$, (LMF) should be seen as a formal version of the dynamics

$$\dot{Y}(t) = -\nabla f(X(t)) + \text{"heavy noise"} \quad (7)$$

in the sense that(*i*) the increments of $L(t)$ are no longer Gaussian but heavy-tailed; (*ii*) the variance of $L(t)$ may be infinite; and (*iii*) the sample paths of (SMF) may exhibit jump discontinuities of arbitrary magnitude.

This non-diffusive behavior is much closer to what one would intuitively expect from the "heavy noise" regime. Formally, referring to Appendix C for a more detailed treatment, a centered Lévy process $L(t)$ is a stochastic process with $L(0) = 0$, independent stationary increments, and sample paths that are right continuous with left limits / *continue à droite, limite à gauche* (càdlàg). In particular, by the Lévy-Itô decomposition theorem [5, Theorem 2.4.16], the Lévy noise process in (LMF) can be decomposed as

$$L(t) = M(t) + S(t) + U(t) \qquad (8)$$

where:

(*a*) $M(t)$ is a continuous square-integrable martingale of the form

$$M_i(t) = \sum_{j=1}^{m} \Sigma_{ij} W_j(t) \qquad (9)$$

where $W(t)$ is a standard Brownian motion in $\mathbb{R}^m$ and $\Sigma \in \mathbb{R}^{n \times m}$ is a diffusion matrix capturing possible correlations in the components of $M_i(t)$; by this token, we refer to $M(t)$ as the *diffusion component* of $L(t)$.

(*b*) $S(t)$ is a purely discontinuous càdlàg square-integrable martingale with jumps bounded by some fixed—but otherwise *arbitrary*—positive constant $c > 0$; we refer to it as the *short jump component* of $L(t)$.

(*c*) $U(t)$ is a purely discontinuous càdlàg martingale with finite $p$-th moments and *unbounded* jumps; we call it the *unbounded jump component* of $L(t)$.

Importantly, a Lévy process has infinite variance only if it exhibits unbounded jumps, so (LMF) concurrently allows the analysis of two important phenomena that cannot occur in (SMF): heavy tails and long, unbounded jumps.

The distribution of these jumps is encoded by the *Lévy measure* of $L(t)$, a measure $\nu$ on $\mathbb{R}^n \backslash \{0\}$ which counts how many jumps of a given magnitude occur in unit time, viz.

$$\nu(\mathbb{B}) = \mathbb{E}[\#\{0 \le t \le 1 : \Delta L(t) \in \mathbb{B}\}], \qquad (10)$$

where $\mathbb{B}$ is a Borel subset of $\mathbb{R}^n \backslash \{0\}$, and

$$\Delta L(t) := L(t) - L(t-) \qquad (11)$$

is the jump process of $L(t)$.[1] With this in mind, the intensity of each component of $L(t)$ can be quantified as follows:

(*a*) *Diffusion component:*

$$\sigma_0^2 = \|\Sigma\|_F^2 = \text{tr}[\Sigma^\top \Sigma]. \qquad (12a)$$

(*b*) *Short jump intensity:*

$$\sigma_{\text{short}}^2 = \int_{0 < \|z\|_* < c} \|z\|_*^2 \, \nu(dz). \qquad (12b)$$

(*c*) *Long jump intensity:*

$$\sigma_{\text{long}}^p = \int_{\|z\|_* \ge c} \|z\|_*^p \, \nu(dz). \qquad (12c)$$

Finally, with a fair amount of hindsight, we also set

$$\sigma_{\text{tame}}^2 = \sigma_0^2 + \sigma_{\text{short}}^2 \quad \text{and} \quad \sigma_{\text{heavy}}^p = \sigma_{\text{long}}^p \qquad (12d)$$

for the tame and heavy part of the noise respectively.

Each of these quantities controls a specific part of the noise process $L(t)$. Clearly, the diffusive regime of (SMF) is recovered when $\sigma_{\text{short}} = \sigma_{\text{long}} = 0$, i.e., when $L(t)$ has no jump component. We elaborate on all this in Appendix C, where we also discuss in detail the well-posedness of (LMF)—that is, the existence and uniqueness of strong solutions from every initial condition $Y(0) \in \mathcal{Y}$.

**3.2. Analysis and results.** We now turn our focus to the convergence guarantees of (LMF) in the context of (Opt). To lighten notation, we will assume in the rest of this section that (LMF) is initialized at the *prox-center* $x_c = \arg\min h$ of $\mathcal{X}$, i.e., with $Y(t) \leftarrow 0$.

To provide some background, we begin with the tame, diffusive regime of (SMF), where [70] showed that the time-averaged process $\bar{X}(t) = (1/t) \int_0^t X(s) \, ds$ enjoys the bound

$$\mathbb{E}[f(\bar{X}(t)) - \min f] = \mathcal{O}(1/\sqrt{t}). \qquad (13)$$

The analysis of [70] hinges on an application of Itô's formula—the chain rule of stochastic calculus [51, 86]—to a carefully designed energy function. To define it, consider first the *Fenchel coupling* [70]

$$F(q, y) = h(q) + h^*(y) - \langle y, q \rangle \qquad \text{for } q \in \mathcal{X}, y \in \mathcal{Y}, \ (14)$$

where $h^*(y) = \max_{u \in \mathcal{X}}\{\langle y, u \rangle - h(u)\}$ denotes the convex conjugate of $h$. In a sense made precise in Appendix B, the Fenchel coupling provides a measure of "primal-dual" divergence between $q \in \mathcal{X}$ and $y \in \mathcal{Y}$, which is compatible with the Bregman setup of Definition 1. Accordingly, to account for the scaling of $Y(t)$ by $\eta$ in (SMF), a natural choice of energy function is the $\eta$-*deflated* Fenchel coupling

$$E(t) = \frac{1}{\eta(t)} F(q, \eta(t) Y(t)). \qquad (15)$$

The first challenge that arises in studying the evolution of $E(t)$ under (LMF) is that the analogue of Itô's lemma for Lévy-type processes includes second-order jump terms corresponding to the discontinuities of $L(t)$. Thus, given that the unbounded jump component $U(t)$ of $L(t)$ may have infinite variance when $p < 2$, we cannot use crude estimates based on smoothness arguments to control these terms as in the diffusive case.

---

[1] Here and throughout, we write $u(t-) := \lim_{s \to t^-} u(s)$ for the left limit of a càdlàg function $u(t)$.

A second major difficulty is that, in general, the Fenchel coupling is Lipschitz smooth *but not $C^2$-smooth*, so it must first be mollified in order to derive a "weak" version of Itô's lemma that only holds as an inequality (as opposed to an equality). Putting all this together is a delicate task whose end result is Theorem C.1, a "weak Itô formula" for Lipschitz-smooth convex functions relative to general Lévy-type integrators. To the best of our knowledge, this result is new in the stochastic analysis literature; however, because the statement itself requires additional technical machinery, we defer the formal presentation and proof to Appendix C.6.

Instead, we only present here the application of this weak Itô formula to $E(t)$. To that end, if we set with some hindsight

$$V_{\text{tame}}^2 = \frac{\sigma_{\text{tame}}^2}{2K} \quad \text{and} \quad V_{\text{heavy}}^p = \frac{\|\mathcal{X}\|^{2-p}\sigma_{\text{heavy}}^p}{K^{p-1}}, \quad (16)$$

we obtain the following explicit bound:

**Proposition 1.** *Under* (LMF), *$E(t)$ is bounded as*

$$E(t) - E(0) \le \int_0^t \langle \nabla f(X(s)), q - X(s)\rangle \, ds + \xi(t) + \zeta(t)$$

$$+ V_{\text{tame}}^2 \int_0^t \eta(s) \, ds + V_{\text{heavy}}^p \int_0^t \eta(s)^{p-1} \, ds$$

$$- \int_0^t \frac{\dot\eta(s)}{\eta(s)^2}[h(q) - h(X(s))] \, ds, \quad (17)$$

*where $\|\mathcal{X}\| := \max_{x,x'}\|x' - x\|$ is the diameter of $\mathcal{X}$, and $\xi(t)$, $\zeta(t)$ are martingales (the former square-integrable).*

We prove Proposition 1 in Appendix D, where we also provide explicit expressions and bounds for $\xi$ and $\zeta$. Thanks to these bounds, we get the following guarantee for (LMF):

**Theorem 1.** *Under* (LMF), *the time-averaged process $\bar{X}(t) = (1/t)\int_0^t X(s) \, ds$ enjoys the bounds*

$$\mathbb{E}[f(\bar{X}(t))] \le \min f + R_0(t) \quad (18)$$

*and, with probability* 1,

$$f(\bar{X}(t)) \le \min f + R_0(t) + \bar{\mathcal{O}}\left(t^{-\frac{p-1}{p}}\right)$$

$$+ \mathcal{O}\left(\sigma_{\text{tame}}\|\mathcal{X}\|\sqrt{\frac{\log\log t}{t}}\right) \quad (19)$$

*where*

$$R_0(t) = \frac{H}{t\eta(t)} + \frac{V_{\text{tame}}^2}{t}\int_0^t \eta(s) \, ds$$

$$+ \frac{V_{\text{heavy}}^p}{t}\int_0^t \eta(s)^{p-1} \, ds \quad (20)$$

*and $H := \max h - \min h$. In particular, if* (LMF) *is run with learning rate $\eta(t) = 1/t^{1/p}$, we have*

$$R_0(t) = \frac{H}{t^{(p-1)/p}} + \frac{p}{p-1}\frac{V_{\text{tame}}^2}{t^{1/p}} + p\frac{V_{\text{heavy}}^p}{t^{(p-1)/p}}$$

$$= \mathcal{O}(1/t^{(p-1)/p}). \quad (21)$$

Theorem 1 is our main result for convex functions, so some remarks are in order.

*Remark* 3. From the expression (21) for $R_0(t)$, we further observe that $f(\bar{X}(t)) \to \min f$ whenever the learning rate satisfies $t\eta(t) \to \infty$ and $\eta(t) \to 0$. Among such choices, the standard schedule $\eta(t) \propto 1/t^{1/p}$ is the one that optimizes the resulting upper bound in $t$. To connect with existing results, this gives $\mathbb{E}[f(\bar{X}(t))] - \min f = \mathcal{O}(1/\sqrt{t})$ in the tame regime $p = 2$, so we recover the diffusive expression (13) for (SMF) under Brownian noise [70]. On the other hand, under heavy noise ($1 < p < 2$), the convergence rate of (LMF) deteriorates smoothly with $p$, matching the corresponding lower bounds of [77]. This deterioration is due expressly to the long-jump term $\mathcal{O}(\sigma_{\text{heavy}}^p/t^{(p-1)/p})$ in (21), and suggests that the Lévy-driven dynamics (LMF) comprise a particularly expressive model for the study of stochastic mirror descent under heavy-tailed noise. ✦

*Remark* 4. We also note that, for any error $\varepsilon > 0$, Theorem 1 shows that running (LMF) with constant $\eta = \mathcal{O}(\varepsilon^{1/(p-1)})$ guarantees $\mathbb{E}[f(\bar{X}(t)) - \min f] \le \varepsilon$ within time $t = \mathcal{O}(\varepsilon^{-p/(p-1)})$. This expression in terms of $\varepsilon$ provides a convenient baseline for comparing convergence rates under constant learning rates; we revisit this point later. ✦

### 3.3. Finer results under (relative) strong convexity.

In the rest of this section, we refine and expand on the convergence guarantees of Theorem 1 for functions that are *strongly convex* relative to either $\|\cdot\|$ or $h$. Formally:

(i) $f$ is *$\mu$-strongly convex* if

$$f(x') \ge f(x) + \langle \nabla f(x), x' - x\rangle + \frac{\mu}{2}\|x' - x\|^2 \quad (22)$$

for all $x, x' \in \mathcal{X}$.

(ii) $f$ is *$\mu$-strongly convex relative to $h$* if

$$f(x') \ge f(x) + \langle \nabla f(x), x' - x\rangle + \mu D(x', x) \quad (23)$$

for all $x \in \mathcal{X}_h$ and all $x' \in \mathcal{X}$.

The notion of relative strong convexity is due to Lu et al. [64], and is a refinement of "vanilla" strong convexity: if $f$ is $\mu$-strongly convex relative to $h$, then it is also $(\mu K)$-strongly convex relative to $\|\cdot\|$. The converse, however, does not hold, especially when $h$ is *steep* (that is, $\mathcal{X}_h = \text{ri}\,\mathcal{X}$). In this case, the Bregman divergence $D(x', x)$ may grow unbounded as $x$ gets closer to the boundary of $\mathcal{X}$; in turn, this means that the gradient of $f$ blows up at the boundary of $\mathcal{X}$, so a strongly convex function like $\|x\|^2$ cannot be strongly convex relative to $h$ in this case. We discuss this issue in more detail later; for now, it suffices to keep in mind that the notion of relative strong convexity is more exacting, but also more aligned with the geometry of $h$ (so one may reasonably expect better guarantees in this setting).

With all this in hand, we will examine the following questions for (LMF) under (relative) strong convexity:

(a) The fraction of time that $X(t)$ spends near $\arg\min f$.

(b) The time it takes $X(t)$ to get close to $\arg\min f$.

(c) The convergence speed of $X(t)$ itself (as opposed to its time average).

To quantify the above, fix a distance tolerance $\delta > 0$, let $x^* \in \arg\min f$ denote a solution of (Opt), and let

$$\mathbb{B}_\delta = \{x \in \mathcal{X} : \|x - x^*\| \le \delta\} \tag{24}$$

be a ball of radius $\delta$ centered at $x^* \in \mathcal{X}$. Finally, with some hindsight, consider the "uncertainty radius"

$$\delta_\eta^2 = \frac{2}{\mu}\left[\eta V_{\text{tame}}^2 + \eta^{p-1} V_{\text{heavy}}^p\right]. \tag{25}$$

We then have the following guarantees for (LMF).

**Theorem 2** (Concentration). *Assume $f$ is $\mu$-strongly convex, and let*

$$\mu_t(\mathbb{B}_\delta) = \frac{1}{t}\int_0^t \mathbb{1}\{X(s) \in \mathbb{B}_\delta\}\, ds \tag{26}$$

*be the fraction of time that $X(t)$ spends in $\mathbb{B}_\delta$ up to time $t$. If (LMF) is run with constant $\eta(t) \equiv \eta$, we have*

$$\mathbb{E}[\mu_t(\mathbb{B}_\delta)] \ge 1 - \frac{\delta_\eta^2}{\delta^2} - \frac{2H}{\eta\mu\delta^2 t} \tag{27a}$$

*and, with probability 1,*

$$\mu_t(\mathbb{B}_\delta) \ge 1 - \frac{\delta_\eta^2}{\delta^2} - \frac{2H}{\eta\mu\delta^2 t} + \bar{\mathcal{O}}\left(t^{-\frac{p-1}{p}}\right) + \mathcal{O}\left(\sigma_{\text{tame}}\|\mathcal{X}\|\sqrt{\frac{\log\log t}{t}}\right). \tag{27b}$$

**Theorem 3** (Hitting time). *Assume $f$ is $\mu$-strongly convex, and let*

$$\tau_\delta = \inf\{t \ge 0 : \|X(t) - x^*\| \le \delta\} \tag{28}$$

*denote the first time that $X(t)$ gets within $\delta$ of $x^*$ for some small enough $\delta > 0$. We then have the following estimates:*

**Case 1.** *Heavy noise: If $p < 2$ and (LMF) is run with $\eta(t) = \left[\mu\delta^2/(8V_{\text{heavy}}^p)\right]^{1/(p-1)}$, then*

$$\mathbb{E}[\tau_\delta] \le 4H\left(\frac{8^{1/p}V_{\text{heavy}}}{\mu\delta^2}\right)^{p/(p-1)}. \tag{29}$$

**Case 2.** *Tame noise: If $p = 2$ and (LMF) is run with $\eta(t) = \mu\delta^2 \cdot K/(2\sigma_{\text{tame}}^2 + 4\sigma_{\text{heavy}}^2)$, then*

$$\mathbb{E}[\tau_\delta] \le \frac{8H}{K}\frac{\sigma_{\text{tame}}^2 + 2\sigma_{\text{heavy}}^2}{\mu^2\delta^4}. \tag{30}$$

**Theorem 4** (Convergence). *Assume $f$ is $\mu$-strongly convex relative to $h$ with $h$ steep. Then:*

(a) *If (LMF) is run with constant $\eta(t) \equiv \eta$, we have:*

$$\mathbb{E}[\|X(t) - x^*\|^2] \le \delta_\eta^2/K + (2H/K)e^{-\mu\eta t}. \tag{31}$$

(b) *If (LMF) is run with $\eta(t) = (1+t)^{-1/p}$, we have:*

$$\mathbb{E}[\|X(t) - x^*\|^2] \le \frac{2}{\mu K}\frac{H}{pt^{(p-1)/p}}$$
$$+ \frac{2}{\mu K}\frac{V_{\text{tame}}^2}{t^{1/p}} + \frac{2}{\mu K}\frac{V_{\text{heavy}}^p}{pt^{(p-1)/p}}$$
$$+ \frac{2H}{K}\exp\left(-\mu p\frac{t^{1-1/p} - 1}{p - 1}\right). \tag{32}$$

*In particular, $X(t) \to x^*$ in mean square as $t \to \infty$.*

Theorems 2–4 are our main results for (LMF) in the strongly convex regime, so we close this section with some remarks.

*Remark 5.* Note that Theorem 4 is stated in the context of *relative*—as opposed to ordinary—strong convexity. This is due to the fact that Grönwall-type arguments for the iterates of mirror descent (as opposed to time averages) require a two-way compatibility between norms and divergences, and this is absent when $f$ is norm-strongly convex. This disparity is well-documented in the literature—for both stochastic [8, 63] and deterministic methods [9, 64]—so it is natural that (LMF) is also subject to it. For completeness, we also record in Appendix D an ergodic convergence result under "vanilla" strong convexity, cf. Theorem D.3. ✦

*Remark 6.* It is also worth noting that Theorems 2–4 paint a complementary picture. At a high level, Theorem 2 shows that, for any fixed $\eta$, there is a "critical radius" $\delta_c \equiv \delta_c(\eta, \mu, \sigma)$ such that, after a transient $\mathcal{O}(1/t)$ regime, $X(t)$ is concentrated within $\delta_c$ of $x^*$, spending only an $\mathcal{O}(\delta_c^2/\delta^2)$ fraction of time $\delta$-away from $x^*$. This is reflected in the guarantee (31) which shows the geometric convergence of $X(t)$ to an $\mathcal{O}(\delta_c)$-neighborhood of $x^*$ before the noise overshadows the drift of the process. In particular, letting $t \to \infty$ in Theorem 4 yields $\limsup_{t\to\infty} \mathbb{E}[\|X(t) - x^*\|^2] \le \delta_c^2/K$, matching the upper bound obtained from Theorem D.3, but applies to the *last iterate* rather than to ergodic averages (and is reached at a geometric rate under relative strong convexity, instead of $1/t$ for standard strong convexity). ✦

*Remark 7.* As in the case of Theorem 1, running (LMF) with $\eta = \mathcal{O}(\varepsilon^{1/(p-1)})$ for a fixed tolerance level $\varepsilon > 0$ guarantees $\mathbb{E}[\|X(t) - x^*\|^2] \le \varepsilon$ within time $t = \mathcal{O}(\varepsilon^{-1/(p-1)}\log(1/\varepsilon))$. Since $p > 1$, this is always sharper than the (ergodic) rate $\mathcal{O}(\varepsilon^{-p/(p-1)})$ of Theorems 1 and D.3; in particular, in the tame regime $p = 2$, this rate matches the well-known $\mathcal{O}(1/\varepsilon)$ rate for stochastic strongly convex optimization up to a logarithmic factor. However,

for $p < 2$, this bound does not match the $\Omega(\varepsilon^{-p/[2(p-1)]})$ lower bound [117, Theorem 5] for smooth strongly convex functions. We discuss this in detail in the next section. ✦

## 4 Analysis and results in discrete time

With the continuous-time analysis in place, we return in this section to the discrete-time algorithmic setup of (SDA), where we show that, mutatis mutandis, the same convergence and concentration patterns persist. The only additional element that we will require over Section 3 is a measure of smoothness for $f$ that is compatible with the Bregman geometry of $h$. This is mandated for two reasons: First, under relative strong convexity, gradients may grow unbounded near the boundary of $\mathcal{X}$, so standard Lipschitz smoothness does not suffice. Second, the heavy noise in the input is filtered through the mirror map $Q$, so, in order to control it in the primal space $\mathcal{X}$, we need a notion of smoothness which, in a sense, commutes with $Q$.

To that end, we will employ the notion of *relative smoothness* due to [11, 17, 64]. Formally, $f$ is said to be *L-smooth relative to h* if $Lh - f$ is convex, or, equivalently, if

$$f(x') \le f(x) + \langle \nabla h(x), x' - x \rangle + LD(x', x) \quad \text{(RS)}$$

for all $x \in \mathcal{X}_h$, $x' \in \mathcal{X}$. Dually to strong convexity, if a function is $L$-smooth relative to $\|\cdot\|$—that is, (RS) holds with $\|x' - x\|^2/2$ in lieu of $D(x', x)$—then it is also $(L/K)$-smooth relative to $h$. The converse however does not hold, and there is a broad range of problems in machine learning and data science—from Poisson inverse problems to entropically regularized optimal transport—that are smooth relative to a suitable Bregman function, but not in the ordinary sense.

With all this in hand, we state below a series of representative results for different incarnations of SMD under heavy noise. For convenience, we write throughout

$$V_\sigma^p = \|\mathcal{X}\|^{2-p} (2/K)^{p-1} \sigma^p \quad (33)$$

in direct analogy with $V_{\text{heavy}}$ in (16). In addition, to streamline our presentation, we assume that all algorithms are initialized at the prox-center $x_c = \arg\min h$ of $\mathcal{X}$, i.e., with $y_1 \leftarrow 0$, and we defer all proofs to Appendix E.

We start with two results in the spirit of Theorem 1.

**Theorem 5** (SDA, convex). *Suppose that $f$ is $L$-smooth relative to $h$ and* (SDA) *is run with learning rate $\eta_t = \beta/t^{1/p}$, $\beta \le 1/(pL)$. Then the time-averaged process $\bar{x}_T = (1/T)\sum_{t=1}^T x_t$ enjoys the bound*

$$\mathbb{E}[f(\bar{x}_T)] \le \min f + \frac{f(x_1) - \min f}{T} + \frac{HK^{p-1} + \beta^p V_\sigma^p}{\beta T^{(p-1)/p}} .$$

*In particular, for $\beta = [H/(p-1)]^{1/p}/V_\sigma$ we have*

$$\mathbb{E}[f(\bar{x}_T)] \le \min f + \frac{f(x_1) - \min f}{T}$$
$$+ \frac{p[H/(p-1)]^{(p-1)/p} V_\sigma}{T^{(p-1)/p}} . \quad (34)$$

Our next result concerns the relatively strongly convex case—so $\arg\min f = \{x^*\}$ for some $x^* \in \mathcal{X}$. Then:

**Theorem 6** (SDA, strongly convex). *Suppose that $f$ is $\mu$-strongly convex and $L$-smooth, both relative to $h$, with $h$ steep. If* (SDA) *is run with constant $\eta_t \equiv \eta \le 1/(pL)$, then*

$$\mathbb{E}[\|x_t - x^*\|^2] \le \frac{\eta^{p-1}V_\sigma^p}{p\mu K} + \frac{2H}{K}(1 - \eta\mu)^{t-1}. \quad (35)$$

**Corollary 1.** *Fix some target accuracy $\varepsilon > 0$. If* (SDA) *is run with small enough $\eta = \mathcal{O}(\varepsilon^{1/p}\sigma)$, then $\mathbb{E}[\|x_t - x^*\|^2] \le \varepsilon$ for all $t = \Omega((\sigma^p/\varepsilon)^{1/(p-1)} \log(1/\varepsilon))$.*

Our next two results echo Theorems 2 and 3, and they estimate the fraction of time that $x_t$ spends away from $\arg\min f$, and the time that it takes to get close to it for the first time.

**Theorem 7** (Concentration). *Assume $f$ is $\mu$-strongly convex and $L$-smooth relatively to $h$, let $\mathbb{B}_\delta$ be a $\delta$-ball in $\mathcal{X}$ around the global minimum $x^*$ of $f$, and let*

$$\mu_T(\mathbb{B}_\delta) = \frac{1}{T} \sum_{t=1}^T \mathbb{1}\{x_t \in \mathbb{B}_\delta\} \quad (36)$$

*be the fraction of time that $x_t$ spends in $\mathbb{B}_\delta$ up to time $T$. If* (SDA) *is run with constant learning rate $\eta_t \equiv \eta$, we have*

$$\mathbb{E}[\mu_T(\mathbb{B}_\delta)] \ge 1 - \frac{\delta_\eta^2}{\delta^2} - \frac{2[f(x_1) - \min f]}{\mu\delta^2 T} - \frac{2H}{\eta\mu\delta^2 T} \quad (37)$$

*where, in a similar vein to (38), we let*

$$\delta_\eta^2 = 2/(\mu p) \cdot \eta^{p-1} V_\sigma^p . \quad (38)$$

**Theorem 8** (Hitting time). *Assume $f$ is $\mu$-strongly convex and $L$-smooth relatively to $h$, and let*

$$\tau_\delta = \inf\{t = 1, 2, \dots : \|x_t - x^*\| \le \delta\} \quad (39)$$

*denote the first time that $x_t$ gets within $\delta$ of $x^*$ for some small enough $\delta > 0$. If* (SDA) *is run with constant learning rate $\eta < [\mu p \delta^2/(2V_\sigma^p)]^{1/(p-1)}$, then*

$$\mathbb{E}[\tau_\delta] \le \frac{2}{\mu} \frac{H/\eta + f(x_1) - \min f + \mu\delta^2/2}{\delta^2 - \delta_\eta^2} . \quad (40)$$

*In particular, if* (SDA) *is run with $\eta = [\mu\delta^2/(2V_\sigma^p)]^{1/(p-1)}$, we have*

$$\mathbb{E}[\tau_\delta] \le \frac{p}{p-1}\left[1 + \frac{2[f(x_1) - \min f]}{\mu\delta^2} + H\left(\frac{2V_\sigma}{\mu\delta^2}\right)^{\frac{p}{p-1}}\right]$$
$$= \mathcal{O}(1/\delta^{2p/(p-1)}) . \quad (41)$$

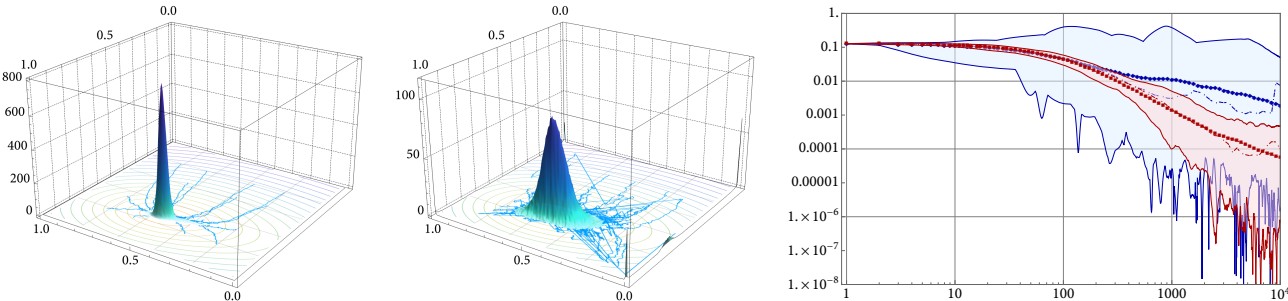

**Figure 1.** The behavior of (SDA) under tame and heavy noise. We considered a strongly convex function over $\mathcal{X} = [0, 1] \times [0, 1]$ and we ran $N = 200$ instances of (SDA) with entropic regularization for $T = 10^4$ iterations and gradients perturbed by a centered $\alpha$-stable distribution with scale parameter $c = 1$ and no skewness. The leftmost plot shows a sample trajectory and the long-run distribution of the iterates of (SDA) for the tame regime ($\alpha = p = 2$); the center plot shows the same information for the heavy-noise regime ($\alpha = p = 3/2$). Finally, the rightmost plot shows the evolution of $f(\bar{x}_T)$ under tame and heavy noise (red and blue coloring respectively; markers = mean, shading = range, and the dashed line represents a sample trajectory); as predicted, $f(\bar{x}_T)$ follows a power law in both cases.

Finally, to complete our range of results, we provide below the analogous result for (LMD) with a variable step-size.

**Theorem 9** (LMD, variable $\gamma$). *Suppose that $f$ is $\mu$-strongly convex and $L$-smooth, both relative to $h$, with $h$ steep. If (LMD) is run with step-size $\gamma_t = \beta/t$, $\beta \leq 1/(pL)$, then*

$$\mathbb{E}\left[\|x_t - x^*\|^2\right] = \begin{cases} \mathcal{O}(1/t^{p-1}) & \text{if } p < 1 + \beta\mu, \\ \mathcal{O}(\log t / t^{p-1}) & \text{if } p = 1 + \beta\mu, \quad (42) \\ \mathcal{O}(1/t^{\beta\mu}) & \text{if } p > 1 + \beta\mu. \end{cases}$$

There are several points worth discussing at this stage.

First, the choice of results to present in this section has not been arbitrary; our aim was to show the broadest range of phenomena that arise. Specifically, by the equivalence of (SDA) and (LMD) for constant $\gamma = \eta$, Theorem 6 also covers (LMD); and by the equivalence of (LMD) and (SMD) in the steep case, the bounds of Theorem 9 also apply directly to (SMD). In Appendix E, we provide a range of supplementary results that complete the picture.

Second, a salient pattern that arises when connecting the corresponding continuous- and discrete-time results is that the derived discrete-time bounds consist of a term that mirrors the corresponding continuous-time guarantee, plus a term involving the value difference $f(x_1) - \min f$, which essentially stems from the "discretization" of (LMF). This suggests that (LMF) is a particularly faithful surrogate for the heavy-noise regime in continuous time—so the study of its properties proves highly beneficial for the discrete-time regime as well.

In terms of rates, the guarantees of Theorem 9 are fairly involved. First, to provide some context, Theorem E.11 in Appendix E shows that running (LMD) with $\gamma_t \propto 1/t^{1/p}$ achieves an $\mathcal{O}(1/t^{(p-1)/p})$ convergence rate. On the surface, this appears weaker than that of Theorem 9, but there are several intricacies at play: First, since $L \geq \mu$, we have $\beta\mu \leq \mu/(pL) \leq 1/p < 1$, so the rates in Theorem 9 are

better when $p \leq 1/(1 - \beta\mu)$, but not otherwise. Concretely, for $\beta = 1/(pL)$, this condition reduces to $p \leq 1 + \mu/L$, which generally holds only when $p$ is close enough to 1, that is, for very heavy noise; in the other cases, the step-size $\gamma_t \propto 1/t^{1/p}$ seems to be preferable.

Finally, we see that (SDA) attains the improved complexity rate $\tilde{\mathcal{O}}\big((\sigma^p/\varepsilon)^{1/(p-1)}\big)$ with a suitably tuned $\eta$ (the case of a variable learning rate is more subtle, cf. Theorem E.7). As we mentioned in Section 3, this does not match the $\mathcal{O}(\varepsilon^{-p/(2p-2)})$ of Zhang et al. [117] for strongly convex functions. A key difference between our setting and [117] is that we work here with functions that are strongly convex relative to a steep regularizer over a bounded domain, so $\nabla f$ grows unbounded at the boundary of $\mathcal{X}$; by contrast, the lower bounds of [117] concern problems whose gradients are bounded over *any* bounded domain, so there is no overlap. Beyond this difference, we conjecture that our rates can be tightened further with a different choice of energy function; we leave this as an open question for future work.

## 5   Concluding remarks

A first important take-away of our results is that stochastic mirror descent remains particularly robust in the presence of heavy noise, and maintains its convergence and concentration guarantees despite the very long jumps exhibited by the noise. A second is that our discrete-time results closely mirror our continuous-time findings, both qualitatively and quantitatively; this indicates that the proposed Lévy model is a faithful proxy for studying SMD in the heavy-noise regime, thus opening up several important directions for further research in the heavy-noise regime. A first such question would be to sharpen Theorem 6 under relative strong convexity, where we conjecture that the lower bound of Zhang et al. [117] is achievable (even with unbounded gradients); a second would be to investigate the use of (LMF) for sampling from constrained spaces in the spirit of [37, 118]. We leave these questions for the future.

## Acknowledgments

This research was supported in part by the French National Research Agency (ANR) in the framework of the PEPR IA FOUNDRY project (ANR-23-PEIA-0003). PM is also a member of the Archimedes Research Unit/Athena RC, and was partially supported by project MIS 5154714 of the National Recovery and Resilience Plan Greece 2.0 funded by the European Union under the NextGenerationEU Program.

## Impact Statement

This paper presents work whose goal is to advance the field of Machine Learning. There are many potential societal consequences of our work, none of which we feel must be specifically highlighted here.

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

# A   Related works

**Stochastic mirror flows and Mirrored Langevin dynamics**   *Stochastic mirror flows* were introduced and studied by Raginsky & Bouvrie [91] as a noisy variant of the continuous-time mirror flow of Nemirovski & Yudin [77], with subsequent works refining their convergence properties and extending the framework to state-dependent diffusion terms and variational inequalities [70, 71]. Accelerated variants were later proposed, notably by Krichene & Bartlett [48] and Xu et al. [113, 114], the latter also establishing convergence rates under relative strong convexity. All these works suppose $L$-smoothness of the objective function. Closely related are the *mirrored Langevin dynamics*, which extend Langevin diffusions to non-Euclidean geometries for sampling measures on constrained spaces. The initial formulation, due to Hsieh et al. [37], considered a constant diffusion coefficient, while later studies introduced a diffusion term adapted to the geometry via the Hessian of the regularizer [26, 56, 118], thereby placing mirrored Langevin dynamics as a special case of Riemannian Langevin methods [94]. Convergence guarantees for convex and relatively strongly convex objectives were also established for these dynamics by Tzen et al. [107] under a constant stepsize schedule.

**Lévy-driven SDEs in stochastic optimization**   Stochastic differential equations driven by Lévy processes have gained attention in machine learning as models for stochastic optimization under heavy-tailed noise. A first line of work focuses on sampling algorithms based on Lévy-driven dynamics, most notably the *fractional Langevin Monte Carlo* method introduced by Şimşekli [101], and further studied by [38, 83, 105, 115, 119], where Brownian motion is replaced by a stable Lévy process. More general *Lévy Langevin Monte Carlo* methods allowing arbitrary Lévy noise were also recently studied in Oechsler [85] and Behme & Schwienhorst [13]. Beyond sampling, Lévy-driven SDEs have been proposed as continuous-time approximations of stochastic gradient methods, notably to explain metastability and escape phenomena observed in deep learning [10, 82, 102, 103, 120]. Finally, Lévy-driven models have also been used to study generalization properties of SGD, suggesting that heavier-tailed processes should achieve better generalization errors [30, 31, 92, 93, 104].

**Heavy-tailed noise in stochastic algorithms**   For *standard SGD*, Zhang et al. [117] showed that iterates may fail to converge for any fixed stepsize under heavy-tailed noise, even for strongly convex objectives, indicating that additional structure is generally required. Convergence can nonetheless be recovered under strong assumptions on the objective, such as a bounded and uniformly $p$-positive definite Hessian [109], or by modifying the algorithm.

In the context of *mirror descent*, Nemirovski & Yudin [77] established convergence for convex objectives with uniformly continuous mirror maps and proved the optimal in-expectation lower bound $\Omega(t^{-(p-1)/p})$. This result was later refined by Vural et al. [108] for uniformly convex regularizers, with improved dimension dependence, and extended to zeroth-order methods in Kornilov et al. [46]. For *projected SGD on bounded domains*, recent work by Fatkhullin et al. [32] and Liu [57] showed that the optimal convex rates can in fact be achieved without modifying the algorithm; the latter also establishing this property for Dual Averaging and AdaGrad in online convex optimization.

A large body of work instead focuses on *gradient clipping* as a stabilization mechanism under heavy-tailed noise. This line of research was initiated by Zhang et al. [117], who proved optimal lower bounds in expectation of order $\Omega(t^{-2(p-1)/p})$ for strongly convex objectives and $\Omega(t^{-(p-1)/(3p-2)})$ for nonconvex smooth ones, and showed that clipped SGD can match these bounds. Subsequent works derived high-probability guarantees matching these rates [97] and extended the analysis to broader settings, including nonconvex [6, 28, 60], composite [34] and distributed optimization [34, 116], as well as to more general $(L_0, L_1)$-smooth objectives [27].

Further refinements include variance-reduced and stabilized variants that can achieve even faster rates under additional structure [58, 90]. The recent work of Wang & Rhee [110] complements this line of work by providing a large-deviations and metastability analysis of clipped SGD without relying on a Lévy-driven SDE approximation. Alternatives to clipping have also been proposed to address heavy-tailed noise without truncation. Notably, normalized gradient methods achieve optimal or near-optimal convergence rates for nonconvex optimization [39, 59, 106, 116].

**Further comparison with [57]**   For discrete-time algorithms, the closest work to ours is [57], which derives several convergence guarantees for Online Gradient Descent and Dual Averaging on bounded domains under heavy-tailed noise. Importantly, while the same general strategy is used to handle unbounded variance (cf. the chain of inequalities leading to [57, Eq. 5]), that work is restricted to standard Euclidean norms, whereas our results hold under general Bregman divergences. This extension to non-Euclidean norms was in fact already alluded to by Liu [57, Remark 4], but carrying it out requires substantially more care than what one might initially expect, notably the need to distinguish between *standard* and *relative* strong convexity—the former being norm-agnostic while the latter accommodates the geometry induced by the

regularizer. In particular, when the regularizer is steep, relative strong convexity requires the objective function to have unbounded gradients, making it necessary to work under the refined *relative smoothness* framework of Bauschke et al. [11]. By contrast, gradients are assumed bounded throughout [57], except for the last-iterate convergence rate of [57, Theorem 10] and its corollaries, which rely instead on the condition

$$\frac{\mu}{2}\|x - x'\|^2 \le f(x) - f(x') - \langle \nabla f(x'), x - x' \rangle \le 2G\|x - x'\| + \frac{H}{2}\|x - x'\|^2, \quad x, x' \in \mathcal{X}, \tag{A.1}$$

for some $\mu, G, H \ge 0$ satisfying $G + H > 0$. One may in fact observe that, in the Euclidean and compact setting of Liu [57], the above condition effectively reduces to a bounded-gradient assumption, despite being formulated as a nonsmoothness condition. However, unlike the other results of [57], its extension to the relatively smooth / relatively strongly convex setting now becomes rather natural, as it effectively amounts to replacing Euclidean norms by Bregman divergences. With that said, this condition was only introduced in a revised version of [57], posted online concurrently with the initial drafting of the present paper, and whose existence was brought to our attention only later by a reviewer.

## B  Basic properties of Bregman methods

In this appendix, we collect some background material, properties and examples regarding the mirror setup of Section 2. The results presented below (or a version thereof) are known in the literature; nevertheless, we provide detailed proofs for completeness and to resolve any conflicts or ambiguities with different conventions in the literature.

**B.1. Refresher.**   With notation as in Section 2, let $\mathcal{V}$ be a $n$-dimensional normed space, let $\mathcal{Y} \coloneqq \mathcal{V}^*$ denote the (algebraic) dual of $\mathcal{V}$, and let $\langle y, x \rangle$ denote the canonical bilinear pairing between $x \in \mathcal{V}$ and $y \in \mathcal{V}^*$. If $\|\cdot\|$ is a norm on $\mathcal{V}$ we will also write

$$\|y\|_* = \max\{\langle y, x \rangle : \|x\| \le 1\} \tag{B.1}$$

for the induced dual norm on $\mathcal{Y}$, so $|\langle y, x \rangle| \le \|x\|\|y\|_*$ for all $x \in \mathcal{V}$ and all $y \in \mathcal{Y}$ by construction. Moreover, given a closed convex subset $\mathcal{X}$ of $\mathcal{V}$, we also define:

1. The *tangent cone* to $\mathcal{X}$ at $q \in \mathcal{X}$:

$$\mathrm{TC}(q) = \mathrm{cl}\{z \in \mathcal{V} : q + tz \in \mathcal{X} \text{ for some } t > 0\} \tag{B.2}$$

   i.e., as the closure of the set of rays emanating from $q$ and meeting $\mathcal{X}$ in at least one other point.

2. The *dual cone* to $\mathcal{X}$ at $q \in \mathcal{X}$:

$$\mathrm{TC}^*(q) = \{w \in \mathcal{Y} : \langle w, z \rangle \ge 0 \text{ for all } z \in \mathrm{TC}(q)\} \tag{B.3}$$

3. The *normal / polar cone* to $\mathcal{X}$ at $q \in \mathcal{X}$:

$$\mathrm{PC}(q) = \{w \in \mathcal{Y} : \langle w, z \rangle \le 0 \text{ for all } z \in \mathrm{TC}(q)\} \tag{B.4}$$

Following standard conventions in the field [95], convex functions will be allowed to take values in the extended real line $\mathbb{R} \cup \{\infty\}$, and we will denote the *effective domain* of a convex function $f \colon \mathcal{V} \to \mathbb{R} \cup \{\infty\}$ as

$$\mathrm{dom}\, f \coloneqq \{x \in \mathcal{V} : f(x) < \infty\}. \tag{B.5}$$

When there is no danger of confusion, we will identify a convex function $f \colon \mathcal{V} \to \mathbb{R}$ with its restriction on $\mathrm{dom}\, f$; in other words, we will treat $f$ interchangeably as a function on $\mathrm{dom}\, f$ with values in $\mathbb{R}$, or as a function on $\mathcal{V}$ with values in $\mathbb{R} \cup \{\infty\}$ (and finite on $\mathrm{dom}\, f$).

Throughout the sequel, we will assume that all functions under study are *proper*, that is, $\mathrm{dom}\, f \ne \emptyset$. Then, given a proper function $f \colon \mathcal{V} \to \mathbb{R} \cup \{\infty\}$, the *subdifferential* of $f$ at $x \in \mathrm{dom}\, f$ is defined as

$$\partial f(x) \coloneqq \{y \in \mathcal{Y} : f(x') \ge f(x) + \langle y, x' - x \rangle \text{ for all } x' \in \mathcal{V}\} \tag{B.6}$$

and we denote the *domain of subdifferentiability* of $f$ as

$$\mathrm{dom}\, \partial f = \{x \in \mathcal{V} : \partial f(x) \ne \emptyset\}. \tag{B.7}$$

With all this in hand, a *regularizer* on a closed convex subset $\mathcal{X}$ of $\mathcal{V}$ is a continuous function $h\colon \mathcal{X} \to \mathbb{R}$ which is *strongly convex* relative to $\|\cdot\|$, i.e., there exists some $K > 0$ such that

$$h(\lambda x + (1 - \lambda)x') \le th(x) + (1 - \lambda)h(x') - \frac{K}{2}\lambda(1 - \lambda)\|x' - x\|^2 \tag{B.8}$$

for all $x, x' \in \mathcal{X}$ and for all $\lambda \in [0, 1]$. By standard arguments [16, 96], this immediately implies that

$$h(x') \ge h(x) + h'(x; x' - x) + \frac{K}{2}\|x' - x\|^2 \quad \text{for all } x, x' \in \mathcal{X}, \tag{B.9}$$

where

$$h'(x; x' - x) = \lim_{\theta \to 0^+} [h(x + \theta(x' - x)) - h(x)]/\theta \tag{B.10}$$

denotes the one-sided directional derivative of $h$ at $x$ along the direction of $x' - x$.

As per Definition 1, we will say that $h$ is a *Bregman regularizer* if $\partial h$ admits a continuous selection $\nabla h(x) \in \partial h(x)$ for all $x \in \operatorname{dom} \partial h$. For such a regularizer, we recall the following definitions:

1. The *prox-domain* of $h$:
$$\mathcal{X}_h \coloneqq \operatorname{dom} \partial h \tag{B.11a}$$

2. The *convex conjugate* $h^*\colon \mathcal{Y} \to \mathbb{R}$ of $h$:
$$h^*(y) \coloneqq \max_{x \in \mathcal{X}}\{\langle y, x\rangle - h(x)\} \quad \text{for all } y \in \mathcal{Y}. \tag{B.11b}$$

3. The *mirror map* $Q\colon \mathcal{Y} \to \mathcal{X}$ induced by $h$:
$$Q(y) \coloneqq \arg\max_{x \in \mathcal{X}}\{\langle y, x\rangle - h(x)\} \quad \text{for all } y \in \mathcal{Y}. \tag{B.11c}$$

4. The *Fenchel coupling* $F\colon \mathcal{X} \times \mathcal{Y} \to \mathbb{R}$ defined by $h$:
$$F(q, y) = h(q) + h^*(y) - \langle y, q\rangle \quad \text{for all } q \in \mathcal{X}, y \in \mathcal{Y}. \tag{B.11d}$$

5. The *Bregman divergence* $D\colon \mathcal{X} \times \mathcal{X}_h \to \mathbb{R}$ associated to $h$:
$$D(q, x) = h(q) - h(x) - \langle \nabla h(x), q - x\rangle \quad \text{for all } q \in \mathcal{X}, x \in \mathcal{X}. \tag{B.11e}$$

6. The *prox-mapping* $P\colon \mathcal{X}_h \times \mathcal{Y} \to \mathcal{X}_h$ induced by $h$:
$$P_x(w) = \arg\min_{u \in \mathcal{X}}\{\langle w, x - u\rangle + D(u, x)\} \quad \text{for all } q \in \mathcal{X}, y \in \mathcal{Y}. \tag{B.11f}$$

*Remark* B.1. Of the items above, only the last two require $h$ to be a Bregman regularizer; the rest apply to any regularizer. ✦

The proposition below provides some basic properties linking the above:

**Proposition B.1.** *Let $h$ be a $K$-strongly convex regularizer on $\mathcal{X}$. Then:*

(*a*) *$Q$ is single-valued on $\mathcal{Y}$.*

(*b*) *For all $x \in \mathcal{X}_h$ and all $y \in \mathcal{Y}$, we have*

$$x = Q(y) \quad \text{if and only if} \quad y \in \partial h(x). \tag{B.12}$$

(*c*) *The image $\operatorname{im} Q$ of $Q$ is equal to the prox-domain of $h$, and we have*

$$\operatorname{ri} \mathcal{X} \subseteq \operatorname{im} Q = \mathcal{X}_h \subseteq \mathcal{X}. \tag{B.13}$$

(*d*) *The convex conjugate $h^*\colon \mathcal{Y} \to \mathbb{R}$ of $h$ is differentiable and satisfies*

$$Q(y) = \nabla h^*(y) \quad \text{for all } y \in \mathcal{Y}. \tag{B.14}$$

(e) *Q is $(1/K)$-Lipschitz continuous, that is,*

$$\|Q(y') - Q(y)\| \le (1/K)\|y' - y\|_* \quad \text{for all } y, y' \in \mathcal{Y}. \tag{B.15}$$

(f) *Fix some $y \in \mathcal{Y}$ and let $x = Q(y)$. Then, for all $x' \in \mathcal{X}$ we have:*

$$h'(x; x' - x) \ge \langle y, x' - x \rangle. \tag{B.16}$$

(g) *Fix some $y \in \mathcal{Y}$, and let $x = Q(y)$. Then $Q(y + w) = x$ for all $w \in \mathrm{PC}(x)$.*

*Proof.* For the most part, these properties are well known in the literature (except possibly the last one), so we only provide a pointer or a short sketch for most of them.

(a) This follows from the fact that $h$ is strongly convex, so the arg max in (B.11c) is attained and is unique for all $y \in \mathcal{Y}$.

(b) By Fermat's rule [95, Chap. 26], we readily see that $x$ solves (B.11c) if and only if $y - \partial h(x) \ni 0$, that is, if and only if $y \in \partial h(x)$. Since this implies that $\partial h$, our claim follows.

(c) By (B.12), we readily get $\operatorname{im} Q = \mathcal{X}_h$. As for the second part of our claim, it follows from basic properties of the subdifferential, cf. Rockafellar [95, Chap. 26].

(d) This is simply Danskin's theorem, see e.g., Bertsekas [16, Proposition 5.4.8, Appendix B].

(e) This is a consequence of the fact that $h^*$ is $(1/K)$-Lipschitz smooth, cf. Rockafellar & Wets [96, Theorem 12.60(b)].

(f) Since $y \in \partial h(x)$ by (B.12), we readily get that

$$h(x + \theta(x' - x)) \ge h(x) + \theta\langle y, x' - x \rangle \quad \text{for all } \theta \in [0, 1]. \tag{B.17}$$

Hence, by rearranging and taking the limit $\theta \to 0^+$, we conclude that

$$h'(x; x' - x) = \lim_{\theta \to 0^+} \frac{h(x + \theta(x' - x)) - h(x)}{\theta} \ge \langle y, x' - x \rangle \tag{B.18}$$

as claimed.[2]

(g) By (B.12) it suffices to show that $y + w \in \partial h(x)$ for all $w \in \mathrm{PC}(x)$. However, if $w \in \mathrm{PC}(x)$, we also have $\langle w, x' - x \rangle \le 0$ for all $x' \in \mathcal{X}$, and hence, with $y \in \partial h(x)$, we readily get

$$\begin{aligned} h(x') &\ge h(x) + \langle y, x' - x \rangle \\ &\ge h(x) + \langle y + w, x' - x \rangle \quad \text{for all } x' \in \mathcal{X}. \end{aligned} \tag{B.19}$$

This shows that $y + w \in \partial h(x)$ and completes our proof. ∎

Following [68, 74], we also define the *Fenchel coupling* associated to $h$ as

$$F(q, y) = h(q) + h^*(y) - \langle y, q \rangle \quad \text{for all } q \in \mathcal{X}, y \in \mathcal{Y}. \tag{B.20}$$

The next proposition shows that the Fenchel coupling can be seen as a "primal-dual" measure of divergence between $q \in \mathcal{X}$ and $y \in \mathcal{Y}$:

**Proposition B.2.** *Let $h$ be a Bregman regularizer on $\mathcal{X}$. Then, for all $q \in \mathcal{X}$ and all $y \in \mathcal{Y}$, $x = Q(y)$, we have:*

(a) $F(q, y) \ge 0$ *with equality if and only if $q = x$.* (B.21a)

(b) $F(q, y) \ge D(q, x)$ *with equality whenever $x \in \operatorname{ri} \mathcal{X}$.* (B.21b)

(c) $F(q, y) \ge \frac{1}{2} K \|Q(y) - q\|^2.$ (B.21c)

*Proof.* These properties are all fairly well known in the literature; we only provide a quick proof for completeness.

---

[2]The existence of the limit is guaranteed by elementary convex analysis arguments, cf. Bertsekas [16, App. B].

(a) By the Fenchel–Young inequality, we have $h(q) + h^*(y) \geq \langle y, q \rangle$ for all $q \in \mathcal{X}$, $y \in \mathcal{Y}$, with equality if and only if $y \in \partial h(q)$. Our claim then follows from (B.12).

(b) By definition, we have

$$
\begin{aligned}
F(q, y) &= h(q) + h^*(y) - \langle y, q \rangle \\
&= h(q) + \langle y, x \rangle - h(x) - \langle y, q \rangle = h(q) - h(x) - \langle y, q - x \rangle .
\end{aligned}
\tag{B.22}
$$

Since $x \in \operatorname{ri} \mathcal{X}$ and $y \in \partial h(x)$, we readily get $\langle y, q - x \rangle = h'(x; q - x) = \langle \nabla h(x), q - x \rangle$, showing that $F(q, y) = D(q, x)$ in this case. Otherwise, by (B.16), we get $\langle y, q - x \rangle \geq h'(x; q - x) = \langle \nabla h(x), q - x \rangle$, and the desired inequality follows.

(c) Arguing as above, we have

$$
\begin{aligned}
F(q, y) &= h(q) + h^*(y) - \langle y, q \rangle \\
&= h(q) + \langle y, x \rangle - h(x) - \langle y, q \rangle && \triangleright \text{ since } y \in \partial h(x) \\
&\geq h(q) - h(x) - h'(x; q - x) && \triangleright \text{ by Proposition B.1} \\
&\geq \tfrac{1}{2} K \|x - q\|^2 && \triangleright \text{ by (B.8)}
\end{aligned}
$$

so our proof is complete. ∎

Our last result at this point is a useful differentiation formula for the Fenchel coupling:

**Lemma B.1.** *For all $q \in \mathcal{X}$ and all $y \in \mathcal{Y}$, we have:*

$$
\nabla_y F(q, y) = Q(y) - q .
\tag{B.23}
$$

*Proof.* The proof follows immediately from Danskin's theorem, cf. Eq. (B.14) of Proposition B.1. ∎

**B.2. Update lemmas.** Moving forward, we note that the basic update step of (LMD) can be written as

$$
y^+ = y + w \quad \text{and} \quad x^+ = Q(y^+)
\tag{B.24}
$$

for some $y, w \in \mathcal{Y}$.

We begin with the equivalence of the primal and dual updates—of (SMD) and (LMD) respectively—when $h$ is steep, that is, $\mathcal{X}_h = \operatorname{ri} \mathcal{X}$. The key element of this equivalence is the general proposition below.

**Proposition B.3.** *Fix some $y \in \mathcal{Y}$, $x = Q(y)$, and $w \in \mathcal{Y}$. Suppose further that $y - \nabla h(x)$ annihilates all tangent vectors to $\mathcal{X}$ at $x$, i.e.,*

$$
\langle y - \nabla h(x), x' - x \rangle = 0 \quad \text{for all } x' \in \mathcal{X}.
\tag{B.25}
$$

*Then:*

$$
P_x(w) = Q(y + w) .
\tag{B.26}
$$

**Corollary B.1.** *Proposition B.3 holds whenever $x = Q(y) \in \operatorname{ri} \mathcal{X}$; in particular, if $h$ is steep, (B.26) holds for all $y \in \mathcal{Y}$.*

**Corollary B.2.** *Suppose that (SMD) and (LMD) are initialized respectively at $x_1 \in \mathcal{X}_h$ and $y_1 \in \mathcal{Y}$, with $x_1 = Q(y_1)$. If $h$ is steep and the algorithms are run with the same input sequence $g_t$, $t = 1, 2, \ldots$, then they generate the same sequence of iterates.*

*Proof of Proposition B.3.* By the definitions (B.11f) and (B.11c) of $P$ and $Q$ respectively, we have:

$$
\begin{aligned}
P_x(w) &= \operatorname*{arg\,min}_{x' \in \mathcal{X}} \{ \langle w, x - x' \rangle + D(x', x) \} \\
&= \operatorname*{arg\,min}_{x' \in \mathcal{X}} \{ \langle w, x - x' \rangle + h(x') - h(x) - \langle \nabla h(x), x' - x \rangle \} \\
&= \operatorname*{arg\,min}_{x' \in \mathcal{X}} \{ \langle w, x - x' \rangle + h(x') - h(x) - \langle y, x' - x \rangle \} \\
&= \operatorname*{arg\,max}_{x' \in \mathcal{X}} \{ \langle y + w, x' \rangle - h(x') \} = Q(y + w) .
\end{aligned}
$$
∎

We now proceed to state a series of identities and estimates for the Fenchel coupling before and after an update of the form (B.24). These results are not new, cf. [42, 74, 75] and references therein; however, the assumptions used to derive them vary in the literature, so we provide detailed proofs for completeness.

The first is a primal-dual version of the so-called "three-point identity" for Bregman functions [25]:

**Lemma B.2.** *Fix some $q \in \mathcal{X}$, $y \in \mathcal{Y}$, and let $x = Q(y)$. Then, for all $y^+ \in \mathcal{Y}$, we have:*

$$F(q, y^+) = F(q, y) + F(x, y^+) + \langle y^+ - y, x - q \rangle. \tag{B.27}$$

*Proof.* By definition, we have:

$$F(q, y^+) = h(q) + h^*(y^+) - \langle y^+, q \rangle \tag{B.28a}$$
$$F(q, y) = h(q) + h^*(y) - \langle y, q \rangle \tag{B.28b}$$
$$F(x, y^+) = h(x) + h^*(y^+) - \langle y^+, x \rangle \tag{B.28c}$$

Thus, subtracting (B.28b) and (B.28c) from (B.28a), and rearranging, we get

$$F(q, y^+) = F(q, y) + F(x, y^+) - h(x) - h^*(y) + \langle y^+, x \rangle - \langle y^+ - y, q \rangle. \tag{B.29}$$

Our assertion then follows by recalling that $x = Q(y)$, so $h(x) + h^*(y) = \langle y, x \rangle$. ∎

The next result we present concerns the Fenchel coupling before and after a direct update step; similar results exist in the literature, but we again provide a proof for completeness.

**Lemma B.3.** *Fix some $q \in \mathcal{X}$ and $y, w \in \mathcal{Y}$. Then, letting $x = Q(y)$, $y^+ = y + w$, and $x^+ = Q(y^+)$ as per (B.24), we have:*

$$F(q, y^+) = F(q, y) + \langle w, x^+ - q \rangle - F(x^+, y) \tag{B.30a}$$

$$\leq F(q, y) + \langle w, x - q \rangle + \frac{1}{2K} \|w\|_*^2. \tag{B.30b}$$

*Proof.* By the three-point identity (B.27), we have

$$F(q, y) = F(q, y^+) + F(x^+, y) + \langle y - y^+, x^+ - q \rangle \tag{B.31}$$

so our first claim is immediate. For our second claim, rearranging terms and employing the Fenchel–Young inequality gives

$$
\begin{aligned}
F(q, y^+) &+ \langle w, x^+ - q \rangle - F(x^+, y) \\
&= F(q, y) + \langle w, x - q \rangle + \langle w, x^+ - x \rangle - F(q, y) \\
&\leq F(q, y) + \langle w, x - q \rangle + \frac{1}{2K} \|w\|_*^2 + \frac{K}{2} \|x - q\|^2 - F(q, y) \tag{B.32}
\end{aligned}
$$

so our claim follows from Proposition B.2. ∎

In particular, as a consequence of Lemma B.3 we obtain the Lipschitzness of the Fenchel coupling with respect to its second variable:

**Lemma B.4.** *Fix some $q \in \mathcal{X}$ and suppose that the diameter $\|\mathcal{X}\|$ of $\mathcal{X}$ is finite. Then, for all $y, y^+ \in \mathcal{Y}$,*

$$|F(q, y^+) - F(q, y)| \leq \|\mathcal{X}\| \|y^+ - y\|_*. \tag{B.33}$$

*Proof.* Applying the first claim of Lemma B.3 to both couples $(y, y^+)$ and $(y^+, y)$ and using that $F \geq 0$ (cf. Proposition B.2), we readily get

$$
\begin{aligned}
-\|\mathcal{X}\| \|y - y^+\|_* \leq -\|Q(y^+) - q\| \|y^+ - y\|_* &\leq -\langle y^+ - y, Q(y^+) - q \rangle \\
&\leq F(q, y) - F(q, y^+) \\
&\leq \langle y - y^+, Q(y) - q \rangle \leq \|Q(y) - q\| \|y - y^+\|_* \leq \|\mathcal{X}\| \|y - y^+\|_*, \tag{B.34}
\end{aligned}
$$

which proves the result. ∎

Finally, we present a refinement of Lemma B.3 that will be needed in the sequel to deal with heavy-tailed noise.

**Lemma B.5.** *Fix some $q \in \mathcal{X}$ and $y, w \in \mathcal{Y}$, and suppose that the diameter $\|\mathcal{X}\|$ of $\mathcal{X}$ is finite. Then, letting $x = Q(y), y^+ = y + w$, and $x^+ = Q(y^+)$, we have*

$$F(q, y^+) \leq F(q, y) + \langle w, x - q \rangle + \frac{\|\mathcal{X}\|^{2-r}}{K^{r-1}} \|w\|_*^r \tag{B.35}$$

*for every $r \in [1, 2]$.*

*Proof.* From the first claim of Lemma B.3 and the positiveness of the Fenchel coupling, we get

$$F(q, y^+) = F(q, y) + \langle w, x - q \rangle + \langle w, x^+ - x \rangle - F(q, y) \leq F(q, y) + \langle w, x - q \rangle + \|w\|_* \|x^+ - x\|. \tag{B.36}$$

To continue, note that

$$\|x^+ - x\| \|w\|_* = \|x^+ - x\|^{2-r} \|Q(y^+) - Q(y)\|^{r-1} \|w\|_* \leq \frac{\|\mathcal{X}\|^{2-r}}{K^{r-1}} \|w\|_*^r, \tag{B.37}$$

where the inequality follows by the boundedness of $\mathcal{X}$ and the fact that the mirror map $Q$ is $(1/K)$-Lipschitz (cf. Proposition B.1). This therefore concludes the proof. ∎

## C  A primer on Lévy processes

In this section, we review the necessary background on Lévy processes and stochastic analysis, with an emphasis on the tools required for the convergence analysis developed in the main text. The material presented here is largely standard, and we refer the interested reader to the monographs of Applebaum [5], Bertoin [15] and Sato [99] for comprehensive introductions to the subject. Throughout this section, we assume a filtered probability space $(\Omega, \mathcal{F}, (\mathcal{F}(t))_{t \geq 0}, \mathbb{P})$ satisfying the usual conditions[3].

**C.1. Lévy processes.**  A Lévy process $L = (L(t))_{t \geq 0}$ is a continuous-time stochastic process taking values in $\mathbb{R}^n$ and satisfying:

1. **Initial condition:** $L(0) = 0$ (a.s.)

2. **Independent increments:** For every $0 \leq t_1 < \cdots < t_k$, the increments $L(t_{i+1}) - L(t_i)$ are independent.

3. **Stationary increments:** For every $t, s > 0$, The distribution of $L(t + s) - L(t)$ is the same as $L(s)$.

4. **Stochastic continuity:** For every $\varepsilon > 0$ and $t \geq 0$, $\lim_{s \to 0^+} \mathbb{P}(\|L(t + s) - L(t)\| > \varepsilon) = 0$.

Under the standard assumptions made on the filtered space, one may always assume that $L$ admits càdlàg (i.e., right-continuous with left limits) paths.

Lévy processes encompass many important continuous-time stochastic model; among the most prominent examples are the following:

**Example C.1** (Brownian motion). A *Brownian motion* $W = (W(t))_{t \geq 0}$ is a Lévy process whose increments $W(t + s) - W(t)$ are distributed as $\mathcal{N}(0, sI_n)$. In particular, Brownian motions are the only Lévy processes that admit continuous sample paths, and they admit finite moments of any order.

**Example C.2** (Poisson process). A *Poisson process* $N = (N(t))_{t \geq 0}$ with intensity $\lambda > 0$ is a nondecreasing, integer-valued Lévy process whose increments $N(t + s) - N(t)$ are distributed as $\text{Poisson}(\lambda s)$. In particular, it also admits moments of all orders.

**Example C.3** (Compound Poisson process). Let $Z = (Z_i)_{i \in \mathbb{N}}$ be a sequence of independent and identically distributed random variables, and let $N$ be an independent Poisson process with intensity $\lambda > 0$. Then the stochastic process $X(t) = \sum_{i=1}^{N(t)} Z_i$ is called a *compound Poisson process*. In particular, $X$ is a Lévy process. Moreover, a compound Poisson process admits moments of order $p$ if and only if the common distribution of the $Z_i$ admits moments of order $p$.

---

[3]That is, (a) the probability space $(\Omega, \mathcal{F}, \mathbb{P})$ is complete, meaning that every subset of a $\mathbb{P}$-null set is measurable; and (b) the filtration $(\mathcal{F}(t))_{t \geq 0}$ is right-continuous, i.e., $\mathcal{F}(t) = \bigcap_{u > t} \mathcal{F}(u)$.

**Example C.4** (Stable processes). A rotationally invariant stable process with stability index $\alpha \in (0, 2]$ is a Lévy process $L^\alpha$ such that

$$\mathbb{E}\big[e^{iuL^\alpha(t)}\big] = e^{-\sigma^\alpha \|u\|^\alpha t} \quad \text{for all } u \in \mathbb{R}^n, t \geq 0, \tag{C.1}$$

where $\sigma > 0$ is a scale factor. When $\alpha = 2$, this definition recovers the Brownian motion. By contrast, when $\alpha < 2$, an $\alpha$-stable process admits finite moments only of order $p < \alpha$, making such processes particularly well suited for modeling infinite-variance phenomena. More general definitions of stable processes without rotational invariance are available in the literature, although their explicit characterization is substantially more involved. We refer the interested reader to the monograph of Samoradnitsky & Taqqu [98] for further details.

**C.2. Jump processes and Poisson random measures.** As illustrated by the previous examples, Lévy processes are typically discontinuous in nature, making it essential to characterize the behavior of their *jumps* (i.e., of their discontinuities) in order to understand their long-term behavior.

Let $\Delta L(t) = L(t) - L(t-)$ denote the jump size at time $t$. We define a (random) measure $N(dt, dz)$ on $\mathbb{R}^+ \times (\mathbb{R}^n - \{0\})$ by counting the jumps:

$$N([s, t], B) = \#\{s < r \leq t : \Delta L(r) \in B\} = \sum_{s < r \leq t} \mathbb{1}_B(\Delta L(r)) \tag{C.2}$$

for any Borel set $B \subset \mathbb{R}^n - \{0\}$. Thus, $N([s, t], B)$ counts the number of jumps of size in $B$ occurring in the time interval $[s, t]$. In particular, $(N(t, B))_{t \geq 0} := (N([0, t], B))_{t \geq 0}$ is a Poisson process with intensity $\nu(B)t$, where $\nu$ is the measure on $\mathbb{R}^n - \{0\}$ defined by

$$\nu(B) = \mathbb{E}[N(1, B)] = \mathbb{E}\left[\sum_{0 < t \leq 1} \mathbb{1}_B(\Delta L(t))\right]. \tag{C.3}$$

This measure is known as the *Lévy measure* associated with $L$, and it also satisfies the integrability condition

$$\int_{\mathbb{R}^n - \{0\}} (\|z\|^2 \wedge 1)\nu(dz) < \infty. \tag{C.4}$$

An important consequence of this condition is that $\nu$ is finite on any Borel set that is bounded away from 0, which implies that a Lévy process admits only finitely many large jumps on any finite time interval. In contrast, $\nu$ may be infinite at 0, allowing for infinitely many jumps of small size. Another useful relation links the moments of $L$ to those of $\nu$ [99, Theorem 25.3]:

$$\mathbb{E}[\|L(t)\|^p] < \infty \text{ for all } t \geq 0 \quad \Longleftrightarrow \quad \int_{\|z\| \geq c} \|z\|^p \nu(dz) < \infty \text{ for all } c > 0. \tag{C.5}$$

This result illustrates that the tail behavior of a Lévy process—namely, whether it is heavy-tailed or light-tailed—is entirely determined by the contribution of larges jumps, and not by the small jumps nor by the continuous component.

**Example C.5** (Brownian motion, cont'd). A Brownian motion $W$ has continuous sample paths and therefore admits no jumps; consequently, its Lévy measure is identically zero.

**Example C.6** (Poisson process, cont'd). All jumps of a Poisson process with intensity $\lambda$ are of size 1, and the expected number of jumps occurring over the unit time interval $[0, 1]$ is $\lambda$. Hence, its Lévy measure is given by $\nu = \lambda \delta_1$, where $\delta_1$ denotes the Dirac measure at 1.

**Example C.7** (Compound Poisson process, cont'd). If $X$ is a compound Poisson process generated by a sequence of independent random variables with common distribution $\mu_Z$, then each jump of $X$ is conditionally distributed according to $\mu_Z$, and the expected number of jumps up to time 1 is $\lambda$. As a result, the associated Lévy measure is $\nu = \lambda \mu_Z$.

**Example C.8** (Stable processes, cont'd). If $L^\alpha$ is a rotationally invariant $\alpha$-stable process, then its Lévy measure takes the form $\nu(dz) = C/\|z\|^{1+\alpha}$ for some scaling constant $C > 0$. In contrast to compound Poisson processes, this Lévy measure explodes at 0, implying that stable processes exhibit infinitely many jumps of small magnitude.

The random measure $N(dt, dz)$ is commonly referred to as a *Poisson random measure* on $\mathbb{R}^+ \times (\mathbb{R}^n - \{0\})$ with intensity $\nu(dz)dt$[4]. Associated to $N(dt, dz)$, we also introduce the *compensated Poisson random measure*

$$\tilde{N}(dt, dz) = N(dt, dz) - \nu(dz)dt. \tag{C.6}$$

---

[4]In this section we have introduced Poisson random measures only through the specific construction arising from Lévy processes. More generally, Poisson random measures can be defined independently of Lévy processes and in much greater generality. We refer to [5] and [43] for more comprehensive treatments of this topic.

In contrast to $N(dt, dz)$, the compensated measure enjoys the property that $(\tilde{N}(t, B))_{t \geq 0}$ is a martingale for every bounded Borel set $B \subset \mathbb{R}^n - \{0\}$. This property makes $\tilde{N}(dt, dz)$ particularly well suited as an integrator for stochastic integrals, as discussed in the next subsection.

**C.3. Stochastic integration for jump processes.** To define stochastic integration of a stochastic process $(h(t, z))_{t \geq 0, z \in B}$ with respect to the measures $N(dt, dz)$ and $\tilde{N}(dt, dz)$, it is standard to require that $h(\cdot, z)$ be *predictable*, meaning that $h(t, z)$ is determined by $\mathcal{F}(t-)$[5]. In particular, this condition is satisfied whenever the mapping $t \mapsto h(t, z)$ is *left-continuous*. We then introduce the following class of stochastic processes:

$$\mathcal{H}_\nu^p(T, B; \mathbb{R}^n) = \left\{ h \colon [0, T] \times B \times \Omega \to \mathbb{R}^n : h(\cdot, z) \text{ is predictable for all } z \in B \right.$$
$$\left. \text{and } \mathbb{E}\left[ \int_0^\top \int_B \|h(s, z)\|^p \nu(dz) ds \right] < \infty \right\} \tag{C.7}$$

for $p \in [1, 2]$, $T \geq 0$, and any Borel set $B \subset \mathbb{R}^n - \{0\}$.

1. **Integration with respect to $N(dt, dz)$:** For integration against the raw measure $N(dt, dz)$, the integral is a sum over the jumps of the process. Specifically, if $h(\cdot, z)$ is a predictable process, then

$$\int_0^t \int_B h(s, z) N(ds, dz) = \sum_{0 \leq s \leq t} h(s, \Delta L(s)) \mathbb{1}_B(\Delta L(s)) \quad \text{for every } t \leq T. \tag{C.8}$$

2. **Integration with respect to $\tilde{N}(dt, dz)$:** There are two standard approaches to defining stochastic integrals with respect to the compensated measure $\tilde{N}(dt, dz)$. First, if $h \in \mathcal{H}_\nu^1(T, B; \mathbb{R}^n)$, one can leverage the integration with respect to $N(dt, dz)$ to get

$$M(t) := \int_0^t \int_B h(s, z) \tilde{N}(ds, dz) = \int_0^t \int_B h(s, z) N(ds, dz) - \int_0^t \int_B h(s, z) \nu(dz) ds \quad \text{for every } t \leq T. \tag{C.9}$$

This construction results to the process $M(t)$ being a *càdlàg martingale* [40, Chapter II, p.62]. Alternatively, if $h \in \mathcal{H}_\nu^2(T, B; \mathbb{R}^n)$, then the stochastic integral $M(t)$ can be defined as the $L^2$-limit of simple functions using the isometry

$$\mathbb{E}\left[ \left\| \int_0^t \int_B h(s, z) \tilde{N}(ds, dz) \right\|^2 \right] = \mathbb{E}\left[ \int_0^t \int_B \|h(s, z)\|^2 \nu(dz) ds \right]. \tag{C.10}$$

In this case, $M(t)$ is a *càdlàg square-integrable martingale* [5, Theorem 4.2.3].

Another class of stochastic processes that will be useful in the remainder of the paper is

$$\mathcal{H}^2(T; \mathbb{R}^n) = \left\{ g \colon [0, T] \times \Omega \to \mathbb{R}^n : g \text{ is predictable and } \mathbb{E}\left[ \int_0^t \|g(s)\|^2 ds \right] < \infty \right\}, \quad T \geq 0. \tag{C.11}$$

In particular, any $g \in \mathcal{H}^2(T; \mathbb{R}^n)$ can be integrated against a $n$-dimensional Brownian motion $W$ as the $L^2$-limit of simple functions, using the standard Itô isometry formula

$$\mathbb{E}\left[ \left\| \int_0^t g(s)^\top dW(s) \right\|^2 \right] = \mathbb{E}\left[ \int_0^t \|g(s)\|^2 ds \right]. \tag{C.12}$$

As in the case of integration with respect to $\tilde{N}(dt, dz)$, the resulting stochastic integral is a *continuous square-integrable martingale*.

---

[5] Rigorously, a stochastic process $(h(t))_{t \geq 0}$ is said to be *predictable* if it is measurable with respect to the smallest $\sigma$-field generated by all adapted, left-continuous stochastic processes $(\psi(t))_{t \geq 0}$

**C.4. The Lévy-Itô decomposition.** Any Lévy process $L$ can be uniquely represented as a sum of a continuous part and independent jump components [5, Theorem 2.4.16]: for every constant $c > 0$,

$$L(t) = \underbrace{b_c t + \Sigma W(t)}_{\text{Continuous part}} + \underbrace{\int_0^t \int_{\|z\|<c} z\tilde{N}(ds, dz)}_{\text{Compensated small jumps}} + \underbrace{\int_0^t \int_{\|z\|\geq c} zN(ds, dz)}_{\text{Large jumps}}, \tag{C.13}$$

where $b_c \in \mathbb{R}^n$ is a deterministic drift, $W$ is a $d$-dimensional Brownian motion independent of $N$, and $\Sigma \in \mathbb{R}^{n\times d}$ is a deterministic diffusion matrix. The constant $c$ acts as a cutoff separating between bounded and unbounded jumps, and may be chosen arbitrarily. For example, if the jumps of $L$ are uniformly bounded by some $M > 0$, one may take $c = M + 1$, in which case the large-jump component vanishes.

It is important to note that the small jumps must be integrated against the compensated measure $\tilde{N}(dt, dz)$, rather than the raw measure $N(dt, dz)$, to ensure that the stochastic integral is well defined even when $\int_{\|z\|<c} \|z\|\nu(dz) = \infty$.

In particular, if $L$ is an integrable centered Lévy process, then it admits the decomposition

$$L(t) = -\left(\int_{\|z\|\geq c} \|z\|\nu(dz)\right)t + \Sigma W(t) + \int_0^t \int_{\|z\|<c} z\tilde{N}(ds, dz) + \int_0^t \int_{\|z\|\geq c} zN(ds, dz) \tag{C.14}$$

$$= \Sigma W(t) + \int_0^t \int_{\|z\|<c} z\tilde{N}(ds, dz) + \int_0^t \int_{\|z\|\geq c} z\tilde{N}(ds, dz) \tag{C.15}$$

$$= M(t) + S(t) + U(t) \tag{C.16}$$

where $M(t)$ is a continuous square-integrable martingale, $S(t)$ is a purely discontinuous square-integrable martingale with bounded jumps, and $U(t)$ is a purely discontinuous martingale with possibly unbounded jumps (see Appendix C.3).

With this decomposition in mind, the intensity of each component of $L$ can then be quantified as follows:

(*a*) *Diffusion intensity:*

$$\sigma_0^2 = \mathbb{E}\big[\|M(t)\|^2\big] = \|\Sigma\|_F^2 = \text{tr}[\Sigma^\top \Sigma]. \tag{C.17}$$

(*b*) *Short jump intensity:*

$$\sigma_{\text{short}}^2 = \mathbb{E}\big[\|S(t)\|^2\big] = \int_{\|z\|<c} \|z\|^2 \nu(dz). \tag{C.18}$$

(*c*) *Long jump intensity:*

$$\sigma_{\text{long}}^p = \int_{\|z\|\geq c} \|z\|^p \nu(dz) \geq C_p \,\mathbb{E}[\|U(t)\|^p], \quad p \geq 0. \tag{C.19}$$

Owing to the relation between $L$ and its Lévy measure, the long jump intensity $\sigma_{\text{long}}^p$ is finite if and only if $L$ admits finite moments of order $p$. Moreover, in general $\sigma_{\text{long}}^p$ does not coincide with $\mathbb{E}[\|U(t)\|^p]$ unless $p = 2$. We refer the interested reader to [84] for a detailed discussion of this discrepancy and for estimates on the constant $C_p$ relating the two quantities.

**C.5. Asymptotic behavior of discontinuous martingales.** In this subsection, we present two classical results from stochastic analysis—accordingly the *law of large numbers* and the *law of iterated logarithm*—adapted to the setting of discontinuous martingales. Unlike the continuous case, these results are less widely known for processes with jumps.

We begin with a strong law of large numbers for stochastic integrals with respect to compensated Poisson random measures. Throughout, we let $N(dt, dz)$ denote the random Poisson measure associated with a Lévy process $L$ having Lévy measure $\nu$ (see Appendix C.2).

**Lemma C.1** (Strong law of large numbers). *Fix some index $p \in [1, 2]$ and let $M(t) = \int_0^t \int_B H(s, z)\tilde{N}(ds, dz)$ with $H \in \mathcal{H}_\nu^p(T, B; \mathbb{R}^n)$ for all $T \geq 0$. If there is a predictable increasing process $A$ such that $\lim A(t) = \infty$ (a.s.) and*

$$\int_0^\infty \int_B \frac{\|H(s, z)\|^2 \wedge \|H(s, z)\|^p}{[1 + A(s)]^p}\nu(dz)ds < \infty \quad (a.s.), \tag{C.20}$$

*then $M(t)/A(t) \to 0$ (a.s.).*

*Proof.* This is a consequence of [54, Lemma 3]. Indeed, $M$ is a centered purely discontinuous martingale since $H \in \mathcal{H}_\nu^p(T, B; \mathbb{R}^n)$ and, following the notations from [54], we can introduce the auxiliary process $W_P$ given by

$$W_P(t) = \sum_{0 < s \leq t} \|\Delta M(s)\|^2 \wedge \|\Delta M(s)\|^p = \int_0^t \int_B \|H(s, z)\|^2 \wedge \|H(s, z)\|^p N(ds, dz). \tag{C.21}$$

In particular, its integrand satisfies

$$\mathbb{E}\left[\int_0^t \int_B \|H(s, z)\|^2 \wedge \|H(s, z)\|^p \nu(dz)ds\right] \leq \mathbb{E}\left[\int_0^t \int_B \|H(s, z)\|^p \nu(dz)ds\right] < \infty \tag{C.22}$$

since $H \in \mathcal{H}_\nu^p(T, B; \mathbb{R}^n)$, and thus

$$\overline{W_P}(t) = \int_0^t \int_B \|H(s, z)\|^2 \wedge \|H(s, z)\|^p \nu(dz)ds \tag{C.23}$$

is the compensator of $W_P$, i.e., the unique predictable càdlàg process starting from 0 with finite variation over bounded intervals and such that $W_P - \overline{W_P}$ is a (local) martingale. According to [54, Lemma 3], the strong law of large number $M(t)/A(t) \to 0$ therefore holds true whenever

$$\infty > \int_0^\infty \frac{d\overline{W_P}(s)}{[1 + A(s)]^p} = \int_0^\infty \int_B \frac{\|H(s, z)\|^2 \wedge \|H(s, z)\|^p}{[1 + A(s)]^p} \nu(dz)ds, \tag{C.24}$$

which is exactly the condition stated in the lemma. ∎

To state the law of iterated logarithm for discontinuous martingales, we first need to introduce another essential object from stochastic analysis: the *predictable quadratic variation*. For any càdlàg square-integrable local martingale $M$, its *predictable quadratic variation* (also sometimes known as *Meyer's angle braket*) is the unique increasing predictable process $\langle M \rangle$ such that $M^2 - \langle M \rangle$ is a local martingale. In particular, when $M$ arises as a stochastic integral with respect to either a Brownian motion or a compensated Poisson measure, the following identities hold:

$$\left\langle \int_0^t g(s)^\top dW(s) \right\rangle = \int_0^t \|g(s)\|^2 ds, \quad g \in \mathcal{H}^2(T; \mathbb{R}^n); \tag{C.25}$$

$$\left\langle \int_0^t \int_B h(s, z) \tilde{N}(ds, dz) \right\rangle = \int_0^t \int_B \|h(s, z)\|^2 \nu(dz)ds, \quad h \in \mathcal{H}_\nu^2(T, B; \mathbb{R}^n). \tag{C.26}$$

We can now formulate the law of iterated logarithm, which characterizes the asymptotic fluctuations of a martingale in terms of its predictable quadratic variation:

**Lemma C.2** (Law of iterated logarithm, [55, Theorem 3]). *Let $M$ be a square-integrable local martingale and assume that there exists a constant $C \geq 0$ such that $|\Delta M(t)| \leq C$ (a.s.) for every $t \geq 0$. Then, the trajectories of $M$ satisfy*

$$\limsup_{t \to \infty} \frac{|M(t)|}{\sqrt{2\langle M \rangle(t) \log\log\langle M \rangle(t)}} = 1 \quad (a.s.) \tag{C.27}$$

*on the event $\{\lim_{t \to \infty} \langle M \rangle(t) = \infty\}$.*

**C.6. An Itô's formula for smooth convex function.** Analogous to the chain rule in classical differential calculus, Itô's formula allows one to describe the evolution of a composition $\psi \circ Y$, where $Y$ is some $n$-dimensional stochastic process and $\psi : \mathbb{R}^n \to \mathbb{R}$ is a deterministic function. Because of the way stochasticity is handled in integration with respect to martingales, Itô's formula typically includes an additional second-order term—commonly referred to as *Itô's correction*—which requires $\psi$ to be at least twice continuously differentiable for the formula to hold.

However, in many applications, such as the analysis of stochastic mirror flows, the function $\psi$ of interest does not satisfy this regularity condition. The goal of this subsection is therefore to introduce a weak version of Itô's formula that apply to a more general class of functions: those that are only convex and $L$-smooth.

Recall that a function $\psi : \mathbb{R}^n \to \mathbb{R}$ is said to be *L-smooth* if it satisfies

$$\|\nabla\psi(y) - \nabla\psi(y')\|_* \leq L\|y - y'\|_* \quad \text{for all } y, y' \in \mathbb{R}^n. \tag{C.28}$$

In particular, $L$-smooth functions that are also convex enjoy the following property:

**Lemma C.3.** *If $\psi$ is an L-smooth convex function, then $\psi$ is almost everywhere twice differentiable with Hessian $\nabla^2\psi$ and*

$$0 \leq \nabla^2\psi \leq LI \quad \text{Lebesgue-a.e.} \tag{C.29}$$

*Proof.* The proof of this result is standard, see e.g., [70, Lemma C.3]. ∎

A weak Itô's formula for smooth convex functions was previously established by Mertikopoulos & Staudigl [70] for Itô diffusions, that is, for processes of the form $Y(t) = Y(0) + \int_0^t f(s)ds + \int_0^t g(s)dW(s)$ where $W$ is a standard Brownian motion. In the following theorem, we extend their result to the broader class of *Lévy-type processes*, which are characterized as follows:

Let $L$ be a $k$-dimensional Lévy process with Lévy measure $\nu$, and let $N(dt, dz)$ be the Poisson random measure controlling its jumps. Then, a *Lévy-type process $Y$* is a $n$-dimensional càdlàg stochastic process of the form

$$Y(t) = Y(0) + \underbrace{\int_0^t f(s)\,ds + \int_0^t g(s)\,dW(s)}_{\text{Continuous part } Y^c(t)} + \underbrace{\int_0^t \int_{\|z\|_* < c} H(s,z)\tilde{N}(ds, dz) + \int_0^t \int_{\|z\|_* \geq c} G(s,z)N(ds, dz)}_{\text{Discontinuous part } Y^d(t)} \tag{C.30}$$

for some constant $c > 0$, where $Y(0) < \infty$ (a.s.), $W$ is a standard $d$-dimensional Brownian motion independent of $N$, and, for every $T \geq 0$,

1. $f\colon \mathbb{R}^+ \times \Omega \to \mathbb{R}^n$ is such that $|f_i|^{1/2} \in \mathcal{H}^2(T; \mathbb{R})$;
2. $g\colon \mathbb{R}^+ \times \Omega \to \mathbb{R}^{n \times d}$ is such that $g_i \in \mathcal{H}^2(T; \mathbb{R}^d)$;
3. $H\colon \mathbb{R}^+ \times \{\|z\|_* < c\} \times \Omega \to \mathbb{R}^n$ is such that $H \in \mathcal{H}_\nu^2(T, \{\|z\|_* < c\}; \mathbb{R}^n)$;
4. $G\colon \mathbb{R}^+ \times \{\|z\|_* \geq c\} \times \Omega \to \mathbb{R}^n$ is predictable.

By virtue of the Lévy-Itô decomposition (cf. Appendix C.4), Lévy-type processes therefore provide a general and flexible framework to construct stochastic integrals with respect to a Lévy process.

This general representation of Lévy-type processes sets the stage for extending Itô's formula: we can now formulate a weak version of the formula that applies to convex $L$-smooth functions, encompassing both the continuous and discontinuous components of $Y$:

**Theorem C.1** (Weak Itô's formula for smooth convex functions). *Let $Y$ be a $n$-dimensional Lévy-type process. If $\psi\colon \mathbb{R}^n \to \mathbb{R}$ is an L-smooth convex function, then*

$$
\begin{aligned}
\psi(Y(t)) - \psi(Y(0)) \leq & \int_0^t \langle \nabla\psi(Y(s-)), dY^c(s)\rangle \\
& + \int_0^t \int_{\|z\|_* < c} [\psi(Y(s-) + H(s,z)) - \psi(Y(s-))]\tilde{N}(ds, dz) \\
& + \int_0^t \int_{\|z\|_* \geq c} [\psi(Y(s-) + G(s,z)) - \psi(Y(s-))]N(ds, dz) \\
& + \frac{L}{2}\left(\int_0^t \|g(s)\|^2 ds + \int_0^t \int_{\|z\|_* < c} \|H(s,z)\|^2 \nu(dz)ds\right),
\end{aligned} \tag{C.31}
$$

*where $Y(s-)$ denotes the left limit of $Y$ at $s$ [6].*

We first prove this theorem under the assumption $G \equiv 0$, that is, when the jumps of $Y$ are of bounded magnitude. This condition is convenient because, in this case, the discontinuous component of $Y$ reduces to a square-integrable martingale, which significantly simplifies the setting.

---

[6]As discussed in Appendix C.3, stochastic integrals with respect to a Poisson random measure require the integrands to be *predictable*. Since $Y$ is right-continuous, its left limit $Y(t-)$ defines a left-continuous process which is predictable by definition, and thus all stochastic integrals appearing in Theorem C.1 are well-defined in this setting.

*Proof of Theorem C.1 (case $G \equiv 0$).* Let $\rho \colon \mathbb{R}^n \to \mathbb{R}$ be the unit mollifier given by

$$\rho(u) = \begin{cases} c \exp\left(-\dfrac{1}{1 - \|u\|^2}\right) & \text{if } \|u\| \le 1, \\ 0 & \text{otherwise;} \end{cases} \tag{C.32}$$

where $c$ is a positive normalizing constant such that $\int \rho(u)du = 1$. For every $\varepsilon > 0$, we also define

$$\rho_\varepsilon(w) = \varepsilon^{-n}\rho(w/\varepsilon) \tag{C.33}$$

and

$$\psi_\varepsilon(y) = (\psi * \rho_\varepsilon)(y) = \int \psi(y - w)\rho_\varepsilon(w)dw. \tag{C.34}$$

From classical smoothing arguments, we note that $\psi_\varepsilon$ is twice continuously differentiable, and so we can apply the classical Itô's formula for Lévy-type stochastic integrals (cf. [5, Theorem 4.4.7]) to get

$$\psi_\varepsilon(Y(t)) - \psi_\varepsilon(Y(0)) = \int_0^t \langle \nabla\psi_\varepsilon(Y), dY^c \rangle \tag{C.35a}$$

$$+ \int_0^t \int_{\|z\|_* < c} [\psi_\varepsilon(Y + H) - \psi_\varepsilon(Y)]\tilde{N}(ds, dz) \tag{C.35b}$$

$$+ \frac{1}{2}\int_0^t \operatorname{tr}\big[\nabla^2\psi_\varepsilon(Y)gg^\top\big]ds \tag{C.35c}$$

$$+ \int_0^t \int_{\|z\|_* < c} [\psi_\varepsilon(Y + H) - \psi_\varepsilon(Y) - \langle \nabla\psi_\varepsilon(Y), H \rangle]\nu(dz)ds, \tag{C.35d}$$

where we have suppressed the dependence on time for notational convenience. To bound Eq. (C.35c), note that the definition of $\psi_\varepsilon$ and Lemma C.3 can be used to obtain

$$\operatorname{tr}\big[\nabla^2\psi_\varepsilon(y)zz^\top\big] = \int_{\mathbb{R}^n} \operatorname{tr}\big[\nabla^2\psi(w)zz^\top\big]\rho_\varepsilon(y - w)dw \le L\|z\|^2 \tag{C.36}$$

for every $y, z \in \mathbb{R}^n$, which leads to

$$\text{(C.35c)} \le \frac{L}{2}\int_0^t \|g(s)\|^2 ds. \tag{C.37}$$

Furthermore, by the classical descent lemma for $L$-smooth functions (see e.g., [79, Theorem 2.1.5]), we get

$$\psi(y + h) - \psi(y) - \langle \nabla\psi(y), h \rangle \le \frac{L}{2}\|h\|^2 \tag{C.38}$$

for all $y, h \in \mathbb{R}^n$, so that

$$\text{(C.35d)} \le \frac{L}{2}\int_0^t \int_{\|z\|_* < c} \|H(s, z)\|^2\nu(dz)ds, \tag{C.39}$$

where the right-hand side is almost-surely finite by assumption on $H$. We therefore obtain that

$$\psi_\varepsilon(Y(t)) - \psi_\varepsilon(Y(0)) \le \int_0^t \langle \nabla\psi_\varepsilon(Y), dY^c \rangle$$

$$+ \int_0^t \int_{\|z\|_* < c} [\psi_\varepsilon(Y + H) - \psi_\varepsilon(Y)]\tilde{N}(ds, dz)$$

$$+ \frac{L}{2}\left(\int_0^t \|g\|^2 ds + \int_0^t \int_{\|z\|_* < c} \|H\|^2\nu(dz)ds\right). \tag{C.40}$$

To prove the weak Itô's formula, it only remains to show that we can replace $\psi_\varepsilon$ by $\psi$ when taking the almost-sure limit as $\varepsilon \to 0^+$ on both sides of the inequality. For the sake of readability, let us denote $L(\psi_\varepsilon)$ the left-hand side of Eq. (C.40) and $R(\psi_\varepsilon)$ its right-hand side. We begin by establishing some groundwork that will become quite useful in the

remaining of the proof. First, note that for every $y, y' \in \mathbb{R}^n$ and every differentiable function $\phi \colon \mathbb{R}^n \to \mathbb{R}$, we can write $\phi(y) - \phi(y') = \int_0^1 \langle \nabla \phi(y_r), y - y' \rangle dr$ where $y_r = y + r(y' - y)$. Applying this equality to $\psi$ and $\psi_\varepsilon$, we get

$$|\psi(y) - \psi(y') - \psi_\varepsilon(y) + \psi_\varepsilon(y')| = \left| \int_0^1 \langle \nabla \psi(y_r) - \nabla \psi_\varepsilon(y_r), y - y' \rangle dr \right| \le \|y - y'\| \int_0^1 \|\nabla \psi(y_r) - \nabla \psi_\varepsilon(y_r)\| dr. \quad \text{(C.41)}$$

Now, by using the definition of $\psi_\varepsilon$ and the $L$-smoothness of $\psi$, we can further derive

$$\|\nabla \psi(y_r) - \nabla \psi_\varepsilon(y_r)\| \le \int_{\mathbb{R}^n} \|\nabla \psi(y_r) - \nabla \psi(y_r - w)\| \rho_\varepsilon(w) dw \le L \int_{\mathbb{R}^n} \|w\| \varepsilon^{-n} \rho(w/\varepsilon) dw$$

$$= L\varepsilon \int_{\|u\| < 1} \|u\| \rho(u) du$$

$$\le L\varepsilon, \quad \text{(C.42)}$$

where the second-to-last inequality comes from the substitution $u = w/\varepsilon$ and the fact that $\rho(u)$ is zero outside of $\|u\| < 1$. In particular, it means that $\nabla \psi_\varepsilon$ converges uniformly to $\nabla \psi$ on all $\mathbb{R}^n$ and not only on compact subsets. Coming back to Eq. (C.41), the previous estimate therefore yields

$$|\psi(y) - \psi(y') - \psi_\varepsilon(y) + \psi_\varepsilon(y')| \le L\varepsilon \|y - y'\| \quad \text{(C.43)}$$

for every $y, y' \in \mathbb{R}^n$. Applying this inequality to the left-hand side $L$, we then readily get

$$|L(\psi) - L(\psi_\varepsilon)| \le L\varepsilon \|Y(t) - Y(0)\|, \quad \text{(C.44)}$$

which implies that $L(\psi_\varepsilon) \to L(\psi)$ almost-surely as $\varepsilon \to 0^+$ since $Y(t)$ and $Y(0)$ are both finite by assumption. Now, to derive the limit of $R(\psi_\varepsilon)$, we use the $L^1$-norm and expend $dY^c$ to get

$$\mathbb{E}[|R(\psi) - R(\psi_\varepsilon)|] \le \int_0^t \mathbb{E}[|\langle \nabla \psi(Y) - \nabla \psi_\varepsilon(Y), f \rangle|] ds + \mathbb{E}\left[ \left| \int_0^t \langle \nabla \psi(Y) - \nabla \psi_\varepsilon(Y), g dW \rangle \right| \right]$$

$$+ \mathbb{E}\left[ \left| \int_0^t \int_{\|z\|_* < c} [\psi(Y + H) - \psi(Y) - \psi_\varepsilon(Y + H) + \psi_\varepsilon(Y)] \tilde{N}(ds, dz) \right| \right]$$

$$\le \int_0^t \mathbb{E}[\|\nabla \psi(Y) - \nabla \psi_\varepsilon(Y)\| \|f\|] ds + \left[ \int_0^t \mathbb{E}\left[ \|\nabla \psi(Y) - \nabla \psi_\varepsilon(Y)\|^2 \|g\|^2 \right] ds \right]^{1/2}$$

$$+ \left[ \int_0^t \int_{\|z\|_* < c} \mathbb{E}\left[ |\psi(Y + H) - \psi(Y) - \psi_\varepsilon(Y + H) + \psi_\varepsilon(Y)|^2 \right] \nu(dz) ds \right]^{1/2} \quad \text{(C.45)}$$

where the second inequality comes from Jensen's inequality and Itô's isometry formula for stochastic integrals. Thanks to the estimates Eqs. (C.41) and (C.42) that we have previously established, it follows that

$$\mathbb{E}[|R(\psi) - R(\psi_\varepsilon)|] \le L\varepsilon \left( \mathbb{E}\left[ \int_0^t \|f\| ds \right] + \left[ \int_0^t \mathbb{E}\left[ \|g\|^2 \right] ds \right]^{1/2} + \left[ \int_0^t \int_{\|z\|_* < c} \mathbb{E}\left[ |H|^2 \right] \nu(dz) ds \right]^{1/2} \right), \quad \text{(C.46)}$$

where each integral is finite by assumption on the coefficients. We therefore get $R(\psi_\varepsilon) \to R(\psi)$ in the $L^1$-norm as $\varepsilon \to 0^+$, and thus there exists a subsequence $\varepsilon(i) \to 0$ such that $R(\psi_{\varepsilon(i)}) \to R(\psi)$ (a.s.). Along this subsequence we also get $L(\psi_{\varepsilon(i)}) \to L(\psi)$ (a.s.), which proves that $L(\psi) \le R(\psi)$ (a.s.), hence finishing the proof. ∎

Having established Theorem C.1 for Lévy-type processes with bounded jumps, we now extend the result to the general case, where jumps of unbounded magnitude may occur. A key property of Lévy processes—and their associated Lévy measure—is that only finitely many large jumps can happen over any finite time interval. Exploiting this fact, we can apply the previously proven case $G \equiv 0$ on each subinterval between large jumps, and then account for the large jumps separately.

*Proof of Theorem C.1 (general case).* Let $(T_i)_{i \ge 1}$ denote the arrival times of the Poisson process $(N(t, \{\|z\|_* \ge c\}))_{0 \le t \le T}$ controlling the size of large jumps of $Y$ up to time $T$. Since the intensity measure $\nu$ is assumed to be a Lévy measure, it is in particular finite on $\{\|z\|_* \ge c\}$ and so there is only a finite number of jumps occurring in the interval $[0, T]$. Let $N$ denotes

this (random) number of jumps. We further define $T_0 = 0$ and $T_{N+1} = T$ for notational convenience. Fix $t \leq T$ and let $i$ be the random integer such that $t \in [T_i, T_{i+1}]$. At the time $t$, we can therefore write

$$Y(t) = Y(T_i) + \int_{T_i}^t f(s)ds + \int_{T_i}^t g(s)dW(s) + \int_{T_i}^t \int_{\|z\|_* < c} H(s, z)\tilde{N}(ds, dz), \tag{C.47}$$

which falls into the setup of Theorem C.1 with $G \equiv 0$ since there are no large jumps during this time interval, $T_i \leq T < \infty$ by construction, and thanks to the independence of the Poisson process $N(\cdot, \{\|z\|_* \geq c\})$ to $N(\cdot, \{\|z\|_* < c\})$ and $W$. It then follows from the proof of case $G \equiv 0$ that

$$\psi(Y(t)) \leq \psi(Y(T_i)) + \int_{T_i}^t \langle \nabla \psi(Y), dY^c \rangle + \int_{T_i}^t \int_{\|z\|_* < c} [\psi(Y + H) - \psi(Y)]\tilde{N}(ds, dz)$$
$$+ \frac{L}{2}\left(\int_{T_i}^t \|g\|^2 ds + \int_{T_i}^t \int_{\|z\|_* < c} \|H\|^2 \nu(dz)ds\right). \tag{C.48}$$

On the other hand, if we define the Lévy process $L(t) = \int_{\|z\|_* \geq c} z N(t, dz)$, we also have

$$\psi(Y(T_i)) = \psi(Y(T_i-)) + \Delta\psi(Y(T_i)) = \psi(Y(T_i-)) + [\psi(Y(T_i-) + G(T_i, \Delta L(T_i))) - \psi(Y(T_i-))]. \tag{C.49}$$

Since $t \mapsto \psi(Y(t-))$ is left-continuous, applying the case $G \equiv 0$ on each time interval $[T_0, T_1), \ldots, [T_{i-2}, T_{i-1})$ therefore yields

$$\psi(Y(t)) \leq \psi(Y(T_0)) + \int_{T_0}^t \langle \nabla \psi(Y), dY^c \rangle + \int_{T_0}^t \int_{\|z\|_* < c} [\psi(Y + H) - \psi(Y)]\tilde{N}(ds, dz) \tag{C.50}$$
$$+ \frac{L}{2}\left(\int_{T_0}^t \|g\|^2 ds + \int_{T_0}^t \int_{\|z\|_* < c} \|H\|^2 \nu(dz)ds\right)$$
$$+ \sum_{j=1}^i [\psi(Y(T_j-) + G(T_j, \Delta L(T_j))) - \psi(Y(T_j-))]$$
$$= \psi(Y(0)) + \int_0^t \langle \nabla \psi(Y), dY^c \rangle + \int_0^t \int_{\|z\|_* < c} [\psi(Y + H) - \psi(Y)]\tilde{N}(ds, dz)$$
$$+ \frac{L}{2}\left(\int_0^t \|g\|^2 ds + \int_0^t \int_{\|z\|_* < c} \|H\|^2 \nu(dz)ds\right) + \int_0^t \int_{\|z\|_* < c} [\psi(Y + H) - \psi(Y)]N(ds, dz) \tag{C.51}$$

by definition of the stochastic integration with respect to a Poisson random measure (cf. Appendix C). This is exactly the upper bound stated in Theorem C.1, thus concluding the proof. ∎

## D Results in continuous time: Omitted proofs from Section 3

In this section, we study the convergence properties of the *Lévy mirror flow* (LMF), defined as the system of stochastic differential equations given by

$$dY(t) = -\nabla f(X(t))\, dt + dL(t)$$
$$X(t) = Q(\eta(t)Y(t)) \tag{LMF}$$

where $L$ is a $\mathcal{Y}$-valued centered Lévy process with $\mathbb{E}[\|L(t)\|_*^p] < \infty$ for some $p \in (1, 2]$, and $\eta \colon [0, \infty) \to (0, \infty)$ is a nonincreasing differentiable learning rate function.

To simplify notations, throughout this section we identify the dual space $\mathcal{Y}$ with $\mathbb{R}^n$, which allows us to define Lévy processes, Brownian motions and Poisson random measures in the standard way presented in Appendix C. In particular, $L$ admits the Lévy-Itô decomposition

$$L(t) = -\left(\int_{\|z\|_* \geq c} z\nu(dz)\right)t + \Sigma W(t) + \int_0^t \int_{\|z\|_* < c} z\tilde{N}(ds, dz) + \int_0^t \int_{\|z\|_* \geq c} z N(ds, dz), \tag{D.1}$$

where $c \in (0, \infty)$ is an arbitrary cutoff constant (cf. Appendix C.4). Under this decomposition, the quantities introduced in the main text to quantify the noise intensity are given by:

(a) *Diffusion intensity:*

$$\sigma_0^2 = \|\Sigma\|_F^2 = \text{tr}[\Sigma^\top \Sigma]. \tag{D.2}$$

(b) *Short jump intensity:*

$$\sigma_{\text{short}}^2 = \int_{\|z\| < c} \|z\|^2 \nu(dz). \tag{D.3}$$

(c) *Long jump intensity:*

$$\sigma_{\text{long}}^p = \int_{\|z\| \geq c} \|z\|^p \nu(dz). \tag{D.4}$$

Moreover, to highlight that $\sigma_0^2$ and $\sigma_{\text{short}}^2$ correspond to the components of $L$ that admits moments of all orders, whereas $\sigma_{\text{long}}^p$ captures the intensity of the component that may exhibit infinite variance, we also set

$$\sigma_{\text{tame}}^2 = \sigma_0^2 + \sigma_{\text{short}}^2 \quad \text{and} \quad \sigma_{\text{heavy}}^p = \sigma_{\text{long}}^p \tag{D.5}$$

for the tame and heavy part of the noise respectively.

**D.1. Existence and uniqueness of solutions.** We begin this section by addressing an important aspect that was largely left implicit in the main text, namely the well-posedness of the system (LMF). Specifically, under which conditions (LMF) admits a unique global solution for every initial condition.

In order to maintain a sufficiently general framework—one under which the various assumptions considered in the main text can coexist—we study the existence and uniqueness of solutions to (LMF) under the following conditions on the objective function $f$:

**Assumption 1.** $\nabla f \circ Q$ is locally Lipschitz on $\mathcal{Y}$: for every $r > 0$, there exist a nonnegative constant $L_r \geq 0$ such that

$$\|(\nabla f \circ Q)(y) - (\nabla f \circ Q)(y')\|_* \leq L_r \|y - y'\|_* \quad \text{for all } y, y' \in \mathbb{B}_r(0). \tag{$\mathcal{H}_1$}$$

**Assumption 2.** $\nabla f \circ Q$ satisfies a global growth condition on $\mathcal{Y}$: there exists $M > 0$ such that

$$\|(\nabla f \circ Q)(y)\|_* \leq M(1 + \|y\|_*) \quad \text{for all } y \in \mathcal{Y}. \tag{$\mathcal{H}_2$}$$

Assumption ($\mathcal{H}_1$) guarantees local existence and uniqueness of solutions, while ($\mathcal{H}_2$) prevents finite-time explosion and thus ensures global well-posedness. Together, these conditions yield the following result:

**Proposition D.1** (Existence and uniqueness of solutions). *Under assumptions ($\mathcal{H}_1$) and ($\mathcal{H}_2$), the stochastic differential equation* (LMF) *admits a unique càdlàg strong solution $X$ staying in $\mathcal{X}$ at all times. Moreover, if $h$ is steep, then $X(t) \in \text{ri} \, \mathcal{X}$ (a.s.) whenever $X(0) \in \text{ri} \, \mathcal{X}$.*

*Proof.* Replacing $X(t)$ by $Q(\eta(t)Y(t))$, the system (LMF) can be written as the stochastic differential equation

$$dY(t) = -(\nabla f \circ Q)(\eta(t)Y(t))dt + dL(t) \tag{D.6}$$

evolving in $\mathcal{Y}$. Let $r > 0$ and $y, y' \in \mathbb{B}_r(0)$. For every $t \geq 0$, we have $\eta(t)y, \eta(t)y' \in \mathbb{B}_{\eta(t)r}(0) \subseteq \mathbb{B}_{\eta(0)r}(0)$ thanks to the fact that $\eta$ is a nonincreasing positive function. In particular, the local Lipschitzness ($\mathcal{H}_1$) of $\nabla f \circ Q$ yields

$$\|(\nabla f \circ Q)(\eta(t)y) - (\nabla f \circ Q)(\eta(t)y')\|_* \leq L_{\eta(0)r} \|\eta(t)(y - y')\|_* \leq \eta(0)L_{\eta(0)r} \|y - y'\|_* \tag{D.7}$$

for all $y, y' \in \mathbb{B}_0(r)$ and every $r > 0$. Moreover, we also get

$$\|(\nabla f \circ Q)(\eta(t)y)\|_* \leq M(1 + \|\eta(t)y\|_*) \leq M(1 + \eta(0)\|y\|_*) \leq \max(1, \eta(0))M(1 + \|y\|_*) \tag{D.8}$$

by the global growth condition ($\mathcal{H}_2$). Since the upper bounds of both Eqs. (D.7) and (D.8) are uniform in $t$, we can therefore use a standard existence theorem for solutions to SDEs driven by semimartingales (see e.g., [65, Theorem 1.3.4]) to obtain

that (LMF) admits a unique strong solution $Y$ having càdlàg paths.[7] Since $X(t) = Q(\eta(t)Y(t)) \in \mathcal{X}$ by definition of (LMF) and of the mirror map $Q$, it follows that the primal trajectories are also càdlàg and remain in $\mathcal{X}$ at any times. Furthermore, if $h$ is steep, then $\mathrm{im}(Q) = \mathrm{ri}\,\mathcal{X}$ (cf. Proposition B.1), which immediately yields the second claim. ∎

*Remark* D.1. It is worth noting that, by standard arguments, ($\mathcal{H}_2$) follows directly from ($\mathcal{H}_1$) if the local Lipschitz condition on $\nabla f \circ Q$ is strengthened to a *global* Lipschitz condition. This specific kind of global condition was already employed in the analysis of *Mirror Langevin dynamics* for sampling problems, cf. Zhang et al. [118].

In the next two lemmas, we discuss standard settings under which the objective function $f$ satisfies the above assumptions ensuring existence and uniqueness of solution.

**Lemma D.1.** *If $\nabla f$ is globally $L$-Lipschitz continuous, then both ($\mathcal{H}_1$) and ($\mathcal{H}_2$) hold true.*

*Proof.* The mirror map $Q$ is $(1/K)$-Lipschitz continuous since $h$ is $K$-strongly convex (cf. Proposition B.1), and thus

$$\|(\nabla f \circ Q)(y) - (\nabla f \circ Q)(y')\|_* \leq L\|Q(y) - Q(y')\| \leq \frac{L}{K}\|y - y'\|_*. \tag{D.9}$$

This leads to $\nabla f \circ Q$ being globally Lipschitz continuous on $\mathcal{Y}$, which directly implies ($\mathcal{H}_1$) and ($\mathcal{H}_2$). ∎

*Remark* D.2. The globally Lipschitz setting of Lemma D.1 coincide with the framework commonly adopted in prior analyses of stochastic mirror flows—and, more specifically, of the stochastic mirror flow with diffusive noise (SMF); see for instance [70, 91, 113]. By contrast, relaxing this requirement to ($\mathcal{H}_1$) and ($\mathcal{H}_2$) allows the use of objective functions whose gradients may blow up at the boundary of the state $\mathcal{X}$, a feature that is essential when $h$ is steep and $f$ is strongly convex relative to $h$.

**Lemma D.2.** *If $h$ is steep and $\nabla f$ is locally Lipschitz on $\mathrm{ri}\,\mathcal{X}$, then $f$ satisfies ($\mathcal{H}_1$).*

*Proof.* Fix $y_0 \in \mathcal{Y}$. Since $h$ is steep, we know that $x_0 := Q(y_0)$ belongs to $\mathrm{ri}\,\mathcal{X}$. In particular, there exists a relatively compact subset $\mathcal{K} \subset \mathrm{ri}\,\mathcal{X}$ such that $x_0 \in \mathcal{K}$. By the local Lipschitzness of $\nabla f$, it follows that it is Lipschitz continuous on $\mathcal{K}$ with some finite constant $L(\mathcal{K}) \geq 0$. Now, due to $Q$ being continuous on $\mathcal{Y}$, we can find a neighborhood $\mathcal{U}$ of $y_0$ such that $Q(y) \in \mathcal{K}$ whenever $y \in \mathcal{U}$. We then proceed to show that $\nabla f \circ Q$ is Lipschitz continuous on $\mathcal{U}$. To do so, let $y, y' \in \mathcal{U}$. By construction of $\mathcal{U}$, we have $Q(y), Q(y') \in \mathcal{K}$, and thus

$$\|(\nabla f \circ Q)(y) - (\nabla f \circ Q)(y')\|_* \leq L(\mathcal{K})\|Q(y) - Q(y')\| \leq \frac{L(\mathcal{K})}{K}\|y - y'\|_*, \tag{D.10}$$

where we have used that $Q$ is $(1/K)$-Lipschitz continuous on $\mathcal{Y}$ (see Proposition B.1). It follows that, for every $y_0 \in \mathcal{Y}$, $\nabla f \circ Q$ is Lipschitz continuous on a neighborhood of $y_0$, and thus $\nabla f \circ Q$ is locally Lipschitz on $\mathcal{Y}$ as needed. ∎

*Remark* D.3. If $h$ is not steep, the conclusion of Lemma D.2 remains valid provided that $\nabla f$ is locally Lipschitz on $\mathcal{X}_h \equiv \mathrm{dom}\,\partial h$.

Beyond providing existence and uniqueness of solutions to (LMF), assumptions ($\mathcal{H}_1$) and ($\mathcal{H}_2$) also allows us to obtain the finiteness of the moments of $Y(t)$ up to order $p$.

**Lemma D.3** (Moments of the dual process)**.** *Under assumptions ($\mathcal{H}_1$) and ($\mathcal{H}_2$), the solution $Y(t)$ to (LMF) satisfies $\mathbb{E}[\sup_{s \leq t}\|Y(s)\|_*^p] < \infty$ for every $t \geq 0$.*

*Proof.* Fix an horizon $T > 0$ and let $0 \leq t \leq T$. Applying the standard inequality $(a + b + c)^p \leq 3^{p-1}(a^p + b^p + c^p)$ to the definition of (LMF), we obtain

$$\|Y(t)\|_*^p \leq 3^{p-1}\left\{\|Y(0)\|_*^p + \left\|\int_0^t (\nabla f \circ Q)(\eta(s)Y(s))ds\right\|_*^p + \|L(t)\|_*^p\right\}$$

---

[7]To be rigorous, the argument above only establishes existence and uniqueness for the SDE $dY'(t) = -(\nabla f \circ Q)(\eta(t-)Y'(t-))dt + dL(t)$, where the left limit of the drift is taken. This is due to the fact that stochastic processes are required to be predictable—and therefore almost left-continuous—for being used as integrands, see Appendix C.3. However, in our case, $Y'$ has càdlàg paths and thus it can only have at most countably many jumps. Consequently, $Y'(t-) = Y'(t)$ Lebesgue almost everywhere. Since $\eta$ and $\nabla f \circ Q$ are continuous, it follows that $\int_0^t (\nabla f \circ Q)(\eta(s-)Y'(s-))ds = \int_0^t (\nabla f \circ Q)(\eta(s)Y'(s))ds$, which implies that $Y'$ is in fact also the unique solution to (LMF).

$$\leq 3^{p-1}\left\{\|Y(0)\|_*^p + t^{p-1}\int_0^t \|(\nabla f \circ Q)(\eta(s)Y(s))\|_*^p \, ds + \sup_{r \leq T}\|L(r)\|_*^p\right\}$$

$$\leq 3^{p-1}\left\{\|Y(0)\|_*^p + \sup_{r \leq T}\|L(r)\|_*^p + t^{p-1}\int_0^t M^p(1 + \|\eta(s)Y(s)\|_*)^p \, ds\right\}$$

$$\leq 3^{p-1}\left\{\|Y(0)\|_*^p + \sup_{r \leq T}\|L(r)\|_*^p + 2^{p-1}M^p t^{p-1} + 2^{p-1}M^p\eta(0)^p t^{p-1}\int_0^t \|Y(s)\|_*^p \, ds\right\}$$

$$\leq 3^{p-1}\left\{\|Y(0)\|_*^p + \sup_{r \leq T}\|L(r)\|_*^p + 2^{p-1}M^p T^{p-1} + 2^{p-1}M^p\eta(0)^p T^{p-1}\int_0^t \|Y(s)\|_*^p \, ds\right\}, \tag{D.11}$$

where the second line comes from Hölder's inequality and the third one from the global growth condition ($\mathcal{H}_2$). For notational convenience, let us define

$$Z(T) = 3^{p-1}\left\{\|Y(0)\|_*^p + \sup_{r \leq T}\|L(r)\|_*^p + 2^{p-1}M^p T^{p-1}\right\} \quad \text{and} \quad C(T) = 6^{p-1}M^p\eta(0)^p T^{p-1}. \tag{D.12}$$

Taking the supremum over $s \leq t$ therefore yields

$$\sup_{s \leq t}\|Y(s)\|_*^p \leq Z(T) + C(T)\int_0^t \sup_{u \leq s}\|Y(u)\|_*^p \, ds. \tag{D.13}$$

Given the particular form of the above inequality, one would be tempted to take the expectation and proceed by applying Grönwall's lemma. However, at this stage we do not know if the left-hand side of Eq. (D.13) admits a finite expectation since this is the result we aim to prove, hence rendering the argument slightly more intricate. To proceed, we instead need to localize our bound using the stopping times $\tau_n = \{t \geq 0 : \|Y(t)\|_* \geq n\}$. We first show that $\mathbb{E}\big[\sup_{s \leq \tau_n \wedge t}\|Y(s)\|_*^p\big]$ is finite. Indeed, note that if $s < \tau_n \wedge t$ then $\|Y(s)\|_* < n$ by definition. On the other hand, for $s = \tau_n \wedge t$,

$$\|Y(\tau_n \wedge t)\|_* = \|Y(\tau_n \wedge t-) + \Delta Y(\tau_n \wedge t)\|_* = \|Y(\tau_n \wedge t-) + \Delta L(\tau_n \wedge t)\|_*$$

$$\leq \|Y(\tau_n \wedge t-)\|_* + \|L(\tau_n \wedge t)\|_* + \|L(\tau_n \wedge t-)\|_*$$

$$\leq n + 2\sup_{r \leq T}\|L(r)\|_*. \tag{D.14}$$

But $L(r)$ is a martingale with finite $p$-th moment, so Doob's $L^p$-maximal inequality (see e.g., Karatzas & Shreve [45]) can be applied to the positive submartingale $\|L(r)\|_*$ to obtain

$$\mathbb{E}[\sup_{r \leq T}\|L(r)\|_*^p] \leq \left(\frac{p}{p-1}\right)^p \mathbb{E}[\|L(T)\|_*^p] < \infty. \tag{D.15}$$

Combined with Eq. (D.14), this yields

$$\mathbb{E}[\sup_{s \leq \tau_n \wedge t}\|Y(s)\|_*^p] \leq 2^{p-1}\left\{n^p + 2\,\mathbb{E}[\sup_{r \leq T}\|L(r)\|_*^p]\right\} < \infty, \tag{D.16}$$

which proves that $g_n(t) \coloneqq \mathbb{E}[\sup_{s \leq \tau_n \wedge t}\|Y(s)\|_*^p]$ is a well-defined function on $[0, T]$. We can therefore apply Eq. (D.13) with $t \leftarrow \tau_n \wedge t$ and take the expectation on both sides, leading to

$$g_n(t) \leq \mathbb{E}[Z(T)] + C(T)\,\mathbb{E}\left[\int_0^{\tau_n \wedge t}\sup_{u \leq s}\|Y(u)\|_*^p \, ds\right] \leq \mathbb{E}[Z(T)] + C(T)\int_0^t g_n(s)ds, \tag{D.17}$$

where $\mathbb{E}[Z(T)]$ is finite thanks to Eq. (D.16). Invoking Grönwall's lemma then provides the estimate

$$g_n(t) \leq \mathbb{E}[Z(T)]e^{C(T)t} \leq \mathbb{E}[Z(T)]e^{C(T)T}. \tag{D.18}$$

Note that this upper bound does not depend on $n$ and that $\sup_{s \leq \tau_n \wedge t}\|Y(s)\|_*^p$ increases to $\sup_{s \leq t}\|Y(s)\|_*^p$ as $n \to \infty$, so the monotone convergence theorem yields

$$\mathbb{E}[\sup_{s \leq t}\|Y(s)\|_*^p] \leq \mathbb{E}[Z(T)]e^{C(T)T} < \infty, \tag{D.19}$$

hence concluding the proof. ∎

In particular, Lemma D.3 implies that the drift term of (LMF) admits finite moments of order $p$ even when the gradient of the objective function explodes at the boundary of $\mathcal{X}$.

**Corollary D.1.** *Under assumptions* $(\mathcal{H}_1)$ *and* $(\mathcal{H}_2)$,

$$\mathbb{E}\left[\int_0^t \|\nabla f(X(s))\|_*^p\, ds\right] < \infty \quad \text{for every } t \geq 0. \tag{D.20}$$

*Proof.* Thanks to the growth condition $(\mathcal{H}_2)$, we can write

$$\mathbb{E}\left[\int_0^t \|\nabla f(X(s))\|_*^p\, ds\right] = \mathbb{E}\left[\int_0^t \|(\nabla f \circ Q)(\eta(s)Y(s))\|_*^p\, ds\right] \leq \mathbb{E}\left[\int_0^t M^p(1 + \|\eta(s)Y(s)\|_*)^p\, ds\right]$$

$$\leq 2^{p-1}M^p\left(1 + \eta(0)^p\, \mathbb{E}[\sup_{s\leq t}\|Y(s)\|_*^p]\right) \tag{D.21}$$

where the right-hand side is finite due to Lemma D.3, therefore proving the claim. ∎

**D.2. Convex setting.** Inspired by the continuous-time analysis of Mertikopoulos & Staudigl [70], in this section we study the convergence of (LMF) for convex functions by analyzing the behavior of the $\eta$-*deflated* Fenchel coupling

$$E(q, y; t) = \frac{1}{\eta(t)}F(q, \eta(t)y), \quad q \in \mathcal{X}, y \in \mathcal{Y} \tag{D.22}$$

along trajectories of (LMF). This analysis naturally calls for Itô's formula, the stochastic counterpart of the standard chain rule of differential calculus. Unfortunately, the classical version of Itô's formula only applies to functions that are twice continuously differentiable, which is generally not the case for $y \mapsto E(q, y; t)$ unless $h$ is steep. Nevertheless, as we show in Appendix C.6, Itô's formula can be extended to functions that that are merely $L$-smooth and convex. Since $E$ satisfies these conditions by Lemma B.1, we may apply Theorem C.1 to derive the following result:

**Lemma D.4** (Stochastic descent lemma). *For every fixed* $q \in \mathcal{X}$, *the process* $E(t) \equiv E(q, Y(t); t)$ *satisfies*

$$dE \leq -\frac{\dot{\eta}}{\eta^2}(h(q) - h(X))dt - \langle X - q, \nabla f(X)\rangle dt + \frac{\eta\sigma_{\text{tame}}^2}{2K}dt + dI + d\xi + d\zeta, \tag{D.23}$$

*where:*

1. $I(t) = \int_0^t \int_{\|z\|_*\geq c} \frac{1}{\eta}[F(q, \eta Y + \eta z) - F(q, \eta Y)]\nu(dz)ds - \int_0^t \int_{\|z\|_*\geq c} \langle X - q, z\rangle\nu(dz)ds,$

2. $\xi(t) = \int_0^t \langle X - q, \Sigma dW\rangle + \int_0^t \int_{\|z\|_*< c} \frac{1}{\eta}[F(q, \eta Y + \eta z) - F(q, \eta Y)]\tilde{N}(ds, dz),$

3. $\zeta(t) = \int_0^t \int_{\|z\|_*\geq c} \frac{1}{\eta}[F(q, \eta Y + \eta z) - F(q, \eta Y)]\tilde{N}(ds, dz).$

*Proof.* First, let us define the auxiliary stochastic process $Z(t) = \eta(t)Y(t)$. Due to the integration by parts theorem for semi-martingales and thanks to the fact that $\eta$ is of finite variation, we get (cf. [44, Lemma 20.10])

$$dZ(t) = Y(t-)d\eta(t) + \eta(t-)dY(t)$$
$$= \dot{\eta}(t)Y(t-)dt + \eta(t)dY(t)$$
$$= \dot{\eta}Y dt - \eta\left[\nabla f(X) + \int_{\|z\|_*\geq c} z\nu(dz)\right]dt + \eta\Sigma dW + \eta\int_{\|z\|_*<c} z\tilde{N}(dt, dz) + \eta\int_{\|z\|_*\geq c} zN(dt, dz), \tag{D.24}$$

where we have suppress the dependence on time in the last equality to lighten the notations. Note that $Z(t)$ is evidently a Lévy-type process thanks to Lemma D.3 and Corollary D.1. Next, we consider the function $\psi(z) = F(q, z)$, which is obviously convex as the sum of convex functions, and which satisfies $\nabla\psi(z) = \nabla h^*(z) - q = Q(z) - q$ and thus is also $(1/K)$-smooth by Proposition B.1. Consequently, we can again use the integration by parts formula in tandem with the weak Itô's formula of Theorem C.1 to obtain

$$dE = -\frac{\dot{\eta}}{\eta^2}\psi(Z)dt + \frac{1}{\eta}d\psi(Z)$$

$$\leq -\frac{\dot{\eta}}{\eta^2}\psi(Z)dt + \frac{1}{\eta}\langle\nabla\psi(Z), dZ^c\rangle + \frac{\eta\sigma_{\text{tame}}^2}{2K}dt$$

$$+ \int_{\|z\|_* < c}\frac{1}{\eta}[\psi(Z+\eta z) - \psi(Z)]\tilde{N}(dt, dz) + \int_{\|z\|_* \geq c}\frac{1}{\eta}[\psi(Z+\eta z) - \psi(Z)]N(dt, dz), \quad (D.25)$$

where $dZ^c$ denotes the continuous part of $Z$. Note that by definition of (LMF), we have $\nabla\psi(Z) = Q(Z) - q = X - q$. Moreover, by optimality of $X$, we also have $h^*(Z) = \langle X, Z\rangle - h(X)$ and thus

$$-\frac{\dot{\eta}}{\eta^2}\psi(Z) = -\frac{\dot{\eta}}{\eta^2}F(q, Z) = -\frac{\dot{\eta}}{\eta^2}(h(q) - h(X) + \langle X - q, Z\rangle). \quad (D.26)$$

This leads to

$$-\frac{\dot{\eta}}{\eta^2}\psi(Z)dt + \frac{1}{\eta}\langle\nabla\psi(Z), dZ^c\rangle = \left[-\frac{\dot{\eta}}{\eta^2}\psi(Z) + \frac{\dot{\eta}}{\eta}\langle X - q, Y\rangle\right]dt - \langle X - q, \nabla f(X)\rangle dt - \int_{\|z\|_* \geq c}\langle X - q, z\rangle\nu(dz)dt$$

$$+ \langle X - q, \Sigma dW\rangle$$

$$= -\frac{\dot{\eta}}{\eta^2}(h(q) - h(X))dt + \langle X - q, \nabla f(X)\rangle dt - \int_{\|z\|_* \geq c}\langle X - q, z\rangle\nu(dz)dt$$

$$+ \langle X - q, \Sigma dW\rangle, \quad (D.27)$$

which, when combined with Eq. (D.25) and after rearranging the terms, yields exactly the upper bound on $dE$ stated in the lemma. ∎

*Remark* D.4. When $L$ is continuous—that is, when it reduces to a Brownian motion—both $I(t)$ and $\zeta(t)$ are identically zero, and $\xi(t)$ simplifies to $\int_0^t\langle X - q, \Sigma dW\rangle$. In this case, the analysis is considerably simpler, since the noise process affecting $E$ is just the scalar product between $X(t) - q$ (which is always bounded) and the Brownian motion $\Sigma W(t)$ (which has exponentially light tails). Moreover, it then also recovers the analogous result proved for (SMF) by Mertikopoulos & Staudigl [70] under diffusion-like random perturbations.

In the next three lemmas, we analyze the long-run behavior of the stochastic processes $I, \xi$ and $\zeta$. We begin by showing that $I$ can be upper bounded in terms of the heavy noise intensity $\sigma_{\text{heavy}}^p$ and a specific integral involving the learning rate.

**Lemma D.5** (Estimation of $I$). $I(t) \leq \dfrac{\|\mathcal{X}\|^{2-p}\sigma_{\text{heavy}}^p}{K^{p-1}}\displaystyle\int_0^t\eta(s)^{p-1}ds$, where $\|\mathcal{X}\| := \max_{x,x'}\|x' - x\|$ is the diameter of $\mathcal{X}$.

*Proof.* To bound this quantity, we use Lemma B.5 to obtain

$$\int_{\|z\|_* \geq c}\frac{1}{\eta}[F(q, Z+\eta z) - F(q, Z)]\nu(dz) \leq \int_{\|z\|_* \geq c}\frac{1}{\eta}\left[\langle Z + \eta z - Z, Q(Z) - q\rangle + \frac{\|\mathcal{X}\|^{2-p}}{K^{p-1}}\|Z + \eta z - Z\|_*^p\right]\nu(dz)$$

$$= \int_{\|z\|_* \geq c}\langle X - q, z\rangle\nu(dz) + \frac{\|\mathcal{X}\|^{2-p}}{K^{p-1}}\eta^{p-1}\int_{\|z\|_* \geq c}\|z\|_*^p\nu(dz), \quad (D.28)$$

where the first term compensates the second term of $I$, thus finishing the proof. ∎

Next, we use the stochastic integration tools introduced in Appendix C.3 to show that $\xi$ is a square-integrable martingale, and to characterize its asymptotic long-run behavior via the law of iterated logarithm.

**Lemma D.6** (Asymptotic growth of $\xi$). $\xi$ is a square-integrable martingale satisfying $\xi(t) = \mathcal{O}\left(\sigma_{\text{tame}}\|\mathcal{X}\|\sqrt{t\log\log t}\right)$ with probability 1.

*Proof.* First, note that

$$\|\Sigma^\top(X - q)\|^2 \leq \|X - q)\|^2\|\Sigma\|_F^2 \leq \|\mathcal{X}\|^2\sigma_0^2 \quad (D.29)$$

so $\int_0^t\langle X - q, \Sigma W\rangle$ is a square-integrable martingale. On the other hand, using Lemma B.4, we also get

$$\int_0^t\int_{\|z\|_* < c}\frac{1}{\eta^2}|F(q, Z+\eta z) - F(q, Z)|\nu(dz)ds \leq \|\mathcal{X}\|^2\int_0^t\int_{\|z\|_* < c}\frac{1}{\eta^2}\|Z + \eta z - Z\|_*^2\nu(dz)ds = \|\mathcal{X}\|^2\sigma_{\text{short}}^2 t, \quad (D.30)$$

where $\sigma^2_{\text{short}}$ is finite since $\nu$ is a Lévy measure. As a result,

$$G(t, z) := \frac{1}{\eta}[F(q, Z + \eta z) - F(q, Z)] \in \mathcal{H}^2_\nu(T, \{\|z\|_* < c\}; \mathbb{R}) \quad \text{for every } T \geq 0, \tag{D.31}$$

and thus $\int_0^t \int_{\|z\|_* < c} G(s, z) \tilde{N}(ds, dz)$ is also a square-integrable martingale. It then immediately follows that $\xi$ is itself a square-integrable martingale. To obtain the asymptotic growth of $\xi$, we use the law of iterated logarithm for square-integrable martingales (cf. Lemma C.2). Using standard stochastic calculus rules and the previous computations, we obtain that the predictable quadratic variation of $\xi$ satisfies

$$d\langle\xi\rangle = \|\Sigma^\top(X - q)\|^2 dt + \int_{\|z\|_* < c} \frac{1}{\eta^2}|F(q, Z + \eta z) - F(q, Z)|^2 \nu(dz) dt \leq \sigma^2_{\text{tame}}\|\mathcal{X}\|^2 dt. \tag{D.32}$$

Moreover, the jumps of $\xi$ are bounded as $|\Delta\xi(t)| \leq c\|\mathcal{X}\|$ (a.s.) thanks to Lemma B.4 and the fact that $\|z\|_* \leq c$. Assume first that $\lim_{t\to\infty}\langle\xi\rangle(t) = \infty$. Then, the law of iterated logarithm of Lemma C.2 applies to $\xi$ and yields

$$\limsup_{t\to\infty} \frac{|\xi|}{\sigma_{\text{tame}}\|\mathcal{X}\|\sqrt{t \log\log t}} = \limsup_{t\to\infty} \frac{|\xi|}{\sqrt{2\langle\xi\rangle \log\log\langle\xi\rangle}} \frac{\sqrt{2\langle\xi\rangle \log\log\langle\xi\rangle}}{\sigma_{\text{tame}}\|\mathcal{X}\|\sqrt{t \log\log t}}$$

$$\leq \limsup_{t\to\infty} \frac{|\xi|}{\sqrt{2\langle\xi\rangle \log\log\langle\xi\rangle}} \frac{2\sigma_{\text{tame}}\|\mathcal{X}\|\sqrt{t \log\log t}}{\sigma_{\text{tame}}\|\mathcal{X}\|\sqrt{t \log\log t}} \leq 2, \tag{D.33}$$

where the second-to-last inequality holds thanks to Eq. (D.32) and the fact that $t$ can be taken as large as needed. It follows that $\xi(t) = \mathcal{O}\left(\sigma_{\text{tame}}\|\mathcal{X}\|\sqrt{t \log\log t}\right)$ on the event $\{\lim_{t\to\infty}\langle\xi\rangle(t) = \infty\}$. On the other hand, when $\lim_{t\to\infty}\langle\xi\rangle(t) < \infty$, [55, Proposition 1] implies that $\xi(t)$ converges to an almost-surely finite random variable. For this reason, it holds that $\xi(t) = o\left(\sigma_{\text{tame}}\|\mathcal{X}\|\sqrt{t \log\log t}\right)$ on the event $\{\lim_{t\to\infty}\langle\xi\rangle(t) < \infty\}$, and thus $\xi(t) = \mathcal{O}\left(\sigma_{\text{tame}}\|\mathcal{X}\|\sqrt{t \log\log t}\right)$ (a.s.), which concludes the proof. ∎

Finally, we establish that $\zeta$ is also a martingale using standard stochastic integration arguments. However, $\zeta$ is not square-integrable in general, and we therefore rely on a refined version of the law of large numbers for discontinuous martingales to control its growth rate.

**Lemma D.7** (Asymptotic growth of $\zeta$). *$\zeta$ is a martingale satisfying $\zeta(t) = \bar{\mathcal{O}}\left(t^{1/p}\right)$ with probability $1$.*[8]

*Proof.* Similarly to the proof of Lemma D.6, we have

$$\int_0^t \int_{\|z\|_* \geq c} \frac{1}{\eta}|F(q, Z + \eta z) - F(q, Z)|\nu(dz) \leq \|\mathcal{X}\|\left(\int_{\|z\|_* \geq c} \|z\|_* \nu(dz)\right)t, \tag{D.34}$$

where $\int_{\|z\|_* \geq c} \|z\|_* \nu(dz)$ is finite since $L$ is an integrable process. This means that

$$H(t, z) := \frac{1}{\eta}[F(q, Z + \eta z) - F(q, Z)] \in \mathcal{H}^1_\nu(T, \{\|z\|_* \geq c\}; \mathbb{R}) \quad \text{for all } T \geq 0, \tag{D.35}$$

which immediately leads to $\zeta$ being a martingale. Letting $A(t) = t^{\frac{1}{p}+\varepsilon}$ for some $\varepsilon > 0$, we also have that

$$\phi(t) := \int_{\|z\|_* \geq c} \frac{|H(t, z)|^2 \wedge |H(t, z|^p}{(1 + A(t))^p}\nu(dz) \leq \frac{\|\mathcal{X}\|^p}{t^{1+\varepsilon p}} \int_{\|z\|_* \geq c} \|z\|_*^p \nu(dz) = \frac{\|\mathcal{X}\|^p \sigma^p_{\text{heavy}}}{t^{1+\varepsilon p}} \tag{D.36}$$

for $t$ big enough. It follows that $\phi$ is almost-surely integrable on $[0, \infty)$, and thus $\zeta(t)/t^{\frac{1}{p}+\varepsilon} \to 0$ (a.s.) for every $\varepsilon > 0$ by the strong law of large number (cf. Lemma C.1). It follows that $\zeta(t) = \bar{\mathcal{O}}\left(t^{1/p}\right)$, hence finishing the proof. ∎

---

[8]Recall that $g(t) = \bar{\mathcal{O}}(t^\alpha)$ means that $g(t) = \mathcal{O}(t^{\alpha+o(1)})$, that is, $g(t) = \mathcal{O}(l(t)t^\alpha)$ where $l$ is a function of subpolynomial growth such that $l(t)/t^\delta \to 0$ for all $\delta > 0$.

With the preparatory estimates on the drift term $I$ and the martingales $\xi$ and $\zeta$ in hand, we are now in a position to prove Proposition 1 from the main text (restated below for convenience):

**Proposition 1.** *Under* (LMF), $E(t)$ *is bounded as*

$$E(t) - E(0) \leq \int_0^t \langle \nabla f(X(s)), q - X(s) \rangle \, ds + \xi(t) + \zeta(t)$$
$$+ V_{\text{tame}}^2 \int_0^t \eta(s) \, ds + V_{\text{heavy}}^p \int_0^t \eta(s)^{p-1} \, ds$$
$$- \int_0^t \frac{\dot{\eta}(s)}{\eta(s)^2} [h(q) - h(X(s))] \, ds \,, \tag{17}$$

*where* $\|\mathcal{X}\| := \max_{x,x'} \|x' - x\|$ *is the diameter of* $\mathcal{X}$*, and* $\xi(t)$*,* $\zeta(t)$ *are martingales (the former square-integrable).*

*Proof of Proposition 1.* The result follows directly from Lemma D.4 by integrating over the interval $[0, t]$ and invoking the bound on $I$ established in Lemma D.5. The martingale properties of $\xi$ and $\zeta$—the former being square-integrable and the latter integrable—is then a consequence of Lemmas D.6 and D.7. ∎

On the other hand, the previous result also allows us to derive the following theorem, which will serve as a baseline for many of the convergence rates established in the sequel.

**Theorem D.1.** *For every fixed* $q \in \mathcal{X}$*,* (LMF) *enjoys the bounds*

$$\mathbb{E}\left[\int_0^t \langle X(s) - q, \nabla f(X(s)) \rangle ds\right] \leq t R_0(t) \tag{D.37}$$

*and, with probability one,*

$$\int_0^t \langle X(s) - q, \nabla f(X(s)) \rangle ds \leq t R_0(t) + \bar{\mathcal{O}}\left(t^{1/p}\right) + \mathcal{O}\left(\sigma_{\text{tame}} \|\mathcal{X}\| \sqrt{t \log \log t}\right), \tag{D.38}$$

*where*

$$R_0(t) = \frac{\langle x_0 - q, y_0 \rangle}{t} + \frac{H}{t\eta(t)} + \frac{\sigma_{\text{tame}}^2}{2Kt} \int_0^t \eta(s) \, ds + \frac{\|\mathcal{X}\|^{2-p} \sigma_{\text{heavy}}^p}{K^{p-1}t} \int_0^t \eta(s)^{p-1} \, ds \tag{D.39}$$

*and* $H := \max h - \min h$.

*Proof.* First, note that $\langle X(s-) - q, \nabla f(X(s)) \rangle = \langle X(s) - q, \nabla f(X(s)) \rangle$ Lebesgue-a.e. since the jumps of $L$ are countable, and hence

$$\int_0^t \langle X(s-) - q, \nabla f(X(s)) \rangle ds = \int_0^t \langle X(s) - q, \nabla f(X(s)) \rangle ds \quad (a.s.). \tag{D.40}$$

Moreover, the monotony of $\eta$ and the compactness of $\mathcal{X}$ allows us to write

$$- \int_0^t \frac{\dot{\eta}}{\eta^2} (h(q) - h(X)) ds \leq H \int_0^t -\frac{\dot{\eta}}{\eta^2} ds = \frac{H}{\eta(t)} - \frac{H}{\eta(0)}, \tag{D.41}$$

where we recall that $H = \max h - \min h$. On the other hand, from the definition of $E$ and by optimality of $X(0) = x_0$, we also get

$$E(0) = \frac{1}{\eta(0)} F(q, \eta(0)y_0) = \frac{1}{\eta(0)} [h(q) - h(x_0) + \langle x_0 - q, \eta(0)y_0 \rangle] \leq \frac{H}{\eta(0)} + \langle x_0 - q, y_0 \rangle. \tag{D.42}$$

Combining these estimates with those of Lemmas D.4–D.7, rearranging the terms, and using that $E(t) \geq 0$ finally lead to the upper bound (D.38). The in-expectation inequality (D.37) also immediately follows since $\mathbb{E}[\xi(t)] = \mathbb{E}[\zeta(t)] = 0$ in virtue of the fact that they are martingales starting from zero. ∎

*Remark* D.5. Note that, compared to the definition of $R_0(t)$ given in the main text, the version presented here includes the additional term $\frac{1}{t} \langle x_0 - q, y_0 \rangle$. This accounts for the fact that (LMF) may be initialized at any point $x_0 \in \mathcal{X}_h$, rather than only at the prox-center $x_c = \arg \min h$ of $\mathcal{X}$, which was assumed in the main text for simplicity.

With Theorem D.1 in hand, we can therefore establish Theorem 1 as a direct application of the convexity of $f$. The following result presents a slightly more general version of Theorem 1, covering the case where (LMF) is not necessarily initialized at the prox-center of $\mathcal{X}$:

**Theorem D.2** (Convergence rate for convex functions). *Under* (LMF), *the time-averaged process* $\bar{X}(t) = (1/t) \int_0^t X(s)\, ds$ *enjoys the bounds*

$$\mathbb{E}[f(\bar{X}(t))] \leq \min f + R_0(t) \tag{D.43}$$

*and, with probability* 1,

$$f(\bar{X}(t)) \leq \min f + R_0(t) + \bar{\mathcal{O}}\left(t^{-\frac{p-1}{p}}\right) + \mathcal{O}\left(\sigma_{\text{tame}}\|\mathcal{X}\|\sqrt{\frac{\log\log t}{t}}\right) \tag{D.44}$$

*In particular, if* (LMF) *is run with learning rate* $\eta(t) = 1/t^{1/p}$, *we have*

$$R_0(t) = \frac{\langle x_0 - q, y_0 \rangle}{t} + \frac{p}{p-1} \frac{\sigma_{\text{tame}}^2}{2Kt^{1/p}} + \frac{K^{p-1}H + p\|\mathcal{X}\|^{2-p}\sigma_{\text{heavy}}^p}{K^{p-1}t^{(p-1)/p}} = \mathcal{O}(1/t^{(p-1)/p})\,. \tag{D.45}$$

*Proof.* Since $f$ is convex,

$$\int_0^t \langle X(s) - x^*, \nabla f(X(s)) \rangle ds \geq \int_0^t [f(X(s)) - f(x^*)]ds \geq t\big[f(\bar{X}(t)) - \min f\big], \tag{D.46}$$

and the results therefore follow directly from Theorem D.1. ∎

*Remark* D.6. By convexity, we also get

$$\int_0^t \langle X(s) - x^*, \nabla f(X(s)) \rangle ds \geq \int_0^t [f(X(s)) - f(x^*)]ds \geq t\left[\min_{0 \leq s \leq t} f(X(s)) - \min f\right]. \tag{D.47}$$

Consequently, the upper bounds of Theorem D.2 remain valid when $f(\bar{X}(t))$ is replaced by $\min_{0 \leq s \leq t} f(X(s))$.

**D.3. Strongly convex setting.** We now specialize our results to functions that are strongly convex, either with respect to the Euclidean norm $\|\cdot\|$ or relative to the regularizer $h$.

First, assume that $f$ is $\mu$-strongly convex in the usual sense, that is,

$$f(x') \geq f(x) + \langle \nabla f(x), x' - x \rangle + \frac{\mu}{2}\|x' - x\|^2 \tag{D.48}$$

for all $x, x' \in \mathcal{X}$. Applying the convergence guarantees of Theorem D.1 under this strong convexity assumption then yields the following rate of convergence of the time-averages toward the minimum of $f$:

**Theorem D.3** (Convergence rate for strongly convex functions). *If $f$ is $\mu$-strongly convex with minimum $x^* \in \mathcal{X}$, then the time-averaged process $\bar{X}(t)$ of* (LMF) *enjoys the convergence rates*

$$\mathbb{E}[\|\bar{X}(t) - x^*\|^2] \leq \frac{2}{\mu}R_0(t) \tag{D.49}$$

*and, with probability* 1,

$$\|\bar{X}(t) - x^*\|^2 \leq \frac{2}{\mu}R_0(t) + \bar{\mathcal{O}}\left(t^{-\frac{p-1}{p}}\right) + \mathcal{O}\left(\sigma_{\text{tame}}\|\mathcal{X}\|\sqrt{\frac{\log\log t}{t}}\right) \tag{D.50}$$

*In particular, if* (LMF) *is run with learning rate* $\eta(t) = 1/t^{1/p}$, *we have*

$$R_0(t) = \frac{\langle x_0 - q, y_0 \rangle}{t} + \frac{p}{p-1} \frac{\sigma_{\text{tame}}^2}{2Kt^{1/p}} + \frac{K^{p-1}H + p\|\mathcal{X}\|^{2-p}\sigma_{\text{heavy}}^p}{K^{p-1}t^{(p-1)/p}} = \mathcal{O}(1/t^{(p-1)/p})\,, \tag{D.51}$$

*and $\bar{X}(t) \to x^*$ both in mean square and with probability* 1 *as $t \to \infty$.*

*Proof.* By strong convexity of $f$,

$$\int_0^t \langle X(s) - x^*, \nabla f(X(s)) \rangle \geq \int_0^t \left[ f(X(s)) - \min f + \frac{\mu}{2} \|X(s) - x^*\|^2 \right] ds$$

$$\geq \frac{\mu}{2} \int_0^t \|X(s) - x^*\|^2 ds \geq \frac{\mu t}{2} \|\bar{X}(t) - x^*\|^2, \tag{D.52}$$

and Theorem D.1 therefore yields the desired upper bounds. ∎

*Remark* D.7. Similarly to the convex case, Theorem D.3 also yields the following complexity estimate: for any precision $\varepsilon > 0$, choosing a constant learning rate $\eta = \mathcal{O}\left((\varepsilon/\bar{\sigma})^{\frac{1}{p-1}}\right)$ ensures that $\mathbb{E}\left[\|\bar{X}(t) - x^*\|^2\right] \leq \varepsilon$ for every times $t = \Omega\left(\bar{\sigma}^{\frac{1}{p-1}} \varepsilon^{-\frac{p}{p-1}}\right)$.

As in the main text, we now estimate the amount of time that $X(t)$ spends in a neighborhood of $\arg \min f$. The following theorem serves as an extension of Theorem 2 to the setting where (LMF) is initialized at an arbitrary point of $\mathcal{X}$, rather than at its prox-center.

**Theorem D.4** (Concentration around global minimum). *Assume that $f$ is $\mu$-strongly convex with minimum $x^*$, and let*

$$\mu_t(\mathcal{U}_\delta) = \frac{1}{t} \int_0^t \mathbb{1}\{X(s) \in \mathcal{U}_\delta\} \, ds \tag{D.53}$$

*be the fraction of time that $X(t)$ spends in $\mathcal{U}_\delta = \{x \in \mathcal{X} : \|x - x^*\| \leq \delta\}$ up to time $t$. If (LMF) is run with constant learning rate $\eta(t) \equiv \eta$, we have*

$$\mathbb{E}[\mu_t(\mathcal{U}_\delta)] \geq 1 - B_t(\delta) \tag{D.54a}$$

*and, with probability* $1$,

$$\mu_t(\mathcal{U}_\delta) \geq 1 - B_t(\delta) + \bar{\mathcal{O}}\left(t^{-\frac{p-1}{p}}\right) + \mathcal{O}\left(\sigma_{\text{tame}}\|\mathcal{X}\| \sqrt{\frac{\log \log t}{t}}\right) \tag{D.54b}$$

*where*

$$B_t(\delta) = \frac{2\langle x_0 - x^*, y_0 \rangle}{\mu \delta^2 t} + \frac{2H}{\eta \mu \delta^2 t} + \frac{\eta \sigma_{\text{tame}}^2}{\mu K \delta^2} + \frac{2\eta^{p-1}\|\mathcal{X}\|^{2-p} \sigma_{\text{heavy}}^p}{\mu K^{p-1} \delta^2}. \tag{D.55}$$

*Proof.* Since $\mathbb{1}\{\|X(t) - x^*\| > \delta\} \leq \delta^{-2} \|X(t) - x^*\|^2$, the occupation measure $\mu_t$ satisfies

$$\mu_t(\mathcal{X} \setminus \mathcal{U}_\delta) = \frac{1}{t} \int_0^t \mathbb{1}\{\|X(s) - x^*\| > \delta\} ds \leq \frac{1}{\delta^2 t} \int_0^t \|X(s) - x^*\|^2 ds \leq \frac{2}{\mu \delta^2 t} \int_0^t \langle X(s) - x^*, \nabla f(X(s)) \rangle ds, \quad \text{(D.56)}$$

and Theorem D.4 therefore follows from Theorem D.1. ∎

*Remark* D.8. If the stochastic process $X$ is *positive recurrent* on ri $\mathcal{X}$—that is, if any relatively compact subset of ri $\mathcal{X}$ is reached by $X$ in finite expected time—then the occupation measure $\mu_t$ converges to the so-called *invariant measure* $\mu$ of $X$. This means that $\mu$ is the unique probability measure such that $X(t) \sim \mu$ whenever $X(0) \sim \mu$. Under ergodicity of the dynamics, $\mu$ is also the measure towards which the law of $X(t)$ converges from *any* initial condition. In this regards, Theorem D.4 therefore provides a non-asymptotic estimate on the concentration of the law of $X(t)$ around neighborhoods of the objective's minimum $x^*$. For the stochastic mirror flow (SMF) with constant learning rate, Mertikopoulos & Staudigl [70] showed that the primal trajectories are indeed positive recurrent—and hence admit a unique probability invariant measure—under Brownian-like perturbations. Extending this result to (LMD) is, however, substantially more challenging: Lévy-driven SDEs exhibit more intricate dynamical properties, and establishing positive recurrence in this setting thus requires additional care. We therefore leave this question open for future works.

As illustrated in Remark D.7, Theorem D.3 provides a lower bound on the time beyond which the ergodic averages remain within at most $\varepsilon$ of the objective's minimum in the mean square norm. The following theorem complements this result by giving an estimate on the average time required for the *last iterate* itself to reach a $\delta$-neighborhood of the minimum.

**Theorem D.5** (Hitting time estimate). *Assume that $f$ is $\mu$-strongly convex with minimum $x^* \in \mathcal{X}$, and let*

$$\tau_\delta = \inf\{t \geq 0 : \|X(t) - x^*\| \leq \delta\} \tag{D.57}$$

*denote the first time that $X(t)$ gets within $\delta$ of $x^*$ for some precision $\delta > 0$. If (LMF) is run with constant learning rate*

$$\eta < \min\left(\frac{\mu K}{2\sigma_{\text{tame}}^2}\delta^2, \left(\frac{\mu K}{4K^{2-p}\|\mathcal{X}\|^{2-p}\sigma_{\text{heavy}}^p}\right)^{\frac{1}{p-1}}\delta^{\frac{2}{p-1}}\right), \tag{D.58}$$

*then the averaged value of $\tau_\delta$ is bounded as*

$$\mathbb{E}[\tau_\delta] \leq \frac{2KF(x^*, \eta y_0)}{\mu K \eta \delta^2 - \eta^2 \sigma_{\text{tame}}^2 - 2\eta^p K^{2-p}\|\mathcal{X}\|^{2-p}\sigma_{\text{heavy}}^p}. \tag{D.59}$$

*Proof.* Since the learning $\eta(t)$ is assumed to be constant, Lemmas D.4–D.7 yield

$$E(t) \leq E(0) - \int_0^t \langle X - x^*, \nabla f(X)\rangle ds + \frac{\eta}{2K}\sigma_{\text{tame}}^2 t + \frac{\eta^{p-1}\|\mathcal{X}\|^{2-p}}{K^{p-1}}\sigma_{\text{heavy}}^p t + \xi(t) \tag{D.60}$$

where $\xi(t) := \xi(t) + \zeta(t)$ is a martingale. Moreover, $f$ is assumed to be $\mu$-strongly convex and so $\langle X - x^*, \nabla f(X)\rangle \geq \frac{\mu}{2}\|X - x^*\|^2$. In particular, for any fixed $t \geq 0$, we get

$$0 \leq \mathbb{E}[E(\tau_\delta \wedge t)] \leq E(0) - \frac{\mu}{2}\mathbb{E}\left[\int_0^{\tau_\delta \wedge t}\|X(s-) - x^*\|^2 ds\right] + \left(\frac{\eta}{2K}\sigma_{\text{tame}}^2 t + \frac{\eta^{p-1}\|\mathcal{X}\|^{2-p}}{K^{p-1}}\sigma_{\text{heavy}}^p\right)\mathbb{E}[\tau_\delta \wedge t], \quad \text{(D.61)}$$

where we have used that $\xi$ is a martingale starting from 0 and that $\tau_\delta \wedge t$ is a bounded stopping time to get $\mathbb{E}[\xi(\tau_\delta \wedge t)] = 0$. Now, note that by definition of $\tau_\delta$, we have $\|X(s-) - x^*\|^2 \geq \delta^2$ for every $s \leq \tau_\delta \wedge t$ due to $X$ having càdlàg paths. It follows that

$$0 \leq E(0) - \left(\frac{\mu}{2}\delta^2 - \frac{\eta}{2K}\sigma_{\text{tame}}^2 + \frac{\eta^{p-1}\|\mathcal{X}\|^{2-p}\sigma_{\text{heavy}}^p}{K^{p-1}}\right)\mathbb{E}[\tau_\delta \wedge t]. \tag{D.62}$$

Now, under the assumption Eq. (D.58) made on the size of $\eta$, it is clear that the factor in front of $\mathbb{E}[\tau_\delta \wedge t]$ is strictly positive, which leads after rearranging the terms to

$$\mathbb{E}[\tau_\delta \wedge t] \leq \frac{2KE(0)}{\mu K \delta^2 - \eta \sigma_{\text{tame}}^2 - 2\eta^{p-1}K^{2-p}\|\mathcal{X}\|^{2-p}\sigma_{\text{heavy}}^p}. \tag{D.63}$$

Since the function $t \mapsto \mathbb{E}[\tau_\delta \wedge t]$ is nondecreasing and since the right-hand side of Eq. (D.63) is uniform on $t$, the monotone convergence theorem can be then invoked to prove the upper bound claimed in the theorem. ∎

In particular, for a sufficiently small target radius $\delta$, the hitting time estimate of Theorem D.5 can be optimized by an appropriate choice of learning rate $\eta$, making its dependence on $\delta$ explicit. This leads to Theorem 3, whose statement is here split in two parts into order to provide precise conditions on the target $\delta$ depending on the nature of the noise. We first consider the case $p < 2$, for which the upper bound is determined solely by the heavy-tailed intensity of the noise.

**Corollary D.2** (Hitting time estimate, case $p < 2$). *Under the same assumptions as in Theorem D.5, further assume that $y_0 = 0$ and $p < 2$. Then, for every precision $\delta > 0$ satisfying*

$$\delta^2 \leq \frac{2\sigma_{\text{tame}}^2\|\mathcal{X}\|}{\mu}\left(\frac{2\sigma_{\text{heavy}}^p}{\sigma_{\text{tame}}^2}\right)^{\frac{1}{2-p}}, \tag{D.64}$$

*running (LMF) with constant learning rate*

$$\eta = \left(\frac{\mu K}{8K^{2-p}\|\mathcal{X}\|^{2-p}\sigma_{\text{heavy}}^p}\right)^{\frac{1}{p-1}}\delta^{\frac{2}{p-1}} \tag{D.65}$$

*yields the hitting-time bound*

$$\mathbb{E}[\tau_\delta] \le \frac{4H}{\mu K}\left(\frac{8\|\mathcal{X}\|^{2-p}\sigma_{\text{heavy}}^p}{\mu}\right)^{\frac{1}{p-1}}\delta^{-\frac{2p}{p-1}}.$$  (D.66)

*Proof.* For simplicity, let us define $a = \sigma_{\text{tame}}^2$, $b = 2K^{2-p}\|\mathcal{X}\|^{2-p}\sigma_{\text{heavy}}^p$ and $c = \mu K$. In particular, the expression of the learning rate $\eta$ can be written as

$$\eta = \left(\frac{c}{4b}\right)^{\frac{1}{p-1}}\delta^{\frac{2}{p-1}},$$  (D.67)

and the condition of Eq. (D.64) on the size of $\delta$ implies that

$$\left(\frac{c}{2b}\right)^{\frac{1}{p-1}}\delta^{\frac{2}{p-1}} \le \left(\frac{c}{2a}\right)\delta^2.$$  (D.68)

We therefore obtain the upper bound

$$\eta \le \frac{1}{2^{1/(p-1)}}\left(\frac{c}{2a}\right)\delta^2 \le \frac{c}{4a}\delta^2,$$  (D.69)

where we have used that $p \in (1, 2)$ to get $2^{1/(p-1)} \ge 2$. As a result, the denominator of Eq. (D.59) satisfies

$$\eta\left(c\delta^2 - a\eta - b\eta^{p-1}\right) \ge \left(\frac{c}{4b}\right)^{\frac{1}{p-1}}\delta^{\frac{2}{p-1}}\left(c\delta^2 - a\frac{c}{4a}\delta^2 - b\frac{c}{4b}\delta^2\right) = \frac{c}{2}\left(\frac{c}{4b}\right)^{\frac{1}{p-1}}\delta^{\frac{2p}{p-1}}.$$  (D.70)

Moreover, we also get $F(x^*, \eta y_0) = h(x^*) - h(x_0) \le H$ since $y_0 = 0$. The hitting time estimate of Theorem D.5 therefore yields

$$\mathbb{E}[\tau_\delta] \le \frac{4KH}{c}\left(\frac{4b}{c}\right)^{\frac{1}{p-1}}\delta^{-\frac{2p}{p-1}},$$  (D.71)

which recovers our claim when substituting $a$, $b$ and $c$ with their respective values. ∎

On the other hand, when the noise is *tame*—that is, when $p = 2$—we obtain the following optimized upper bound:

**Corollary D.3** (Hitting time estimate, case $p = 2$)**.** *Under the same assumptions as in Theorem D.5, further assume that $y_0 = 0$ and $p = 2$. Then, for every precision $\delta > 0$, running (LMF) with constant learning rate*

$$\eta = \frac{\mu K}{2\sigma_{\text{tot}}^2}\delta^2$$  (D.72)

*yields the hitting-time bound*

$$\mathbb{E}[\tau_\delta] \le \frac{8\sigma_{\text{tot}}^2 H}{\mu^2 K}\delta^{-4},$$  (D.73)

*where $\sigma_{\text{tot}}^2 = \sigma_{\text{tame}}^2 + 2\sigma_{\text{heavy}}^2$ is the total intensity of the noise.*

*Proof.* The upper bound is obtained directly when replacing $\eta$ by its value in Theorem D.5 and by bounding the numerator as in the proof of Corollary D.2. ∎

We now strengthen our assumptions by considering objectives $f$ that are *$\mu$-strongly convex relative to $h$*, that is, functions satisfying

$$f(x') \ge f(x) + \langle \nabla f(x), x' - x\rangle + \mu D(x', x)$$  (D.74)

for all $x \in \mathcal{X}_h$ and all $x' \in \mathcal{X}$. This sharper condition allows us to refine the convergence analysis of (LMF) to obtain stronger bounds on the last iterate rather than just on ergodic averages. We begin by establishing the first part of Theorem 9, which corresponds to the case where (LMF) is run with a constant learning rate:

**Theorem D.6** (Convergence rate for relatively strongly convex functions, constant learning rate)**.** *Let $h$ be a steep regularizer. If $f$ is $\mu$-strongly convex relative to $h$ with minimum $x^* \in \mathcal{X}$ and if $\eta(t) \equiv \eta > 0$ is constant, then*

$$\mathbb{E}[\|X(t) - x^*\|^2] \leq \frac{\sigma_{\text{tame}}^2}{\mu K^2}\eta + \frac{2\|\mathcal{X}\|^{2-p}\sigma_{\text{heavy}}^p}{\mu K^p}\eta^{p-1} + \frac{2D(x^*, x_0)}{K}e^{-\mu\eta t}. \tag{D.75}$$

Before proving Theorem D.6, we first establish a preliminary technical lemma that will allows us to rigorously differentiate the map $\mathbb{E}[E(t)]$ with respect to $t$.

**Lemma D.8.** *Let $F \colon [0, \infty) \to \mathbb{R}$ be a function such that*

$$F(t) - F(s) \leq \int_s^t g(r)dr \quad \text{for every } t \geq s \geq 0, \tag{D.76}$$

*where $g \colon [0, \infty) \to \mathbb{R}$ is a locally integrable function. Then $F$ is almost everywhere differentiable with $F'(t) \leq g(t)$.*

*Proof.* Let $G(t) = \int_0^t g(r)dr$ and $H = F - G$. Then Eq. (D.76) implies that $H(t) \leq H(s)$ for all $t \geq s \geq 0$, and thus $H$ is a nonincreasing function. In virtue of Lebesgue differentiation theorem, $H$ is therefore almost everywhere differentiable (on open intervals) and satisfies $H'(t) \leq 0$. Since $F = G + H$ is the sum of two almost everwhere differentiable function, it implies in turn that $F$ is also as such and that $F'(t) = G'(t) + H'(t) \leq g(t)$, which concludes the proof. ∎

*Proof of Theorem D.6.* Starting from Lemma D.4 and the estimates of Lemmas D.5–D.7, and noting that $D(x^*, X(t)) = F(x^*, \eta Y(t))$ since $X(t) \in \mathrm{ri}\,\mathcal{X}$ for all times (consequence of $h$ being steep, cf. Proposition B.2), we get

$$\frac{1}{\eta}\mathbb{E}[D(x^*, X(t))] = \mathbb{E}[E(t)]$$

$$\leq \mathbb{E}[E(s)] - \mathbb{E}\left[\int_s^t \langle X - x^*, \nabla f(X)\rangle dr\right] + \frac{\eta}{2K}\sigma_{\text{tame}}^2\int_s^t dr + \frac{\eta^{p-1}\|\mathcal{X}\|^{2-p}}{K^{p-1}}\sigma_{\text{heavy}}^p\int_s^t dr$$

$$\leq \frac{1}{\eta}\mathbb{E}[D(x^*, X(s))] - \int_s^t\left\{\mathbb{E}[f(X) - f(x^*) + \mu D(x^*, X)] - \left(\frac{\eta\sigma_{\text{tame}}^2}{2K} + \frac{\eta^{p-1}\|\mathcal{X}\|^{2-p}\sigma_{\text{heavy}}^p}{K^{p-1}}\right)\right\}dr$$

$$\leq \frac{1}{\eta}\mathbb{E}[D(x^*, X(s))] - \int_s^t\left\{\mu\,\mathbb{E}[D(x^*, X)] - \left(\frac{\eta\sigma_{\text{tame}}^2}{2K} + \frac{\eta^{p-1}\|\mathcal{X}\|^{2-p}\sigma_{\text{heavy}}^p}{K^{p-1}}\right)\right\}dr, \tag{D.77}$$

where the second inequality comes from the relative strong convexity of $f$, and the last inequality is due to $x^*$ being the minimum of $f$. Since the above inequality holds true for every $t \geq s \geq 0$, it follows from Lemma D.8 that the function $\varphi(t) := \mathbb{E}[D(x^*, X(t))]$ is almost-everywhere differentiable and that it satisfies

$$\varphi'(t) \leq -\mu\eta\varphi(t) + c_1\eta^2 + c_2\eta^p, \tag{D.78}$$

where we have set $c_1 = \frac{1}{2K}(\sigma_{\text{tame}}^2)$ and $c_2 = \frac{\|\mathcal{X}\|^{2-p}}{K^{p-1}}\sigma_{\text{heavy}}^p$. In particular, we can therefore use Grönwall's lemma to obtain

$$\mathbb{E}[D(x^*, X(t))] = \varphi(t) \leq \varphi(0)e^{-\mu\eta t} + \frac{c_1\eta^2 + c_2\eta^p}{\mu\eta}\left(1 - e^{-\mu\eta t}\right)$$

$$\leq \frac{\sigma_{\text{tame}}^2}{2K\mu}\eta + \frac{\|\mathcal{X}\|^{2-p}\sigma_{\text{heavy}}^p}{\mu K^{p-1}}\eta^{p-1} + D(x^*, x_0)e^{-\mu\eta t}, \tag{D.79}$$

and the result then follows thanks to the inequality $\|X(t) - x^*\|^2 \leq \frac{2}{K}D(x^*, X(t))$ (cf. Proposition B.2). ∎

*Remark* D.9. A result analogous to Theorem D.6 for Brownian-type stochastic perturbations was established by Tzen et al. [107] for the *mirror Langevin dynamics*, a special case of (SMF) with constant learning rate in which the diffusion coefficient is proportional to the Hessian of the regularizer. Their result arises as a byproduct of an interpretation of continuous-time mirror descents methods through the lens of inverse optimal control.

Finally, we now establish the second part of Theorem 9, concerning the convergence rates of the last iterate when the learning rate is decreasing.

**Theorem D.7** (Convergence rate for relatively strongly convex functions, decreasing learning rate). *Let h be a steep regularizer. If f is μ-strongly convex relative to h with minimum $x^* \in \mathcal{X}$ and if $\eta(t) = (1+t)^{-1/p}$, then*

$$\mathbb{E}[\|X(t) - x^*\|^2] \leq \frac{\sigma_{\text{tame}}^2}{\mu K^2} t^{-\frac{1}{p}} + \frac{2}{\mu K}\left(\frac{\|\mathcal{X}\|^{2-p}\sigma_{\text{heavy}}^p}{K^{p-1}} + \frac{H}{p}\right) t^{-\frac{p-1}{p}} + \frac{2D(x^*,x_0)}{K} \exp\left[-\frac{\mu p}{p-1}\left(t^{\frac{p-1}{p}} - 1\right)\right]. \tag{D.80}$$

*In particular, $X(t) \to x^*$ in mean square as $t \to \infty$.*

*Proof.* Following the same arguments as in the proof of Theorem D.6 but with a non-constant learning rate, we obtain that the function $\psi(t) := \frac{1}{\eta(t)}\mathbb{E}[D(x^*, X(t))]$ is almost-everywhere differentiable and that it satisfies

$$\psi'(t) \leq -\mu\,\mathbb{E}[D(x^*, X)] + c_1\eta(t) + c_2\eta(t)^{p-1} - \frac{\dot{\eta}(t)}{\eta^2(t)}\mathbb{E}[h(x^*) - h(X)]$$

$$\leq -\mu\eta(t)\psi(t) + c_1\eta(t) + c_2\eta(t)^{p-1} - \frac{\dot{\eta}(t)}{\eta^2(t)}H, \tag{D.81}$$

where we recall that $c_1 = \frac{\sigma_{\text{tame}}^2}{2K}$ and $c_2 = \frac{\|\mathcal{X}\|^{2-p}}{K^{p-1}}\sigma_{\text{heavy}}^p$. Letting $\eta(t) = (1+t)^{-1/p}$, we therefore get

$$\psi'(t) \leq -\frac{\mu}{(1+t)^{\frac{1}{p}}}\psi(t) + \frac{c_1}{(1+t)^{\frac{1}{p}}} + \frac{c_2}{(1+t)^{\frac{p-1}{p}}} + \frac{H}{p(1+t)^{\frac{p-1}{p}}}, \tag{D.82}$$

and Grönwall's lemma can then be invoked to derive

$$\psi(t) \leq \psi(0)\exp\left(-\mu\int_0^t \frac{1}{(1+s)^{\frac{1}{p}}}ds\right) + \int_0^t\left(\frac{c_1}{(1+s)^{\frac{1}{p}}} + \frac{c_3}{(1+s)^{\frac{p-1}{p}}}\right)\exp\left(-\mu\int_s^t \frac{1}{(1+r)^{\frac{1}{p}}}dr\right)ds, \tag{D.83}$$

where we have set $c_3 = c_2 + \frac{H}{p}$. To estimate the integral appearing in the right-hand side of (D.83), we need the following intermediate lemma:

**Lemma D.9.** *For every constants $a, b \geq 0$, $c > 0$, and all exponents $p \in (1, 2]$,*

$$I(t) := \int_0^t\left(\frac{a}{(1+s)^{\frac{1}{p}}} + \frac{b}{(1+s)^{\frac{p-1}{p}}}\right)\exp\left(-c\int_s^t \frac{1}{(1+r)^{\frac{1}{p}}}dr\right)ds \leq \frac{1}{c}\left(a + b(1+t)^{\frac{2-p}{p}}\right). \tag{D.84}$$

*Proof.* To begin with, note that

$$\int_s^t \frac{1}{(1+r)^{1/p}}dr = \frac{p}{p-1}\left[(1+t)^{\frac{p-1}{p}} - (1+s)^{\frac{p-1}{p}}\right], \tag{D.85}$$

so $I(t)$ can be written as

$$I(t) = \exp\left(-\frac{cp}{p-1}(1+t)^{\frac{p-1}{p}}\right)\int_0^t\left(a(1+s)^{-\frac{1}{p}} + b(1+s)^{-\frac{p-1}{p}}\right)\exp\left(\frac{cp}{p-1}(1+s)^{\frac{p-1}{p}}\right)ds. \tag{D.86}$$

Using the substitution $r \leftrightarrow (1+s)^{\frac{p-1}{p}}$, we further get

$$I(t) = \exp\left(-\frac{cp}{p-1}(1+t)^{\frac{p-1}{p}}\right)\frac{p}{p-1}\int_1^{(1+t)^{\frac{p-1}{p}}}\left(a + br^{\frac{2-p}{p-1}}\right)\exp\left(\frac{cp}{p-1}r\right)dr. \tag{D.87}$$

Now, since $\frac{2-p}{p-1} \geq 0$ and $0 \leq r \leq (1+t)^{\frac{p-1}{p}}$, we also have $r^{\frac{2-p}{p-1}} \leq (1+t)^{\frac{2-p}{p}}$ and thus

$$
\begin{aligned}
I(t) &\leq \exp\left(-\frac{cp}{p-1}(1+t)^{\frac{p-1}{p}}\right) \frac{p}{p-1}\left(a + b(1+t)^{\frac{2-p}{p}}\right) \int_1^{(1+t)^{\frac{p-1}{p}}} \frac{p}{p-1} \exp\left(\frac{cp}{p-1}r\right) dr \\
&= \frac{1}{c}\left(a + b(1+t)^{\frac{2-p}{p}}\right)\left[1 - \exp\left(\frac{cp}{p-1}\left(1 - (1+t)^{\frac{p-1}{p}}\right)\right)\right] \\
&\leq \frac{1}{c}\left(a + b(1+t)^{\frac{2-p}{p}}\right),
\end{aligned}
\tag{D.88}
$$

which concludes the proof. $\blacksquare$

*Proof of Theorem D.7 (cont'd).* Using Lemma D.9 in Eq. (D.83) and replacing the function $\psi(t)$ by its expression, we readily get

$$
\mathbb{E}\left[(1+t)^{1/p} D(x^*, X(t))\right] \leq D(x^*, x_0) \exp\left(-\frac{\mu p}{p-1}\left((1+t)^{\frac{p-1}{p}} - 1\right)\right) + \frac{1}{\mu}\left(c_1 + c_3(1+t)^{\frac{2-p}{p}}\right)
\tag{D.89}
$$

which, after dividing both sides by $(1+t)^{1/p}$ and using using the inequality $\|X(t) - x^*\|^2 \leq \frac{2}{K} D(x^*, X(t))$, yields

$$
\mathbb{E}\left[\|X(t) - x^*\|^2\right] \leq \frac{2c_1}{K\mu}(1+t)^{-\frac{1}{p}} + \frac{2c_3}{K\mu}(1+t)^{-\frac{p-1}{p}} + \frac{2D(x^*, x_0)}{K} \exp\left(-\frac{\mu p}{p-1}\left((1+t)^{\frac{p-1}{p}} - 1\right)\right).
\tag{D.90}
$$

The final result is then obtained when replacing $c_1$ and $c_3$ with their respective values. $\blacksquare$

*Remark* D.10. For a learning rate $\eta \propto t^{-1/p}$, Theorem D.7 implies that for any $\varepsilon > 0$, the accuracy $\mathbb{E}\left[\|X(t) - x^*\|^2\right] < \varepsilon$ is reached for times $t \geq \mathcal{O}\left(\varepsilon^{-\frac{p}{p-1}}\right)$. Unlike the complexity bound for constant learning rates, this rate is of the same order as those for convex or strongly convex functions, and therefore does not even attain the optimal $\mathcal{O}(1/\varepsilon)$ complexity for $p = 2$. As we discuss further in the discrete-time section, this limitation arises because a positive term proportional to $-\dot{\eta}/\eta^2$ always appears in the computations. This is a consequence of the learning rate acting on the aggregated gradient steps rather than on individual steps, which prevents the use of a $1/t$-type learning rate to achieve sharper convergence rates, as is standard in the analysis of stochastic gradient methods.

# E  Results in discrete time: Omitted proofs from Section 4

In this section, we draw inspiration from the continuous-time analysis of (LMF) to study the convergence properties of the Stochastic Dual Averaging (SDA) algorithm given by

$$
\begin{aligned}
y_{t+1} &= y_t - g_t \\
x_{t+1} &= Q(\eta_{t+1} y_{t+1})
\end{aligned}
\tag{SDA}
$$

where $\eta_t > 0$ is a nonincreasing *learning rate* sequence and $g_t$ is a *stochastic gradient* of the form $g_t = \nabla f(x_t) + U_t$ with gradient noise term $U_t = \mathsf{U}(x_t; \omega_t)$ satisfying the following assumptions:

(a) *Unbiasedness:* $\mathbb{E}[\mathsf{U}(x; \omega)] = 0$ for all $x \in \operatorname{dom} \partial f$.

(b) *Bounded $p$-th central moments:* $\mathbb{E}[\|\mathsf{U}(x; \omega)\|_*^p] \leq \sigma^p$ for some $p \in (1, 2]$ and all $x \in \operatorname{dom} \partial f$.

Throughout this section, we further assume that $f$ is *L-smooth relative to $h$*, that is,

$$
f(x') \leq f(x) + \langle \nabla h(x), x' - x \rangle + LD(x', x) \quad \text{for all } x \in \mathcal{X}_h, x' \in \mathcal{X}..
\tag{E.1}
$$

**E.1.  Convex setting.**  We begin by exploring the case where $f$ is a convex function. As in the continuous-time analysis, our approach is based on tracking the evolution of the *$\eta$-deflated Fenchel coupling*

$$
E_t(q, y) = \frac{1}{\eta_t} F(q, \eta_t y) \quad q \in \mathcal{X}, y \in \mathcal{Y}
\tag{E.2}
$$

along the iterates of (SDA). In contrast to (LMF), however, the discrete-time nature of (SDA) allows us to derive the evolution of $E_t$ without needing a weak stochastic chain rule (Itô's formula), which considerably simplifies the analysis. This leads to the following result, which can be viewed as a discrete-time analogue of the classical *three-point identity* (Lemma B.2) for the $\eta$-deflated Fenchel coupling.

**Lemma E.1** (Three-point identity for $\eta$-deflated Fenchel coupling). *For every $q \in \mathcal{X}$, the process $E_t := E_t(q, y_t)$ along trajectories of* (SDA) *satisfies*

$$E_{t+1} - E_t = \left(\frac{1}{\eta_{t+1}} - \frac{1}{\eta_t}\right)(h(q) - h(x_{t+1})) + \langle x_t - q, y_{t+1} - y_t \rangle + \langle x_{t+1} - x_t, y_{t+1} - y_t \rangle - \frac{1}{\eta_t}F(x_{t+1}, \eta_t y_t). \tag{E.3}$$

*Proof.* First, note that
$$F(x_{t+1}, \eta_t y_t) - F(q, \eta_t y_t) = h(x_{t+1}) - h(q) - \eta_t \langle x_{t+1} - q, y_t \rangle, \tag{E.4}$$

and, by optimality of $x_{t+1}$, we also have

$$\begin{aligned}
F(q, \eta_{t+1} y_{t+1}) &= h(q) + h^*(\eta_{t+1} y_{t+1}) - \eta_{t+1} \langle q, y_{t+1} \rangle \\
&= h(q) + \eta_{t+1} \langle x_{t+1}, y_{t+1} \rangle - h(x_{t+1}) - \eta_{t+1} \langle q, y_{t+1} \rangle \\
&= h(q) - h(x_{t+1}) + \eta_{t+1} \langle x_{t+1} - q, y_{t+1} \rangle.
\end{aligned} \tag{E.5}$$

This leads to

$$\begin{aligned}
E_{t+1} - E_t + \frac{1}{\eta_t}F(x_{t+1}, \eta_t y_t) &= \frac{1}{\eta_{t+1}}F(q, \eta_{t+1} y_{t+1}) + \frac{1}{\eta_t}(F(x_{t+1}, \eta_t y_t) - F(q, \eta_t y_t)) \\
&= \left(\frac{1}{\eta_{t+1}} - \frac{1}{\eta_t}\right)(h(q) - h(x_{t+1})) + \langle x_{t+1} - q, y_{t+1} - y_t \rangle \\
&= \left(\frac{1}{\eta_{t+1}} - \frac{1}{\eta_t}\right)(h(q) - h(x_{t+1})) + \langle x_t - q, y_{t+1} - y_t \rangle + \langle x_{t+1} - x_t, y_{t+1} - y_t \rangle, \tag{E.6}
\end{aligned}$$

which concludes the proof after rearranging the terms. ∎

We now invoke the relative smoothness of the objective together with the bounded $p$-th central moment assumption on the gradient noise to obtain the following estimate:

**Lemma E.2** (Heavy-tailed noise estimate). *If $f$ is relatively $L$-smooth and $\eta_t \leq \frac{1}{pL}$, then the iterates of* (SDA) *satisfy*

$$\langle x_{t+1} - x_t, y_{t+1} - y_t \rangle - \frac{1}{\eta_t}F(x_{t+1}, \eta_t y_t) \leq f(x_t) - f(x_{t+1}) + \frac{2^{p-1}\|\mathcal{X}\|^{2-p}}{pK^{p-1}}\eta_t^{p-1}\|U_t\|_*^p. \tag{E.7}$$

*Proof.* Since $y_{t+1} - y_t = -g_t = -(\nabla f(x_t) + U_t)$ and $p \in (1, 2]$, we can decompose the left-hand side of Eq. (E.7) as

$$\langle x_{t+1} - x_t, y_{t+1} - y_t \rangle - \frac{1}{\eta_t}F(x_{t+1}, \eta_t y_t) = \langle \nabla f(x_t), x_t - x_{t+1} \rangle - \frac{1}{p\eta_t}F(x_{t+1}, \eta_t y_t) \tag{E.8}$$

$$+ \langle U_t, x_t - x_{t+1} \rangle - \frac{p-1}{p\eta_t}F(x_{t+1}, \eta_t y_t). \tag{E.9}$$

Focusing first on bounding (E.8), the relative smoothness of $f$ yields

$$\begin{aligned}
(E.8) &\leq f(x_t) - f(x_{t+1}) + LD(x_{t+1}, x_t) - \frac{1}{p\eta_t}F(x_{t+1}, \eta_t y_t) \\
&\leq f(x_t) - f(x_{t+1}) + LF(x_{t+1}, \eta_t y_t) - \frac{1}{p\eta_t}F(x_{t+1}, \eta_t y_t) \\
&= f(x_t) - f(x_{t+1}) - \frac{L}{\eta_t}\left(\frac{1}{pL} - \eta_t\right)F(x_{t+1}, \eta_t y_t)
\end{aligned}$$

$$\leq f(x_t) - f(x_{t+1}), \tag{E.10}$$

where the second inequality comes from $D(x, Q(y')) \leq F(x, y')$ (cf. Proposition B.2) and the last inequality from the assumption that $\eta_t \leq 1/(pL)$. On the other hand, for (E.9), we use the Cauchy-Schwarz inequality and the fact that $F(x, y) \geq \frac{K}{2}\|x - Q(y)\|^2$ (see Proposition B.2) to get

$$\text{(E.9)} \leq \|U_t\|_* \|x_{t+1} - x_t\| - \frac{p-1}{p\eta_t}\frac{K}{2}\|x_{t+1} - x_t\|^2. \tag{E.11}$$

Since the diameter $\|\mathcal{X}\|$ of $\mathcal{X}$ is assumed to be finite, we know that $\|x_{t+1} - x_t\| \leq \|\mathcal{X}\|$ and therefore

$$\|x_{t+1} - x_t\|^2 = \|\mathcal{X}\|^{-\frac{2-p}{p-1}}\|\mathcal{X}\|^{\frac{2-p}{p-1}}\|x_{t+1} - x_t\|^2 \geq \|\mathcal{X}\|^{-\frac{2-p}{p-1}}\|x_{t+1} - x_t\|^{\frac{2-p}{p-1}}\|x_{t+1} - x_t\|^2 = \|\mathcal{X}\|^{-\frac{2-p}{p-1}}\|x_{t+1} - x_t\|^{\frac{p}{p-1}}. \tag{E.12}$$

Putting this estimate back into Eq. (E.11) then yields

$$\text{(E.9)} \leq \|U_t\|_* \|x_{t+1} - x_t\| - \frac{p-1}{p}\frac{K\|\mathcal{X}\|^{-\frac{2-p}{p-1}}}{2\eta_t}\|x_{t+1} - x_t\|^{\frac{p}{p-1}} \leq \max_{w \geq 0}\left(\|U_t\|_* w - \frac{p-1}{p}Cw^{\frac{p}{p-1}}\right), \tag{E.13}$$

where we have defined $C = \frac{K}{2\eta_t}\|\mathcal{X}\|^{-\frac{2-p}{p-1}}$ for conciseness. The maximization problem is achieved by $w = (\|U_t\|_*/C)^{p-1}$ and thus

$$\text{(E.9)} \leq \|U_t\|_*\left(\frac{\|U_t\|_*}{C}\right)^{p-1} - \frac{p-1}{p}C\left(\frac{\|U_t\|_*}{C}\right)^{\frac{p}{p-1}(p-1)} = \left(1 - \frac{p-1}{p}\right)C^{1-p}\|U_t\|_*^p = \frac{2^{p-1}\|\mathcal{X}\|^{2-p}}{pK^{p-1}}\eta_t^{p-1}\|U_t\|_*^p. \tag{E.14}$$

Combining the upper bounds for (E.8) and (E.9) therefore yields the desired result. ∎

We can now combine the first two lemmas to obtain the following theorem, which—similarly to its continuous-time analogue—serves as the baseline for most of our subsequent results.

**Theorem E.1.** *If $f$ is relatively $L$-smooth, then for every $q \in \mathcal{X}$ and every nonincreasing learning rate sequence with $\eta_1 \leq \frac{1}{pL}$, the iterates of (SDA) satisfy*

$$\mathbb{E}\left[\sum_{t=1}^{T}\langle x_t - q, \nabla f(x_t)\rangle\right] \leq \langle x_1 - q, y_1\rangle + f(x_1) - \min f + \frac{H}{\eta_{T+1}} + \frac{2^{p-1}\|\mathcal{X}\|^{2-p}}{pK^{p-1}}\sigma^p\sum_{t=1}^{T}\eta_t^{p-1}. \tag{E.15}$$

*Proof.* Summing the equality of Lemma E.1 over $t = 1, \dots, T$ and using the upper bound of Lemma E.2, we get

$$0 \leq \mathbb{E}[E_T] \leq E_1 + \sum_{t=1}^{T}\left(\frac{1}{\eta_{t+1}} - \frac{1}{\eta_t}\right)\mathbb{E}[h(q) - h(x_{t+1})] - \mathbb{E}\left[\sum_{t=1}^{T}\langle x_t - q, \nabla f(x_t)\rangle\right] + f(x_1) - \mathbb{E}[f(x_{T+1})]$$

$$- \sum_{t=1}^{T}\mathbb{E}[\langle x_t - q, U_t\rangle] + \frac{2^{p-1}\|\mathcal{X}\|^{2-p}}{pK^{p-1}}\sum_{t=1}^{T}\eta_t^{p-1}\mathbb{E}[\|U_t\|_*^p]. \tag{E.16}$$

Note that since $\eta_t$ is a nonincreasing sequence and since $h$ is continuous on the compact set $\mathcal{X}$,

$$\sum_{t=1}^{T}\left(\frac{1}{\eta_{t+1}} - \frac{1}{\eta_t}\right)\mathbb{E}[h(q) - h(x_{t+1})] \leq \sum_{t=1}^{T}\left(\frac{1}{\eta_{t+1}} - \frac{1}{\eta_t}\right)H = \frac{H}{\eta_{T+1}} - \frac{H}{\eta_1}. \tag{E.17}$$

Moreover, by optimality of $x_1$,

$$E_1 = \frac{1}{\eta_1}F(q, \eta_1 y_1) = \frac{1}{\eta_1}(h(q) - h(x_1) - \langle \eta_1 y_1, q - x_1\rangle) \leq \frac{H}{\eta_1} - \langle q - x_1, y_1\rangle. \tag{E.18}$$

Now, by the blanket assumptions made on the noise term $U_t$ and by definition of the filtration $\mathcal{F}_t$, we have

$$\mathbb{E}[\|U_t\|_*^p] = \mathbb{E}[\mathbb{E}[\|U(x_t; \omega_t)\|_*^p \mid \mathcal{F}_t]] \leq \sigma^p \tag{E.19}$$

and

$$\mathbb{E}[\langle x_t - q, U_t\rangle] = \mathbb{E}[\mathbb{E}[\langle x_t - q, \mathsf{U}(x_t;\omega_t)\rangle \mid \mathcal{F}_t]] = \mathbb{E}[\langle x_t - q, \mathbb{E}[\mathsf{U}(x_t;\omega_t) \mid \mathcal{F}_t]\rangle] = 0. \qquad (\text{E.20})$$

All of these estimates finally lead to

$$\mathbb{E}\left[\sum_{t=1}^{T}\langle x_t - q, \nabla f(x_t)\rangle\right] \le \langle x_1 - q, y_1\rangle + f(x_1) - \min f + \frac{H}{\eta_{T+1}} + \frac{2^{p-1}\|\mathcal{X}\|^{2-p}}{pK^{p-1}}\sigma^p \sum_{t=1}^{T}\eta_t^{p-1}, \qquad (\text{E.21})$$

hence concluding the proof. ∎

Specializing Theorem E.9 to convex objectives yields the following result, which generalizes Theorem 5 from the main text to arbitrary learning rates and arbitrary initial conditions.

**Theorem E.2** (Convergence rate of (SDA) for convex functions). *If $f$ is convex and relatively $L$-smooth, then for every nonincreasing stepsize sequence with $\gamma_1 \le \frac{1}{pL}$ and for every $x^* \in \arg\min f$, the iterates of (SDA) satisfy*

$$\mathbb{E}[f(\bar{x}_T) - \min f] \le \frac{\langle x_1 - x^*, y_1\rangle + f(x_1) - \min f}{T} + \frac{H}{T\eta_{T+1}} + \frac{2^{p-1}\|\mathcal{X}\|^{2-p}\sigma^p}{pK^{p-1}}\frac{\sum_{t=1}^{T}\eta_t^{p-1}}{T}, \qquad (\text{E.22})$$

*where $\bar{x}_T = \frac{1}{T}\sum_{t=1}^{T} x_t$. In particular, the choice $\eta_t = \beta t^{-1/p}$ with $\beta \le \frac{1}{pL}$ yields*

$$\mathbb{E}[f(\bar{x}_T) - \min f] \le \frac{\langle x_1 - x^*, y_1\rangle + f(x_1) - \min f}{T} + \left(\frac{H}{\beta} + \frac{2^{p-1}\beta^{p-1}\|\mathcal{X}\|^{2-p}\sigma^p}{K^{p-1}}\right)T^{-\frac{p-1}{p}}. \qquad (\text{E.23})$$

*Proof.* Due to $f$ being convex,

$$\sum_{t=1}^{T}\langle x_t - x^*, \nabla f(x_t)\rangle \ge \sum_{t=1}^{T}[f(x_t) - f(x^*)] \ge T[f(\bar{x}(t)) - \min f], \qquad (\text{E.24})$$

and the convergence rate for general learning rates is therefore an immediate consequence of Theorem E.1. The specialized bound for $\eta_t = \beta t^{-1/p}$ is then established by using the upper bound $\sum_{t=1}^{T} t^{-\frac{p-1}{p}} \le pT^{1/p}$ obtained with a standard integral estimation. ∎

*Remark* E.1. For any precision $\varepsilon > 0$, running (SDA) with learning rate $\eta = \mathcal{O}\left((\varepsilon/\sigma^p)^{\frac{1}{p-1}}\right)$ yields $\mathbb{E}[f(\bar{x}_T) - \min f] \le \varepsilon$ for every $T \ge \mathcal{O}\left((\sigma/\varepsilon)^{\frac{p}{p-1}}\right)$, thus matching the complexity obtained in the continuous-time setting, cf. Remark 4.

**E.2. Strongly convex setting.** We now turn to the convergence analysis of (SDA) for strongly convex objectives. Leveraging the upper bound in Theorem E.9, we first obtain a convergence guarantee for the ergodic averages that holds for any strongly convex objective, independently of the choice of regularizer.

**Theorem E.3** (Convergence rate of (SDA) for strongly convex functions). *If $f$ is $\mu$-strongly convex with minimum $x^* \in \mathcal{X}$ and relatively $L$-smooth, then for every nonincreasing stepsize sequence with $\gamma_1 \le \frac{1}{pL}$, the iterates of (SDA) satisfy*

$$\mathbb{E}\left[\|\bar{x}_T - x^*\|^2\right] \le 2\frac{\langle x_1 - x^*, y_1\rangle + f(x_1) - \min f}{\mu T} + \frac{2H}{\mu T\eta_{T+1}} + \frac{2^p\|\mathcal{X}\|^{2-p}\sigma^p}{\mu pK^{p-1}}\frac{\sum_{t=1}^{T}\eta_t^{p-1}}{T}. \qquad (\text{E.25})$$

*In particular, the choice $\eta_t = \beta t^{-1/p}$ with $\beta \le \frac{1}{pL}$ yields*

$$\mathbb{E}\left[\|\bar{x}_T - x^*\|^2\right] \le 2\frac{\langle x_1 - x^*, y_1\rangle + f(x_1) - \min f}{\mu T} + \frac{2}{\mu}\left(\frac{H}{\beta} + \frac{2^{p-1}\beta^{p-1}\|\mathcal{X}\|^{2-p}\sigma^p}{K^{p-1}}\right)T^{-\frac{p-1}{p}}. \qquad (\text{E.26})$$

*Proof.* By strong convexity of $f$,

$$\sum_{t=1}^{T}\langle x_t - x^*, \nabla f(x_t)\rangle \ge \frac{\mu}{2}\sum_{t=1}^{T}\|x_t - x^*\|^2 \ge \frac{\mu T}{2}\|\bar{x}_T - x^*\|^2, \qquad (\text{E.27})$$

which directly yields the results by Theorem E.1. ∎

*Remark* E.2. Similarly to Remark E.1, we get that for every $\varepsilon > 0$, the learning rate $\eta = \mathcal{O}\left((\varepsilon/\sigma^p)^{\frac{1}{p}}\right)$ achieves the accuracy $\mathbb{E}\left[\|\bar{x}_T - x^*\|^2\right] \leq \varepsilon$ for all horizon $T \geq \mathcal{O}\left((\sigma/\varepsilon)^{\frac{p}{p-1}}\right)$.

We now proceed by estimating the fraction of time that (SDA) spends away from $\arg \min f$, and the time that it takes to get close to it for the first time. These results specialize to Theorems 7 and 8 from the main text when the algorithms are initialized to the prox-center $x_c = \arg \min h$ of $\mathcal{X}$.

**Theorem E.4** (Concentration around global minimum)**.** *Assume that $f$ is $\mu$-strongly convex with minimum $x^*$ and $L$-relatively smooth. Let*

$$\mu_T(\mathcal{U}_\delta) = \frac{1}{T} \sum_{t=1}^{T} \mathbb{1}\{x_t \in \mathcal{U}_\delta\} \tag{E.28}$$

*be the fraction of time that $x_t$ spends in $\mathcal{U}_\delta = \{x \in \mathcal{X} : \|x - x^*\| \leq \delta\}$ up to time $T$. If (SDA) is run with constant learning rate $\eta_t \equiv \eta$, we have*

$$\mathbb{E}[\mu_T(\mathcal{U}_\delta)] \geq 1 - \frac{2\langle x_1 - x^*, y_1\rangle}{\mu\delta T} - \frac{2(f(x_1) - \min f)}{\mu\delta^2 T} - \frac{2H}{\eta\mu\delta^2 T} - \frac{2^p \eta^{p-1} \|\mathcal{X}\|^{2-p}\sigma^p}{\mu p K^{p-1}\delta^2}. \tag{E.29}$$

*Proof.* Markov's inequality together with the strong convexity of $f$ yield

$$\mathbb{E}[\mu_T(\mathcal{X} \setminus \mathcal{U}_\delta)] = \frac{1}{T}\sum_{t=1}^{T} \mathbb{P}(\|x_t - x^*\| > \delta) \leq \frac{1}{T}\sum_{t=1}^{T} \frac{\mathbb{E}\left[\|x_t - x^*\|^2\right]}{\delta^2} \leq \frac{2}{\mu\delta^2 T} \mathbb{E}\left[\sum_{t=1}^{T}\langle x_t - x^*, \nabla f(x_t)\rangle\right], \tag{E.30}$$

and the concentration bound therefore follows directly from Theorem E.1. ∎

**Theorem E.5** (Hitting time estimate)**.** *Assume that $f$ is $\mu$-strongly convex with minimum $x^* \in \mathcal{X}$ and relatively $L$-smooth. Let*

$$\tau_\delta = \inf\{t \in \mathbb{N} : \|x_t - x^*\| \leq \delta\} \tag{E.31}$$

*denote the first time $x_t$ gets within $\delta$ of $x^*$ for some precision $\delta > 0$. If (SDA) is run with constant learning rate*

$$\eta < \left(\frac{\mu p K}{2^p K^{2-p}\|\mathcal{X}\|^{2-p}\sigma^p}\right)^{\frac{1}{p-1}} \delta^{\frac{2}{p-1}}, \tag{E.32}$$

*then the average value of $\tau_\delta$ is bounded as*

$$\mathbb{E}[\tau_\delta] \leq \frac{2pKF(x^*, \eta y_1) + 2pK\left(f(x_1) - \min f + \frac{\mu\delta^2}{2}\right)\eta}{\mu p K\delta^2\eta - 2^p K^{2-p}\|\mathcal{X}\|^{2-p}\sigma^p\eta^p}. \tag{E.33}$$

*In particular, running (SDA) with $y_1 = 0$ and*

$$\eta = \left(\frac{\mu K}{2^p K^{2-p}\|\mathcal{X}\|^{2-p}\sigma^p}\right)^{\frac{1}{p-1}} \delta^{\frac{2}{p-1}} \tag{E.34}$$

*yields the hitting-time estimate*

$$\mathbb{E}[\tau_\delta] \leq \frac{p}{p-1}\left\{\frac{2H}{\mu K}\left(\frac{2^p\|\mathcal{X}\|^{2-p}\sigma^p}{\mu}\right)^{\frac{1}{p-1}}\delta^{-\frac{2p}{p-1}} + \frac{2(f(x_1) - \min f)}{\mu}\delta^{-2} + 1\right\}. \tag{E.35}$$

*Remark* E.3. Since $y_1 = 0$, the quantity $f(x_1) - \min f$ in Eq. (E.35) is in fact equal to $f(\arg \min h) - f(x^*)$, that is, to the value difference between the *prox-center* $x_c = \arg \min h$ and the minimizer $x^*$ of $f$.

*Proof.* Since the learning rate $\eta$ is supposed constant, summing the equality of Lemma E.1 over $t = 1, \ldots, T$ and invoking the upper bound of Lemma E.2 yields

$$0 \leq E_T \leq E_1 - \sum_{t=1}^{T}\langle x_t - x^*, \nabla f(x_t)\rangle + f(x_1) - \min f - \sum_{t=1}^{T}\langle x_t - x^*, U_t\rangle + \frac{2^{p-1}\|\mathcal{X}\|^{2-p}\eta^{p-1}}{pK^{p-1}}\sum_{t=1}^{T}\|U_t\|_*^p. \tag{E.36}$$

By analogy with the arguments used to prove the hitting time estimate for continuous (LMF) dynamics (cf. Theorem D.5), we then apply the above inequality to the bounded stopping times $T \leftarrow \tau_\delta \wedge T$ and take the expectation on both sides, which leads to

$$0 \leq E_1 + f(x_1) - \min f - \mathbb{E}\left[\sum_{t=1}^{\tau_\delta \wedge T} \langle x_t - x^*, \nabla f(x_t) \rangle \right] + \frac{2^{p-1} \|\mathcal{X}\|^{2-p} \eta^{p-1}}{pK^{p-1}} \mathbb{E}\left[\sum_{t=1}^{\tau_\delta \wedge T} \|U_t\|_*^p \right] - \mathbb{E}\left[\sum_{t=1}^{\tau_\delta \wedge T} \langle x_t - x^*, U_t \rangle \right].$$

Before proceeding further, recall that $\mathcal{F}_t$ is the filtration generated by $x_s$, $1 \leq s \leq t$, and that $U_t$ is a martingale difference sequence. In particular, $U_t$ is $\mathcal{F}_{t+1}$-measurable and $\mathbb{E}[U_t \mid \mathcal{F}_t] = 0$. By the tower property of conditional expectations, we can therefore write

$$\mathbb{E}\left[\sum_{t=1}^{\tau_\delta \wedge T} \langle x_t - x^*, U_t \rangle \right] = \sum_{t=1}^{T} \mathbb{E}[\mathbb{1}\{\tau_\delta \geq t\} \langle x_t - x^*, U_t \rangle] = \sum_{t=1}^{T} \mathbb{E}[\mathbb{E}[\mathbb{1}\{\tau_\delta \geq t\} \langle x_t - x^*, U_t \rangle \mid \mathcal{F}_t]]$$

$$= \sum_{t=1}^{T} \mathbb{E}[\mathbb{1}\{\tau_\delta \geq t\} \langle x_t - x^*, \mathbb{E}[U_t \mid \mathcal{F}_t] \rangle]$$

$$= 0, \tag{E.37}$$

where we have used that both $x_t$ and $\mathbb{1}\{\tau_\delta \geq t\}$ are $\mathcal{F}_t$-measurable. Indeed, to prove the latter, one notes that

$$\{\tau_\delta \geq t\} = \bigcap_{s=1}^{t-1} \{\|x_s - x^*\| > \delta\} \in \mathcal{F}_t \tag{E.38}$$

since all events $\{\|x_s - x^*\| > \delta\}$, $1 \leq s \leq t - 1$, belong to $\mathcal{F}_t$. Similarly, the bounded $p$-th central moment condition satisfied by $U_t$ yields

$$\mathbb{E}\left[\sum_{t=1}^{\tau_\delta \wedge T} \|U_t\|_*^p \right] = \sum_{t=1}^{T} \mathbb{E}[\mathbb{1}\{\tau_\delta \geq t\} \|U\|_*^p] = \sum_{t=1}^{T} \mathbb{E}[\mathbb{1}\{\tau_\delta \geq t\} \mathbb{E}[\|U\|_*^p \mid \mathcal{F}_t]] \leq \sigma^p \mathbb{E}\left[\sum_{t=1}^{\tau_\delta \wedge T} 1 \right] = \sigma^p \mathbb{E}[\tau_\delta \wedge T]. \tag{E.39}$$

Moreover, by strong convexity of $f$ and definition of $\tau_\delta$, we also get

$$\sum_{t=1}^{\tau_\delta \wedge T} \langle x_t - x^*, \nabla f(x_t) \rangle \geq \frac{\mu}{2} \sum_{t=1}^{\tau_\delta \wedge T} \|x_t - x^*\|^2 \geq \frac{\mu}{2} \sum_{t=1}^{\tau_\delta \wedge T-1} \|x_t - x^*\|^2 \geq \frac{\mu}{2} \sum_{t=1}^{\tau_\delta \wedge T-1} \delta^2 = \frac{\mu \delta^2}{2}(\tau_\delta \wedge T - 1). \tag{E.40}$$

The previous computations lead to

$$0 \leq E_1 + f(x_1) - \min f - \frac{\mu \delta^2}{2}(\mathbb{E}[\tau_\delta \wedge T] - 1) + \frac{2^{p-1} \|\mathcal{X}\|^{2-p} \eta^{p-1} \sigma^p}{pK^{p-1}} \mathbb{E}[\tau_\delta \wedge T] \tag{E.41}$$

$$= E_1 + f(x_1) - \min f + \frac{\mu \delta^2}{2} - \left(\frac{\mu \delta^2}{2} - \frac{2^{p-1} \|\mathcal{X}\|^{2-p} \sigma^p \eta^{p-1}}{pK^{p-1}}\right) \mathbb{E}[\tau_\delta \wedge T], \tag{E.42}$$

where the multiplicative coefficient in front of $\mathbb{E}[\tau_\delta \wedge T]$ is (strictly) positive whenever the learning rate $\eta$ satisfies

$$\eta < \left(\frac{\mu pK}{2^p K^{2-p} \|\mathcal{X}\|^{2-p} \sigma^p}\right)^{\frac{1}{p-1}} \delta^{\frac{2}{p-1}}. \tag{E.43}$$

Rearranging the terms therefore yields the upper bound

$$\mathbb{E}[\tau_\delta \wedge T] \leq \frac{E_1 + f(x_1) - \min f + \frac{\mu \delta^2}{2}}{\frac{\mu \delta^2}{2} - \frac{2^{p-1} \|\mathcal{X}\|^{2-p} \sigma^p \eta^{p-1}}{pK^{p-1}}} = \frac{2pKF(x^*, \eta y_1) + 2pK\left(f(x_1) - \min f + \frac{\mu \delta^2}{2}\right)\eta}{\mu pK\delta^2 \eta - 2^p K^{2-p} \|\mathcal{X}\|^{2-p} \sigma^p \eta^p}, \tag{E.44}$$

which proves Eq. (E.33) when taking the monotone limit as $T \to \infty$. Now let us assume $y_1 = 0$ so that $F(x^*, \eta y_1) = h(x^*) - h(x_1) \leq H$, and choose

$$\eta = \left(\frac{\mu K}{2^p K^{2-p} \|\mathcal{X}\|^{2-p} \sigma^p}\right)^{\frac{1}{p-1}} \delta^{\frac{2}{p-1}}, \tag{E.45}$$

which is the learning rate maximizing the denominator of Eq. (E.33). For notational convenience, we also define

$$c_1 = 2pKH, \quad c_2 = 2pK\left(f(x_1) - \min f + \frac{\mu\delta^2}{2}\right), \quad c_3 = \mu pK\delta^2, \quad c_4 = 2^p K^{2-p} \|\mathcal{X}\|^{2-p}\sigma^p. \tag{E.46}$$

In particular $\eta = [c_3/(pc_4)]^{1/(p-1)}$ and we therefore obtain from Eq. (E.33) that

$$
\begin{aligned}
\mathbb{E}[\tau_\delta] &\le \frac{c_1 + c_2\eta}{c_3\eta + c_4\eta^p} = \left[c_3\left(\frac{c_3}{pc_4}\right)^{\frac{1}{p-1}} - c_4\left(\frac{c_3}{pc_4}\right)^{\frac{p}{p-1}}\right]^{-1}\left[c_1 + c_2\left(\frac{c_3}{pc_4}\right)^{\frac{1}{p-1}}\right] \\
&= \left[\left(\frac{c_3^p}{pc_4}\right)^{\frac{1}{p-1}}\left(1 - \frac{1}{p}\right)\right]^{-1}\left[c_1 + c_2\left(\frac{c_3}{pc_4}\right)^{\frac{1}{p-1}}\right] \\
&= \frac{p}{p-1}\left(\frac{pc_4}{c_3^p}\right)^{\frac{1}{p-1}}\left[c_1 + c_2\left(\frac{c_3}{pc_4}\right)^{\frac{1}{p-1}}\right] \\
&= \frac{p}{p-1}\left\{c_1\left(\frac{pc_4}{c_3^p}\right)^{\frac{1}{p-1}} + \frac{c_2}{c_3}\right\} \\
&= \frac{p}{p-1}\left\{2pKH\left(\frac{p2^p K^{2-p}\|\mathcal{X}\|^{2-p}\sigma^p}{\mu^p p^p K^p \delta^{2p}}\right)^{\frac{1}{p}} + \frac{2pK\left(f(x_1) - \min f + \frac{\mu\delta^2}{2}\right)}{\mu pK\delta^2}\right\} \\
&= \frac{p}{p-1}\left\{\frac{2H}{\mu K}\left(\frac{2^p\|\mathcal{X}\|^{2-p}\sigma^p}{\mu}\right)^{\frac{1}{p-1}}\delta^{-\frac{2p}{p-1}} + \frac{2\left(f(x_1) - \min f\right)}{\mu}\delta^{-2} + 1\right\}, \tag{E.47}
\end{aligned}
$$

hence concluding the proof. ∎

We next consider objectives whose convexity is adapted to the geometry induced by the regularizer, namely objectives that are *strongly convex relative to h*. Specifically, we assume that

$$f(x') \ge f(x) + \langle \nabla f(x), x' - x \rangle + \mu D(x', x) \tag{E.48}$$

for all $x \in \mathcal{X}_h$ and all $x' \in \mathcal{X}$.

We begin by establishing a convergence guarantee for constant learning rates, yielding a discrete-time analogue of Theorem D.6. The result also slightly refines Theorem 6 from the main text, as it no longer requires (SDA) to be initialized at the prox-center of $\mathcal{X}$.

**Theorem E.6** (Convergence rate of (SDA) for relatively strongly convex functions, constant learning rate). *Let h be a steep regularizer. If f is relatively L-smooth and relatively $\mu$-strongly convex with minimum $x^* \in \mathcal{X}$, then running (SDA) with a constant learning rate $\eta \le \frac{1}{pL}$ achieves*

$$\mathbb{E}\left[\|x_t - x^*\|^2\right] \le \frac{2^{p-1}\|\mathcal{X}\|^{2-p}}{p\mu K^p}\sigma^p\eta^{p-1} + \frac{2D(x^*, x_1)}{K}(1 - \eta\mu)^{t-1}. \tag{E.49}$$

*Proof.* Starting from Lemmas E.1 and E.2 with $q = x^*$ and then using the relative strong convexity of $f$, we readily get

$$\mathbb{E}[E_{t+1}] \le \mathbb{E}[E_t] + \mathbb{E}[\langle x^* - x_t, \nabla f(x_t)\rangle] + \mathbb{E}[f(x_t) - f(x_{t+1})] + \frac{2^{p-1}\|\mathcal{X}\|^{2-p}}{pK^{p-1}}\eta^{p-1}\sigma^p$$

$$\le \mathbb{E}[E_t] + \mathbb{E}[f(x^*) - f(x_t)] - \mu\mathbb{E}[D(x^*, x_t)] + \mathbb{E}[f(x_t) - f(x_{t+1})] + \frac{2^{p-1}\|\mathcal{X}\|^{2-p}}{pK^{p-1}}\eta^{p-1}\sigma^p. \tag{E.50}$$

But since $h$ is steep, we have $E_t = F(x^*, \eta y_t) = D(x^*, x_t)$ for all times $t \geq 1$, and thus

$$\mathbb{E}[D(x^*, x_{t+1})] \leq (1 - \eta\mu)\,\mathbb{E}[D(x^*, x_t)] + \eta\,\mathbb{E}[f(x^*) - f(x_{t+1})] + \frac{2^{p-1}\|\mathcal{X}\|^{2-p}}{pK^{p-1}}\eta^p\sigma^p$$

$$\leq (1 - \eta\mu)\,\mathbb{E}[D(x^*, x_t)] + \frac{2^{p-1}\|\mathcal{X}\|^{2-p}}{pK^{p-1}}\eta^p\sigma^p, \tag{E.51}$$

where we have used in the second inequality that $x^*$ is the minimum of $f$. Note in particular that $\eta\mu \leq 1$ due to the assumption that $\eta \leq 1/(pL)$ and the fact that $\mu \leq L$ (cf. [64, Proposition 1.1] ). By recurrence, we therefore get

$$\mathbb{E}[D(x^*, x_t)] \leq (1 - \eta\mu)^{t-1} D(x^*, x_1) + \frac{2^{p-1}\|\mathcal{X}\|^{2-p}}{p\mu K^{p-1}}\eta^{p-1}\sigma^p, \tag{E.52}$$

and it only remains to use the lower bound $\|x - x'\|^2 \leq \frac{2}{K}D(x, x')$ of Proposition B.2 to obtain the claimed inequality. ∎

*Remark* E.4. Due to Theorem E.6, we note that the learning $\eta = \mathcal{O}\!\left((\varepsilon/\sigma^p)^{\frac{1}{p}}\right)$ for any accuracy $\varepsilon > 0$ achieves the last iterate estimate $\mathbb{E}\!\left[\|x_t - x^*\|^2\right] \leq \varepsilon$ for all times $t \geq \Omega\!\left(\sigma^{\frac{p}{p-1}}\varepsilon^{-\frac{1}{p-1}}\log(1/\varepsilon)\right)$. This complexity rate is better than the one obtained for convex or strongly convex functions, and, as mentioned in the continuous-time section, this also recovers the optimal rate known for first-order stochastic methods when $p = 2$.

Finally, when the learning rate sequence is proportional to $t^{-1/p}$, we leverage relative strong convexity together with a discrete-time Grönwall inequality to obtain the following convergence guarantee:

**Theorem E.7** (Convergence rate of (SDA) for relatively strongly convex functions, decreasing learning rate). *Let $h$ be a steep regularizer. If $f$ is relatively $L$-smooth and relative $\mu$-strongly convex with minimum $x^* \in \mathcal{X}$, then running (SDA) with learning rate $\eta_t = \beta t^{-1/p}$ where $\beta \leq \frac{1}{pL}$ achieves*

$$\mathbb{E}\!\left[\|x_t - x^*\|^2\right] \leq \frac{2}{\mu K p}\left(\frac{H}{\beta} + \frac{2^{p-1}\beta^{p-1}\|\mathcal{X}\|^{2-p}}{K^{p-1}}\sigma^p\right)t^{-\frac{p-1}{p}} + \frac{2D(x^*, x_1)}{K}t^{-\frac{1}{p}}\exp\!\left[-\frac{\mu\beta p}{p-1}\left(t^{\frac{p-1}{p}} - 1\right)\right]. \tag{E.53}$$

*In particular, $x_t \to x^*$ in mean square as $t \to 0$.*

*Proof.* Following the same idea as in the proof of Theorem E.6 but with learning rate $\eta_t = \beta t^{-1/p}$, we obtain

$$\mathbb{E}[E_{t+1}] \leq \mathbb{E}[E_t] + \left(\frac{1}{\eta_{t+1}} - \frac{1}{\eta_t}\right)H - \mu\,\mathbb{E}[D(x^*, x_t)] + \frac{2^{p-1}\|\mathcal{X}\|^{2-p}}{pK^{p-1}}\eta^{p-1}\sigma^p$$

$$= (1 - \mu\eta_t)\,\mathbb{E}[E_t] + \left((t+1)^{1/p} - t^{1/p}\right)\frac{H}{\beta} + \frac{2^{p-1}\|\mathcal{X}\|^{2-p}}{pK^{p-1}}\eta^{p-1}\sigma^p. \tag{E.54}$$

Let $\psi(x) = x^{1/p}$. Then, by the mean value theorem, there exists $\xi \in (t, t+1)$ such that

$$(t+1)^{1/p} - t^{1/p} = \psi(t+1) - \psi(t) \leq \psi'(\xi) = \frac{1}{p}\xi^{-\frac{p-1}{p}} \leq \frac{1}{p}t^{-\frac{p-1}{p}} = \frac{1}{p\beta^{p-1}}\eta_t^{p-1} \tag{E.55}$$

This yields

$$\mathbb{E}[E_{t+1}] \leq (1 - \mu\eta_t)\,\mathbb{E}[E_t] + \frac{1}{p}\left(\frac{H}{\beta^p} + \frac{2^{p-1}\|\mathcal{X}\|^{2-p}\sigma^p}{K^{p-1}}\right)\eta_t^{p-1}. \tag{E.56}$$

Letting $C$ be the constant in front of $\eta_t^{p-1}$ and iterating the inequality for all $t = 1, \ldots, T-1$ then leads to

$$\mathbb{E}[E_T] \leq E_1 \prod_{t=1}^{T-1}(1 - \mu\eta_t) + C\sum_{t=1}^{T-1}\eta_t^{p-1}\prod_{s=t+1}^{T-1}(1 - \mu\eta_s) =: E_1 S_{1,T} + C S_{2,T}. \tag{E.57}$$

To estimate $S_{1,T}$, we use the classic inequality $1 - x \leq e^{-x}$ and an integral bound to obtain

$$S_{1,T} \leq \prod_{t=1}^{T-1} e^{-\mu\eta_t} = \exp\!\left(-\mu\sum_{t=1}^{T-1}\right) = \exp\!\left(-\beta\mu\sum_{t=1}^{T-1}t^{-1/p}\right) \leq \exp\!\left(-\beta\mu\int_1^T t^{-1/p}\,dt\right) = \exp\!\left(-\frac{\mu\beta p}{p-1}\left(T^{\frac{p-1}{p}} - 1\right)\right). \tag{E.58}$$

On the other hand, note that $\eta_t^{p-1} = \eta_t^{p-2}\eta_t \le \eta_T^{p-2}\eta_t$ for every $t = 1, \ldots, T-1$ due to $p \le 2$ and the fact that $\eta_t$ is a nonincreasing sequence. This allows us to write

$$
\begin{aligned}
S_{2,T} \le \eta_T^{p-2} \sum_{t=1}^{T-1} \eta_t \prod_{s=t+1}^{T-1} (1 - \mu\eta_s) &= \frac{\eta_T^{p-2}}{\mu} \sum_{t=1}^{T-1} (1 + \mu\eta_t - 1) \prod_{s=t+1}^{T-1} (1 - \mu\eta_s) \\
&= \frac{\eta_T^{p-2}}{\mu} \sum_{t=1}^{T-1} \left( \prod_{s=t+1}^{T-1} (1 - \mu\eta_s) - \prod_{s=t}^{T} (1 - \mu\eta_s) \right) \\
&= \frac{\eta_T^{p-2}}{\mu} \left( 1 - \prod_{s=1}^{T-1} (1 - \mu\eta_s) \right) \\
&\le \frac{\eta_T^{p-2}}{\mu}.
\end{aligned}
\tag{E.59}
$$

Injecting the bounds for $S_{1,T}$ and $S_{2,T}$ back into Eq. (E.57) and replacing $E_T$ by its expression, it follows that

$$
\mathbb{E}[D(x^*, X_t)] = \eta_t \, \mathbb{E}[E_t] \le \frac{D(x^*, X_0)}{t^{1/p}} \exp\left( -\frac{\mu\beta p}{p-1} \left( t^{\frac{p-1}{p}} - 1 \right) \right) + \frac{1}{\mu p} \left( \frac{H}{\beta} + \frac{2^{p-1}\beta^{p-1}\|\mathcal{X}\|^{2-p}}{K^{p-1}} \sigma^p \right) t^{-\frac{p-1}{p}},
\tag{E.60}
$$

and the claim is therefore proved by using the inequality $\|x^* - X_t\|^2 \le \frac{2}{K} D(x^*, X_t)$. ∎

*Remark* E.5. Analogous to its associated continuous-time result (Theorem D.7), Theorem E.7 only implies that (SDA) achieves a complexity rate of order $t = \Omega\left( \varepsilon^{-\frac{p}{p-1}} \right)$ for learning rates $\eta(t) \propto t^{-1/p}$, which thus does not reach the optimal lower bound even for the tame regime $p = 2$.

### E.3. Extension: Lazy Mirror Descent (LMD) under heavy-tailed noise.

In this section, we extend our previous results to the stochastic mirror descent with lazy updates, described by the system

$$
\begin{aligned}
y_{t+1} &= y_t - \gamma_t g_t \\
x_{t+1} &= Q(y_{t+1})
\end{aligned}
\tag{LMD}
$$

Following the discussion in the main text, the key difference between (LMD) and (SDA) is that, in (LMD), gradient steps are *pre-multiplied* by a stepsize $\gamma_t$, whereas in (SDA) they are multiplied by a learning rate $\eta_t$ *after* aggregation.

As will become clear in the proofs, the convergence results for (LMD) under heavy-tailed noise rely on the same arguments as those used for (SDA) with a constant learning rate. In particular, we no longer need to introduce the $\eta$-deflated Fenchel coupling and can instead directly analyze the evolution of the Fenchel coupling along the iterates of (LMD):

**Lemma E.3** (Three-point identity). *For every $q \in \mathcal{X}$, the iterates of* (LMD) *satisfy*

$$
F(q, y_{t+1}) - F(q, y_t) = \langle x_t - q, y_{t+1} - y_t \rangle + \langle x_{t+1} - x_t, y_{t+1} - y_t \rangle - F(x_{t+1}, y_t).
\tag{E.61}
$$

*Proof.* On one hand, we have

$$
F(x_{t+1}, y_t) - F(q, y_t) = h(x_{t+1}) - h(y) - \langle x_{t+1} - q, x_t \rangle,
\tag{E.62}
$$

and, on the other one, using the optimality of $x_{t+1}$,

$$
F(q, y_{t+1}) = h(q) + h^*(y_{t+1}) - \langle q, y_{t+1} \rangle = h(y) - h(x_{t+1}) + \langle x_{t+1} - q, y_{t+1} \rangle.
\tag{E.63}
$$

Combining both of these equalities therefore yield

$$
F(q, y_{t+1}) + F(x_{t+1}, y_t) - F(q, y_t) = \langle x_{t+1} - q, y_{t+1} - y_t \rangle = \langle x_t - q, y_{t+1} - y_t \rangle + \langle x_{t+1} - x_t, y_{t+1} - y_t \rangle,
\tag{E.64}
$$

hence proving the claim. ∎

Analogously to (SDA), leveraging the relative smoothness of the objective and the moment bounds on the gradient noise leads to the following essential estimate:

**Lemma E.4** (Heavy-tailed noise estimate). *If $f$ is relatively $L$-smooth and $\gamma_t \le \frac{1}{pL}$, then trajectories of (LMD) satisfy*

$$\langle x_{t+1} - x_t, y_{t+1} - y_t \rangle - F(x_{t+1}, x_t) \le \gamma_t [f(x_t) - f(x_{t+1})] + \frac{2^{p-1} \|\mathcal{X}\|^{2-p}}{pK^{p-1}} \gamma_t^p \|U_t\|_*^p. \qquad (E.65)$$

*Proof.* Since $y_{t+1} - y_t = -\gamma_t g_t = -\gamma_t (\nabla f(x_t) + U_t)$, we have

$$\frac{1}{\gamma_t}[\langle x_{t+1} - x_t, y_{t+1} - y_t \rangle - F(x_{t+1}, y_t)] = \langle \nabla f(x_t), x_t - x_{t+1} \rangle - \frac{1}{p\gamma_t} F(x_{t+1}, y_t)$$

$$+ \langle U_t, x_t - x_{t+1} \rangle - \frac{p-1}{p\gamma_t} F(x_{t+1}, y_t), \qquad (E.66)$$

and the rest of the proof can therefore be carried on by invoking the same computations as in Lemma E.2. ∎

From the two lemmas above, we obtain the following theorem, providing the analogue of Theorem E.9 for (LMD).

**Theorem E.8.** *If $f$ is relative $L$-smooth, then for every $q \in \mathcal{X}$ and every nonincreasing stepsize sequence with $\gamma_1 \le \frac{1}{pL}$, the iterates of (LMD) satisfy*

$$\mathbb{E}\left[\sum_{t=1}^T \gamma_t \langle x_t - q, \nabla f(x_t) \rangle \right] \le F(q, y_1) + \gamma_1 [f(x_1) - \min f] + \frac{2^{p-1} \|\mathcal{X}\|^{2-p} \sigma^p}{pK^{p-1}} \sum_{t=1}^T \gamma_t^p. \qquad (E.67)$$

*Proof.* After rearranging the terms, Lemmas E.3 and E.4 yield

$$\gamma_t \langle x_t - q, g_t \rangle \le F(q, y_t) - F(q, y_{t+1}) + \gamma_t [f(x_t - x_{t+1}] + \frac{2^{p-1} \|\mathcal{X}\|^{2-p}}{pK^{p-1}} \gamma_t^p \|U_t\|_*^p. \qquad (E.68)$$

Summing over $t = 1, \ldots, T$ and taking the overall expectation, we therefore get

$$\mathbb{E}\left[\sum_{t=1}^T \gamma_t \langle x_t - q, \nabla f(x_t) \rangle \right] \le F(q, y_1) - F(q, y_{t+1}) + \mathbb{E}\left[\sum_{t=1}^T \gamma_t [f(x_t) - f(x_{t+1})] \right] + \frac{2^{p-1} \|\mathcal{X}\|^{2-p} \sigma^p}{pK^{p-1}} \sum_{t=1}^T \gamma_t^p, \quad (E.69)$$

where we have used the blanket assumptions on $U_t$ as in the proof of Theorem E.1. Now, note that we can write

$$\sum_{t=1}^T \gamma_t [f(x_t) - f(x_{t+1})] = \sum_{t=0}^{T-1} \gamma_{t+1} f(x_{t+1}) - \sum_{t=1}^T \gamma_t f(x_{t+1}) = \gamma_1 f(x_1) - \gamma_T f(x_{T+1}) + \sum_{t=1}^{T-1} (\gamma_{t+1} - \gamma_t) f(x_{t+1}), \quad (E.70)$$

but $x_{t+1} - x_t \le 0$ by assumption and $f(x_{t+1}) \ge \min f$, so

$$\sum_{t=1}^T \gamma_t [f(x_t) - f(x_{t+1})] \le \gamma_1 f(x_1) - \gamma_T f(x_{T+1}) + \sum_{t=1}^{T-1} (\gamma_{t+1} - \gamma_t) \min f$$

$$= \gamma_1 [f(x_1) - \min f] + \gamma_T [\min f - f(x_{T+1})]$$

$$\le \gamma_1 [f(x_1) - \min f]. \qquad (E.71)$$

Moreover, $F(q, y_{t+1}) \ge 0$ by Proposition B.2, and thus

$$\mathbb{E}\left[\sum_{t=1}^T \gamma_t \langle x_t - q, \nabla f(x_t) \rangle \right] \le F(q, y_1) + \gamma_1 [f(x_1) - \min f] + \frac{2^{p-1} \|\mathcal{X}\|^{2-p} \sigma^p}{pK^{p-1}} \sum_{t=1}^T \gamma_t^p, \qquad (E.72)$$

which concludes the proof. ∎

As a first consequence, we obtain a convergence guarantee of (LMD) for convex objectives that are smooth relative to $h$:

**Theorem E.9** (Convergence rate of (LMD) for convex functions). *If $f$ is convex and relative $L$-smooth, then for every nonincreasing stepsize sequence with $\gamma_1 \leq \frac{1}{pL}$, the iterates of* (LMD) *satisfy*

$$\mathbb{E}[f(\tilde{x}_T) - \min f] \leq \frac{F(\arg\min f, y_1) + \gamma_1[f(x_1) - \min f]}{\sum_{t=1}^T \gamma_t} + \frac{2^{p-1}\|\mathcal{X}\|^{2-p}\sigma^p}{pK^{p-1}} \frac{\sum_{t=1}^T \gamma_t^p}{\sum_{t=1}^T \gamma_t}, \qquad (\text{E.73})$$

*where $\tilde{x}_T = \sum_{t=1}^T \gamma_t x_t / \sum_{t^1}^T \gamma_t$. In particular, the choice $\gamma_t = \beta t^{-1/p}$ with $\beta \leq \frac{1}{pL}$ yields*

$$\mathbb{E}[f(\tilde{x}_T) - \min f] \lesssim \left(\frac{1}{\beta} F(\arg\min f, y_1) + f(x_1) - \min f\right) T^{-\frac{p-1}{p}} + \frac{2^{p-1}\beta^{p-1}\|\mathcal{X}\|^{2-p}\sigma^p}{pK^{p-1}}(\log T)T^{-\frac{p-1}{p}}. \qquad (\text{E.74})$$

*Proof.* Let $q = x^* \in \arg\min f$ satisfying $F(x^*, y_1) = F(\arg\min f, y_1)$. Then, invoking the convexity of $f$ yields

$$\sum_{t=1}^T \gamma_t \langle x_t - x^*, \nabla f(x_t) \rangle \geq \sum_{t=1}^T \gamma_t [f(x_t) - f(x^*)] = \left[\frac{\sum_{t=1}^T \gamma_t f(x_t)}{\sum_{t=1}^T \gamma_t} - \min f\right] \sum_{t=1}^T \gamma_t \geq [f(\tilde{x}_T) - \min f] \sum_{t=1}^T \gamma_t, \quad (\text{E.75})$$

and it only remains to apply Theorem E.1 and divide both sides by $\sum_{t=1}^T \gamma_t$ to obtain the result. ∎

*Remark* E.6. Theorem E.9 also recovers exactly the convergence rates established by Nemirovski & Yudin [77, p. 190] and Vural et al. [108, Theorem 6] for optimizing convex function with (SMD). Two points are worth emphasizing: (a) our result remains valid when $f$ is only *relatively* smooth with respect to the regularizer, whereas both of the cited works assume smoothness relative to the standard Euclidean norm, thus making our setting more flexible—especially when $h$ is steep; and (b) their convergence rates also hold for regularizers that are not strongly convex but merely $(1, p/(p-1))$-*uniformly convex*, that is, if

$$D(x', x) = h(x') - h(x) - \langle \nabla h(x), x' - x \rangle \geq \frac{p-1}{p}\|x - x'\|^{\frac{p}{p-1}} \quad \text{for all } x, x' \in \mathcal{X}_h. \qquad (\text{E.76})$$

Importantly, Lemma E.2 and, by extension, Lemma E.4, can be easily extended to this setting, since we then directly obtain $F(x, y') \geq \frac{p-1}{p}\|x - Q(y')\|^{\frac{p}{p-1}}$ by Proposition B.2. Hence, all of our results remain valid when strong convexity is relaxed to uniform convexity of the regularizer.

Analogously to stochastic dual averaging, we can then derive convergence rates for the *last iterate* of (LMD) under relative strong convexity. For a constant stepsize schedule, this gives the following bound, which coincides with that of (SDA).

**Theorem E.10** (Convergence rate of (LMD) for relatively strongly convex functions, constant stepsize). *Let $h$ be a steep regularizer. If $f$ is relatively $L$-smooth and relatively $\mu$-strongly convex with minimum $x^* \in \mathcal{X}$, then running* (LMD) *with constant stepsizes $\gamma \leq \frac{1}{pL}$ achieves*

$$\mathbb{E}[\|x_t - x^*\|^2] \leq \frac{2^{p-1}\|\mathcal{X}\|^{2-p}}{p\mu K^p}\sigma^p\gamma^{p-1} + \frac{2D(x^*, x_1)}{K}(1 - \gamma\mu)^{t-1}. \qquad (\text{E.77})$$

*Proof.* Since the stepsizes are constant, this result follows from the exact same arguments and computations used to prove Theorem E.6 for (SMD) with constant learning rate. ∎

On the other hand, for a variable stepsize of order $1/t^{1/p}$ we obtain the following convergence guarantee, which requires a slightly more refined non-asymptotic version of Chung's lemma in order to be derived.

**Theorem E.11** (Convergence rate of (LMD) for relatively strongly convex functions, decreasing stepsize). *Let $h$ be a steep regularizer. If $f$ is relatively $L$-smooth and relative $\mu$-strongly convex with minimum $x^* \in \mathcal{X}$, then running* (LMD) *with stepsizes $\gamma_t = \beta t^{-1/p}$ where $\beta \leq \frac{1}{pL}$ achieves the convergence rate*

$$\mathbb{E}[\|x_t - x^*\|^2] \leq \frac{2^{p+1}\beta^{p-1}\|\mathcal{X}\|^{2-p}\sigma^p}{\mu K}t^{-\frac{p-1}{p}} + \frac{2D(x^*, x_1)}{K}\exp\left(-\frac{\mu\beta}{2}t^{\frac{p-1}{p}}\right) \qquad (\text{E.78})$$

*for every times $t \geq \left[\frac{2(p-1)}{p\mu\beta}\right]^{\frac{p}{p-1}}$. In particular, $x_t \to x^*$ in mean square as $t \to \infty$.*

*Proof.* Following the same arguments as in the proof of [Theorem E.7](#) but starting from [Lemmas E.3](#) and [E.4](#) instead, we get

$$\mathbb{E}[D(x^*, x_{t+1})] \leq \mathbb{E}[D(x^*, x_t)] - \mu \gamma_t \mathbb{E}[D(x^*, x_t)] + \frac{2^{p-1} \|\mathcal{X}\|^{2-p} \sigma^p}{p K^{p-1}} \gamma_t^p$$

$$= (1 - \mu \gamma_t) \mathbb{E}[D(x^*, x_t)] + \frac{2^{p-1} \|\mathcal{X}\|^{2-p} \sigma^p}{p K^{p-1}} \gamma_t^p. \tag{E.79}$$

In particular, letting $\gamma_t = \beta t^{-1/p}$, we obtain

$$\mathbb{E}[D(x^*, x_{t+1})] \leq \left(1 - \frac{\mu \beta}{t^{1/p}}\right) \mathbb{E}[D(x^*, x_t)] + \frac{2^{p-1} \|\mathcal{X}\|^{2-p} \sigma^p \beta^p}{p K^{p-1}} \frac{1}{t}. \tag{E.80}$$

Defining $c = \mu \beta$ and $d = \frac{1}{p} K^{1-p} \|\mathcal{X}\|^{2-p} \sigma^p \beta^p$, we can then use a non-asymptotic version of Chung's lemma (cf. [41, Theorem 24]) to get that, for every $t \geq t_0 \geq \left[\frac{2(p-1)}{pc}\right]^{p/(p-1)}$,

$$\mathbb{E}[D(x^*, x_t)] \leq \frac{2d}{c} (t + t_0)^{-\frac{p-1}{p}} + D(x^*, x_1) \exp\left(-\frac{ct}{(t+t_0)^{1/p}}\right)$$

$$\leq \frac{2d}{c} t^{-\frac{p}{p-1}} + D(x^*, x_1) \exp\left(-\frac{ct}{(2t)^{1/p}}\right)$$

$$= \frac{2d}{c} t^{-\frac{p-1}{p}} + D(x^*, x_1) \exp\left(-\frac{c}{2} t^{\frac{p-1}{p}}\right), \tag{E.81}$$

and it only remains to use the upper bound $\|x_t - x^*\|^2 \leq \frac{2}{K} D(x^*, x_t)$ to prove the claimed convergence rate. ∎

We conclude this section by proving [Theorem 9](#), restated here for completeness, which provides refined convergence rates for ([LMD](#)) under the standard stepsize $\gamma_t \propto 1/t$.

**Theorem 9** (LMD, variable $\gamma$). *Suppose that $f$ is $\mu$-strongly convex and $L$-smooth, both relative to $h$, with $h$ steep. If* ([LMD](#)) *is run with step-size $\gamma_t = \beta/t$, $\beta \leq 1/(pL)$, then*

$$\mathbb{E}\left[\|x_t - x^*\|^2\right] = \begin{cases} \mathcal{O}(1/t^{p-1}) & \text{if } p < 1 + \beta \mu, \\ \mathcal{O}(\log t / t^{p-1}) & \text{if } p = 1 + \beta \mu, \\ \mathcal{O}(1/t^{\beta \mu}) & \text{if } p > 1 + \beta \mu. \end{cases} \tag{42}$$

*Proof.* Coming back to [Eq. (E.79)](#) and putting $\gamma_t = \beta t^{-1}$, we obtain

$$\mathbb{E}[D(x^*, x_{t+1})] \leq \left(1 - \frac{\beta \mu}{t}\right) \mathbb{E}[D(x^*, x_t)] + \frac{d}{t^p}, \tag{E.82}$$

and the asymptotic convergence rates are therefore direct consequences of Chung's lemma (see e.g., [89, Lemma 4, p.45]). ∎

*Remark* E.7. Non-asymptotic convergence rates could also be obtained through the use of non-asymptotic Chung-type estimates, such as those in [41, Theorem 24], but we restrict our presentation here to asymptotic bounds for simplicity.

*Remark* E.8. Analogous to the convergence rates, we obtain that for every target $\varepsilon > 0$, the stepsize sequence $\gamma_t = \beta/t$ achieves the accuracy $\mathbb{E}\left[\|x_t - x^*\|^2\right] \leq \varepsilon$ for every times

$$t = \begin{cases} \Omega\left(\varepsilon^{-\frac{1}{p-1}}\right) & \text{if } p < 1 + \beta \mu \\ \Omega\left(\left[\frac{1}{\varepsilon} \log\left(\frac{1}{\varepsilon}\right)\right]\right) & \text{if } p = 1 + \beta \mu \\ \Omega\left(\varepsilon^{-\frac{1}{\beta \mu}}\right) & \text{if } p > 1 + \beta \mu. \end{cases} \tag{E.83}$$

In particular, the first two regimes—that is, when $p \leq 1 + \beta \mu$—matches the complexity rate obtained with a optimally tuned constant stepsize. By contrast, when $p > 1 + \beta \mu$, the resulting complexity is always worse than that of the constant stepsize

rule, although it remains sharper than the convex case as long as $p \leq 1 + \frac{\beta\mu}{1-\beta\mu}$. This gap with the optimal lower bounds stems from our use of relative smoothness. Indeed, if one could take $\beta = 1/\mu$, the complexity would become $\Omega(\varepsilon^{-1/(p-1)})$ for all $p \in (1, 2]$, thereby recovering the optimal lower bound for $p = 2$ and matching the constant-stepsize complexity when $p < 2$. However, the argument used in Lemma E.4 additionally requires the condition $\beta \leq 1/(pL)$, preventing this optimal stepsize choice. That said, we believe this limitation is methodological rather than fundamental. For instance, Lu [63] obtained the optimal complexity bound $\Omega(1/\varepsilon)$ for (SMD) with stepsize $\gamma_t \propto 1/t$, under the relative boundedness condition

$$\mathbb{E}\big[\|\mathsf{G}(x;\omega)\|_*^2\big] \leq M^2 \frac{D(q,x)}{\|q-x\|^2} \quad \text{for every } x, q \in \mathcal{X} \tag{E.84}$$

on the stochastic oracle, showing that optimal rates are achievable in stochastic mirror descent under refined assumptions. Extending this approach to heavy-tailed noise is, however, nontrivial. In particular, we conjecture that matching the optimal convergence rate $\mathcal{O}(t^{-2(p-1)/p})$ requires working with the $p/2$-th power of the Bregman divergence, as was done by Fatkhullin et al. [32] for projected SGD (where the Bregman divergence reduces to the Euclidean distance). We therefore leave this question open for future work.

