# OpenReview forum: "Bregman meets Lévy: Stochastic Mirror Descent with Heavy-Tailed Noise in Continuous and Discrete Time"
_ICML.cc/2026/Conference — ICML 2026 regular_

### Official Review · Reviewer_vWAZ · 2026-03-08

**Soundness:** 3
**Presentation:** 3
**Significance:** 3
**Originality:** 3
**Overall Recommendation:** 4
**Confidence:** 3

**Summary:**

The paper investigates the robustness of stochastic mirror descent under heavy-tailed noise. In particular, a continuous-time approximation of stochastic mirror descent under heavy-tailed noise is introduced which is a L\'{e}vy-driven stochastic differential equation. When the $p$-th moments are finite for the L\'{e}vy noise, the paper shows that this continuous-time approximation achieves $\epsilon$-optimality within $O(\epsilon^{-p/(p-1)})$ time in the convex case and $O(\epsilon^{-1/(p-1)})$ for the (relatively) strongly convex case. Discrete time analysis is also presented in the paper.

**Compliance With Llm Reviewing Policy:**

Affirmed.

**Final Justification:**

The response from the author(s) in the rebuttal is satisfactory.

**Key Questions For Authors:**

(1) Theorem 2 on the concentration holds under the assumption that $f$ is $\mu$-strongly convex. What if you have $f$ to be $\mu$-strongly convex relative to $h$?

(2) Theorem 3 on the hitting time holds under the assumption that $f$ is $\mu$-strongly convex.  What if you have $f$ to be $\mu$-strongly convex relative to $h$?

(3) Your complexity discussions, such as those in Remark 7, only keeps track the dependence on $\varepsilon$, the accuracy level. In the case of $\mu$-strong convexity or $\mu$ relative strong convexity, is it possible for you to also discuss the dependency on $\mu$. Similarly, for the discussions for the discrete-time analysis, it would be nice if you can discuss the iteration complexity dependence on $\mu$ and also $L$, the smoothness constant, or on the condition number $\kappa:=L/\mu$.

(4) For reference [16], if it is a book, the title of the book should be capitalized.

**Limitations:**

There are some discussions on the open problems, but not on the limitations.

**Strengths And Weaknesses:**

*Strengths

(1) The study of stochastic mirror descent in the presence of heavy-tailed noises using the continuous-time approximation via a L\'{e}vy-driven SDE is novel and the subsequent theoretical analysis for both the continuous-time and discrete-time dynamics are new.

(2) The paper overall is well written. Technical backgrounds are given to make the paper self-contained. There is extensive literature and many relevant recent papers are cited.

*Weaknesses

(1) One thing that is not very clear to me is that how the continuous-time analysis is helpful to the discrete-time analysis? It seems to me that the proofs for the discrete-time analysis are obtained without using any results from the continuous-time analysis. If that is the case, why is it interesting and important to study the L\'{e}vy-driven SDE approximation of the stochastic mirror descent? If the author(s) can add more discussions why continuous-time approximation is important, and what role it plays to help understand the discrete-time analysis, that will be greatly appreciated.

---

> ### Author Rebuttal · Authors · 2026-03-31
>
> Dear reviewer,
>
> Thank you for your time. We address your remarks and questions below:
>
> > How is the continuous-time analysis helpful to the discrete-time analysis? [...] What role does it play to help understand the discrete-time analysis?
>
> Great question, thanks for raising it. There are several key points to note:
> 1. First, our analysis establishes the following pattern:
> ````Discrete-time bound = O(Continuous-time bound + [f(x_init) - f_min]/T)````
> This can be seen clearly in the bounds of Thms 5 & 6, and also in the extensions to Thms 2 & 3 provided in [our reply to Reviewer zc7A](https://openreview.net/forum?id=69IOkVkTQX&noteId=kxWxMcdIrT). This allows for a particularly nuanced interpretation of the discrete-time bounds: mutatis mutandis, the first term mirrors the continuous-time bound, whereas the second arises from the "discretization" of the process. Thus, while the continuous-time analysis is not a strict logical prerequisite for our discrete-time results, it provides a layered and unified view, and it offers considerable intuition for constructing Lyapunov functions and structuring proofs in a setting where calculus tools are still available (in the "weak Itô" sense).
>
> 2. Beyond our specific setting, there is a vast literature in ML on continuous-time models of SGD/SMD. Often referred to as the "SDE approximation", this approach has played a key role in explaining metastability and escape phenomena in deep learning [10, 77, 97, 98, 115], understanding the dynamics of SGD/SMD in both convex and non-convex settings—cf. the works of Li et al. (ICML 2017, JMLR 2019) on stochastic modified equations, and the works of Raginsky et al. cited in our paper—and studying generalization properties of SGD/SMD under heavy noise [28,30,87,88,99]. In particular, Lévy noise has attracted a lot of interest in ML for designing sampling algorithms—like the fractional Langevin method—as also studied in [37,78,100,110,114].
>
> We understand that this could have been made clearer in the main text, so we will be happy to include this discussion at the first revision opportunity.
>
> > [Thms 2 & 3] hold under the assumption that $f$ is $μ$-strongly convex. What if $f$ is strongly convex relative to $h$?
>
> Good question, thanks!
>
> First, to clarify, if $f$ is relatively strongly convex, it is also strongly convex in the ordinary sense, so our results apply as stated.
>
> Going beyond this "default" guarantee, it is interesting to ask if sharper concentration bounds can be obtained under relative strong convexity. The short answer is "yes" in the Gaussian regime of (SMF):
> - First, for Thm 2, if we consider the evolution of $e^{λ E(t)}$ instead of $E(t)$, a Chernoff-type argument can be used to obtain the concentration guarantee $$𝔼[μ_t(\mathcal{U}_δ)]\ge1-\exp[-O(δ^2-δ_c^2)^2]-O(1/t)$$ which is sharper than the polynomial $δ^2/δ_c^2$ bound of Thm 2.
> - Similarly, for Thm 3, applying Itô formula to the logarithmic energy $\log(E(t)-λ)$ yields the hitting time estimate $$𝔼[τ_δ]=\tilde O(1/δ^2)$$ when $η=O(δ^2)$. For small $δ$, this estimate is again sharper than the $O(1/δ^4)$ bound of Thm 3.
>
> However, the extension to heavy noise presents major challenges: $e^{λ E(t)}$ generally has infinite mean due to the heavy tails, so a similar proof would require studying some higher moment of $E(t)$. As for Thm 3, applying Itô's formula introduces jump terms that require a much finer treatment since the process can "teleport" to a vicinity of $x^*$.
>
> We defer these very interesting directions to future work as they lie beyond the scope of the current paper. That said, we would gladly include the sharper derivations for the Gaussian case in a follow-up post (and the paper) if you find this useful.
>
> > Your complexity discussions [only] track the dependence on $ε$. Is it possible to also discuss the dependency on $μ$ and/or the condition number $κ = L/μ$?
>
> Great question, thanks! In continuous time, our analysis in the appendix actually establishes the following:
> - **$μ$-strongly convex:** $O((με)^{-p/(p-1)})$.
> - **$μ$-relatively strongly convex:** $O((μ^pε)^{-1/(p-1)}\log(1/ε))$.
>
> In discrete time, the situation is trickier, but we can get the following bounds:
> - **$μ$-strongly convex:** $O(κ/ε)+O((με)^{-p/(p-1)})$
> - **$μ$-relatively strongly convex:** $O((μ^p ε)^{-1/(p-1)}\log(1/ε))$
>
> [Please let us know if you require the details of the above computations.]
>
> The last case is very delicate because the condition number captures the disparity of the min-max eigenvalues of the Hessian, an inherently Euclidean concept. In a Bregman setting, the relation between $κ$ and the underlying regularizer is not clear, so we cannot provide a similar expression.
>
> ---
> Thank you again for your time. We will be happy to include the above discussion at the first revision opportunity. In the meantime, we hope and trust that our replies have resolved your concerns and we look forward to any further questions you may have.
>
> Kind regards,
>
> The authors

---

> > ### Author Rebuttal · Reviewer_vWAZ · 2026-04-05
> >
> > The rebuttal is satisfactory. I will raise the score.

---

> > > ### Author Response · Authors · 2026-04-05
> > >
> > > Dear Reviewer vWAZ,
> > >
> > > Thank you for your reply and your increased support—we were very glad to see that our response addressed your questions.
> > >
> > > Kind regards,
> > >
> > > The authors

---

### Official Review · Reviewer_31hV · 2026-03-09

**Soundness:** 2
**Presentation:** 2
**Significance:** 2
**Originality:** 2
**Overall Recommendation:** 3
**Confidence:** 3

**Summary:**

This paper studies Stochastic Mirror Descent (or its variants) in both continuous and discrete time. In continuous time, the authors proposed a new scheme, Lévy Mirror Flow (LMF), and provided some convergence analysis. In discrete time, the authors proved some finite-time convergence rates for Stochastic Dual Averaging (SDA) and Lazy Mirror Descent (LMD).

**Compliance With Llm Reviewing Policy:**

Affirmed.

**Final Justification:**

I am convinced by the authors' contribution to continuous-time analysis. However, for discrete-time analysis, I think they have only a limited set of values.

**Updated Final Justification**

I summarize my thoughts below.

1. My comments are based on [52, v2], and the authors said the version they were aware of is [52, v1]. I choose to believe the authors.

1. However, there are still three points I need to mention:

   - As far as I can check, even in [52, v1], the extension to general norms has already been explicitly pointed out in Remark 1.

   - For people familiar with the theoretical analysis of SGD/SMD/SDA, it is well known that the proofs for Lipschitz and smooth objectives do not differ much once one can properly deal with gradient noise. Clearly, the analysis in [52, v1] already shows how to handle heavy-tailed noise, and importantly, its proof did not really rely on the choice of the Euclidean norm (see Eq. (5) in [52, v1]).

   - The authors cited the following writing in [52, v2] to argue that their analysis for SDA cannot be covered by [52].

     > _the proof strategies for OGD and DA are in different flavor (even for $p=2$)_

     However, after checking [52, v2], I think this is a misleading reply and does not hold, since the complete phrase in [52, v2] is that:

     > _Despite the proof strategies for OGD and DA are in different flavors (even for $p=2$),  the basic idea presented before for OGD still works here, i.e., apply the boundness property of $\mathcal{X}$ to make the term $\Vert \epsilon_t \Vert$ have a correct exponent._

     From my understanding, the above sentence means that Eq. (5) in [52, v1 and v2] works for both SGD and SDA.


   Therefore, even though the authors only noticed [52, v1], the discrete-time analysis in the current paper only has limited value, in my opinion.

1. Even though my attitude remains negative for discrete-time analysis, I am fine to accept this paper due to the following two reasons:

   - After the rebuttal, I am inclined to believe the continuous-time analysis has value and deserves a publication. (However, I should clarify again that I only have limited knowledge in this field.)

   - My above criticism of discrete-time analysis may be tough, as different researchers have their own understanding of theoretical contributions. But I still choose to write it down and will leave it here, as it reflects my evaluation of this paper.

To summarize, I am fine with accepting this paper, even though my final evaluation remains mixed (positive for the continuous-time analysis and negative for the discrete-time one).

**Key Questions For Authors:**

Please refer to **Weaknesses**.

**Limitations:**

yes

**Strengths And Weaknesses:**

**Strengths.** The newly introduced LMF scheme and its continuous-time analysis are new to me (maybe due to my limited knowledge in this field) and could be potentially useful in the future.

**Weaknesses.** I summarize the weaknesses below.

**Technique (Discrete-Time).** As far as I can check, the technique used in the proof for the discrete-time case is highly similar to [52], and the obtained results are even weaker.

   1. For example, in Lemma E.2 (the core lemma for SDA), (E.12) and (E.13) essentially apply the same trick proposed in [52]. For LMD, the same thing holds.

   1. The proved rates hold for (a) the average iterate (convex optimization) and (b) the squared distance from the optimal solution for the last iterate (strongly convex optimization), which can be done by (a) the online-to-batch conversion and (b) the standard analysis for strongly convex optimization (even in the online case). In other words, the results actually can be covered by [52].

   1. Moreover, both metrics mentioned above are weaker than the last-iterate convergence in the function-value gap, considered in [52] for stochastic convex optimization.

   1. Although the theorems presented in this work are under a general norm and seemingly new compared to [52], this extension was already explicitly pointed out in Remark 1 of [52].

**Format.** Per ICML requirements, the citation should be in the Author-Year format instead of the Author-Number style used in the current paper.

**Writing.** The paper should be further polished. For example, I list some typos here:

   1. Line 36, “architetures” should be “architectures”.

   1. Line 664, “variant” should be “variants”.

   1. Lines 665 and 694, “later” should be “latter”.

   1. Line 1240, $\Vert y-y'\Vert_*$ should be $\Vert y-y'\Vert$.

   1. Line 2190, $D(x,Q(y')$ should be $D(x,Q(y'))$.

**Final Remark.** My current ratings are mainly based on the review of discrete-time analysis/results, without accounting for the potential contributions of continuous-time analysis, since I do not have enough background in Lévy-driven SDEs. Therefore, I potentially adjust my scores based on other reviewers' reviews of the continuous-time part (once I can read them) and the authors' corresponding rebuttal.

---

> ### Author Rebuttal · Authors · 2026-03-30
>
> Dear reviewer,
>
> Thank you for your time. We reply to your remarks and questions below:
>
> > My current ratings are mainly based on the review of discrete-time analysis/results, without accounting for the potential contributions of continuous-time analysis, since I do not have enough background in Lévy-driven SDEs.
>
> We appreciate your candor and the time you took to review our paper, especially given the mismatch in background. At the same time, we would like to respectfully note that the continuous-time analysis constitutes a central component of the paper, both technically and conceptually—especially since the weak Itô formula that we derive appears to be completely new in the literature.
>
> We therefore trust that your final evaluation will properly take into consideration the paper as a whole, not only a part thereof.
>
> > In Lemma E.2 (the core lemma for SDA), (E.12–E.13) essentially apply the same trick proposed in [52].
>
> The specific algebraic manipulations used in (E.12–E.13) to obtain an upper bound compatible with heavy-tailed noise are not novel techniques, and can be traced back even further than [52]: for example, a slightly different variant thereof also appears in the paper by Vural et al. [97, Eqs. B.2–B.4]. However, this is but a small part of a much bigger whole: the main novelty of our discrete-time analysis lies in the fact that we are dealing with objectives that do not conform with standard notions of smoothness and have **unbounded gradients**. We elaborate on this below.
>
> > The proved rates hold for (a) the average iterate and (b) the squared distance from the optimal solution for the last iterate. In other words, the results actually can be covered by [52].
>
> We are concerned there is an important misunderstanding here, as this comparison overlooks a **fundamental difference** between the setting of [52] and our own: **our analysis applies to functions with *unbounded gradients;* [52] assumes *bounded* gradients, so it does not cover our results.** This feature is essential to many of the practical applications of mirror descent (like entropically regularized optimal transport, Fisher markets, Poisson inverse problems,...), and it is precisely the reason that $\mathrm{dom}\partial f$ may be *strictly smaller* than $\mathcal{X}=\mathrm{dom}f$ in our case.
>
> In more detail, regarding the last-iterate convergence result mentioned in your review, [52] introduces a "generalized nonsmoothness condition" (Eq. 29) to broaden the class of objective functions under study, cf. Theorem 10 and the corollaries that follow. However, when restricted to a bounded set, this condition implies bounded (sub)gradients, so it cannot cover the unbounded gradient case that is the focus of our analysis.
>
> Furthermore, as far as we could tell, Theorem 10 in [52] is proven **exclusively** for SGD, not even SDA with Euclidean norm—and [52] makes a clear distinction between SGD and SDA on this matter. When run with a variable post-multiplier, the analysis of SDA is different from that of SGD, so we do not see a path extending [52] to objectives with unbounded gradients and SDA schemes with regularizers that are not norm-based.
>
> > Although the theorems presented in this work are under a general norm and seemingly new compared to [52], this extension was already explicitly pointed out in Remark 1 of [52].
>
> We respectively—but firmly—disagree: the extension that this remark suggests is orthogonal to our paper.
> - First, it is essential to distinguish between classical strong convexity (with respect to some norm, possibly non-Euclidean) and _relative strong convexity_ (that is, with respect to a Bregman function—*not a norm*). Both cases are important to study, but the latter is deeply intertwined with the unbounded gradient issue discussed in detail above: if the regularizer is steep ($\|\nabla h\|\to\infty$ at the boundary), **relatively strongly convex functions have *unbounded* gradients.**
> - Second, our analysis leverages the *relative smoothness* framework of Lu, Freund & Nesterov [61], and Bauschke, Bolte & Teboulle [11], which explicitly allows for objective whose gradients blow up at the boundary of $\mathcal{X}$. Because this condition is no longer norm-based (so all notions of symmetry or homogeneity break down), the resulting analysis is very different from the online, non-smooth framework of [52].
>
> To reiterate, we are not aware of any comparable results in the literature for functions whose gradients blow up at the boundary of their domain—and while [52] treats some related questions, there is no overlap.
>
> > Formatting and typos
>
> We will take care of those, thanks.
>
> ---
> Thank you again for your time—we will be happy to include the above discussion and a more detailed comparison with the (orthogonal) results of [52] at the first revision opportunity. In the meantime, we hope and trust that our replies have resolved your concerns and we look forward to any further questions you may have.
>
> Kind regards,
>
> The authors

---

> > ### Author Rebuttal · Reviewer_31hV · 2026-04-03
> >
> > I thank the authors for their response.
> >
> > 1. For continuous-time analysis, I now partially agree with the authors' reply and recognize the contributions. Based on this fact, I decided to increase my rate from 2 to 3.
> >
> > 1. For discrete-time analysis, I am not convinced, as the authors' reply contained many overclaims in my view.
> >
> >     - The authors claimed that (E.12) and (E.13) are not core and said a similar analysis appeared in [97]. However, this statement is misleading. As even mentioned by the authors themselves, [97] relies on a uniformly convex mirror map. This fact is critical and therefore distinguishes the analysis in [97] from that in [52]. This is also why [52] and this paper heavily rely on a bounded domain. Essentially, the trick applied in this paper is the same as that in [52].
> >
> >       In addition, if the authors think (E.12) and (E.13) are not the central steps of discrete-time analysis, which part do you think is the core of the analysis? As far as I can tell, the remaining steps are in fact standard.
> >
> >    - The authors claimed that bounded gradients are required to [52], which is also not correct. For example, while it is true that a bounded domain implies bounded gradients for smooth functions, the analysis and the rate in [52] do not rely on this fact after checking Theorem 10 in [52].
> >
> >    - It seems the authors keep stating that the analysis for Euclidean smoothness (or strong convexity) cannot be applied to the relative smoothness (or strong convexity) version. This is also not accurate. To the best of my knowledge, the analysis for non-accelerated first-order algorithms under these two assumptions essentially follows the same template (even for heavy-tailed optimization). The subtle part will appear only for the accelerated algorithm, which is, however, not considered in this paper.
> >
> > I am open to more discussions.

---

> > > ### Author Response · Authors · 2026-04-05
> > >
> > > Dear Reviewer 31hV,
> > >
> > > Thank you for your time and follow-up. We reply to the points you raise below:
> > >
> > > > For continuous-time analysis, I now recognize the contributions. Based on this fact, I decided to increase my score.
> > >
> > > Thank you for increasing your score. In passing, we also refer you to [our reply to Reviewer zc7A](https://openreview.net/forum?id=69IOkVkTQX&noteId=kxWxMcdIrT) where we complement our analysis with a discrete-time counterpart of Theorems 2 and 3 (for the hitting time and concentration of SDA in discrete time respectively). To the best of our knowledge, there are no similar results in the literature, even in the tame-noise regime.
> > >
> > > > The authors claimed that bounded gradients are required to [52], which is also not correct. [...] The rate in [52] do not rely on this fact after checking Theorem 10 in [52].
> > >
> > > We were confused here, so we revisited [52], and we believe we finally unearthed the source of the confusion:
> > >
> > > **There are 3 versions of [52] : v1 (Aug 10, 2025), v2 (Dec 30, 2025), and v3 (Mar 19, 2026).**
> > >
> > > To clarify the timeline: during the preparation of our manuscript (and until this discussion), we were only aware of v1, which explicitly assumes bounded gradients [52, Eq. 4], in both the statement and proof of Thm 10. In v2, **less than a month before the ICML submission deadline,** this condition was replaced by the assumption $$f(x) - f(y) - \langle \nabla f(y), x-y \rangle \leq 2G\|x-y\| + \frac{H}{2}\| x - y\|^2,$$ with the proof accordingly amended to circumvent [52, Eq.4].
> > >
> > > We agree that this newer formulation is much closer in spirit to our setting, and the resulting arguments are more directly comparable. To be fully transparent, we were simply unaware of the December version of [52] until the discussion period, and we hadn't found anything in [52, v1] suggesting such an extension.
> > >
> > > The connection is now clear, and we will be happy to provide due credit and explain the relation with the updated versions of [52] in detail. At the same time, we would like to point out that, as per the [ICML reviewing guidelines](https://icml.cc/Conferences/2026/ReviewerInstructions), **v2 and v3 of [52] fall under the status of concurrent work,** which we hope you will take into account in your final evaluation.
> > >
> > > > If (E.12) and (E.13) are not the central steps of discrete-time analysis, which part do you think is the core of the analysis? [...] The analysis for non-accelerated first-order algorithms under these two assumptions essentially follows the same template (even for heavy-tailed optimization).
> > >
> > > Please note that a central aspect of our paper is to establish a concrete link between the continuous- and discrete-time settings. As we detailed in [our reply to Reviewer vWAZ](https://openreview.net/forum?id=69IOkVkTQX&noteId=9GFWxmJkHn) (which we will include in our paper in the first revision opportunity), one of the main take-aways of our work is the pattern
> > > ```
> > > Discrete-time bound = O(Continuous-time bound + [f(x_init) - f_min]/T)
> > > ```
> > > This pattern also underpins the discrete-time version of Theorems 2 and 3 (on the concentration and hitting time) that we mentioned above.
> > >
> > > Regarding the discrete-time analysis, a key point is that Theorem 10 of [52] is only proved for SGD, so it does not cover SDA (even under standard Euclidean norms). This distinction is important because, as noted in [52, v2] "_the proof strategies for OGD and DA are in different flavor (even for $p=2$)_". To the best of our knowledge, the closest related work in this direction is the recent paper of Liu and Zhou (ICLR 2024), **which treats SMD (not SDA) and assumes light-tailed noise**.
> > >
> > > We will explicitly cite and discuss this last paper (also updated on 29 Dec 2025 and 19 Mar 2026) at the first revision opportunity. At the same time, although there are similarities in the proof template, we would like to point out that our contribution does not lie in merely extending said results, but also in identifying the precise conditions under which such an extension is valid and providing a complete analysis—neither of which is as standard as a first reading may suggest, see e.g., Theorem E.5 and its proof.
> > >
> > > > [97] relies on a uniformly convex mirror map. This fact is critical and therefore distinguishes the analysis in [97] from that in [52]. [...] Essentially, the trick applied in this paper is the same as that in [52].
> > >
> > > You are right that [97] requires uniform convexity with exponent $p/(p-1)$ over $\mathbb{R}^d$. We mentioned [97] here because, in problems with bounded domains, a strongly convex regularizer is automatically uniformly convex with exponent $p/(p-1)$. That said, we understand your point, so we will cite Liu [52] and provide a detailed explanation of the above in Remark E.5 and elsewhere in the paper.
> > >
> > > ---
> > > We hope this clarifies the relationship with [52] and the wider scope of our contributions (in both continuous- and discrete-time).
> > >
> > > Thank you again for your time. Kind regards,
> > >
> > > The authors

---

### Official Review · Reviewer_Y7Tb · 2026-03-11

**Soundness:** 3
**Presentation:** 3
**Significance:** 3
**Originality:** 3
**Overall Recommendation:** 4
**Confidence:** 1

**Summary:**

This paper deals with the problem of stochastic mirror descent with heavy-tailed noise. To address this, the authors propose the Levy mirror flow as a natural extension of stochastic mirror descent under the heavy-noise regime. Using the Levy mirror flow, the authors obtain relatively robust convergence rates.

**Compliance With Llm Reviewing Policy:**

Affirmed.

**Final Justification:**

I appreciated the author's clear response

**Key Questions For Authors:**

See weakness.

**Limitations:**

yes.

**Strengths And Weaknesses:**

**Strengths**
* This paper is mathematically sophisticated and appears sound.
* I am not an expert in this area. However, the paper is clearly structured, which allowed me to follow the main argument at least at a high level.
* The authors show that stochastic mirror descent in the untamed heavy-noise regime attains robust convergence results in both continuous and discrete time.

**Weakness**
* I do not see this as a weakness, It may come from my limited familiarity with the convex optimization literature. But, for readers who are not familiar with that literature, it would be helpful to provide numerical experiments demonstrating the convergence results (if it is possible) or a conceptual illustration of how heavy-tailed noise benefits the convergence result.

---

> ### Author Rebuttal · Authors · 2026-03-30
>
> Dear reviewer,
>
> Thank you for your time and positive evaluation! We reply to your remarks and questions below:
>
> > It would be helpful to provide numerical experiments demonstrating the convergence results (if it is possible) or a conceptual illustration of how heavy-tailed noise benefits the convergence result.
>
> Due to space constraints, we focused on the theoretical analysis in the current version, but we understand your point that empirical illustrations would improve the reach of our results. At the first revision opportunity, we will be including a set of numerical experiments for SDA in the presence of $\alpha$-stable noise, tracking the quantities that directly reflect the theory, namely (i) convergence of function values; (ii) the hitting time of a $\delta$-neighborhood of the minimum set of $f$; and (iii) the empirical distribution of the process, illustrating in detail how these change when we pass from the Gaussian ($\alpha=2$) to the heavy-tailed regime $(\alpha\to 1)$.
>
> These additions can be implemented in a lightweight manner, readily adaptable to any further suggestions you may have.
>
> ---
> Thank you again for your time and positive evaluation—and please let us know if you have any further questions in the meantime.
>
> Kind regards,
>
> The authors

---

> > ### Author Rebuttal · Reviewer_Y7Tb · 2026-04-03
> >
> > Thank you for the response. I appreciate the plan to include numerical experiments in a revision. I will keep my score unchanged.

---

> > > ### Author Response · Authors · 2026-04-05
> > >
> > > Dear Reviewer Y7Tb,
> > >
> > > Thank you for your reply and your continued support—we were very glad to see that our response addressed your questions.
> > >
> > > Kind regards,
> > >
> > > The authors

---

### Official Review · Reviewer_zc7A · 2026-03-13

**Soundness:** 4
**Presentation:** 3
**Significance:** 3
**Originality:** 4
**Overall Recommendation:** 5
**Confidence:** 2

**Summary:**

This paper aims to examine the robustness of stochastic mirror descent methods under heavy noise. The authors first focus on the Levy mirror flow and derive a weak Ito formula for convex functions that are not $C^2$. Based on this, the authors establish detailed convergence rates for both strongly convex and convex settings. Turning to the discrete-time setting, the authors also derive convergence guarantees consistent with the continuous-time setting.

**Compliance With Llm Reviewing Policy:**

Affirmed.

**Final Justification:**

The authors' rebuttal addresses my concerns and I believe this paper makes a contribution to the study of stochastic mirror descent with heavy-tailed noise, especially from the continuous-time perspective.

**Key Questions For Authors:**

1. In the decomposition of the Levy process, there appears a constant $c$. How to choose this constant and how does it serve as the threshold between the square-integrable martingale and the martingale with finite p-th moments?

2.  Could the authors provide several simple examples of Levy processes with infinite variance?

**Limitations:**

Yes.

**Strengths And Weaknesses:**

[strengths]
1. This paper is well-organized and easy to follow. The analysis is very detailed.
2. The theoretical results are very solid and contribute to the understanding of the stochastic mirror descent under heavy-tailed noise.
3. The techniques are novel in the literature, e.g., the weak Ito formula.

[weaknesses]
1.  Maybe the authors could explain more about the motivations and roles for Theorems 2 and 3. In particular, it seems that the two theorems are limited to the continuous-time setting. To what extent could they capture the behavior for the discrete-time setting?

---

> ### Author Rebuttal · Authors · 2026-03-30
>
> Dear reviewer,
>
> Thank you for your time and positive evaluation! We reply to your remarks and questions below:
>
> > Maybe the authors could explain more about the motivations and roles for Theorems 2 and 3. [...] To what extent could they capture the behavior for the discrete-time setting?
>
> Great question, thanks!
>
> First, to clarify, unlike Theorem 1 (which concerns the convergence of the averaged process $\bar X(t)$ to $\arg\min f$), Theorems 2 and 3 characterize the behavior of $X(t)$ itself. Specifically, Theorem 2 quantifies the long-run concentration of $X(t)$ around the minimizers of $f$ (i.e., the likelihood of observing $X(t)$ at a given distance from $\arg\min f$), while Theorem 3 bounds the first time $X(t)$ enters a $\delta$-neighborhood of a minimizer. These results provide a sharper understanding of the optimization properties of $X(t)$, which is typically the object of interest in practice.
>
> Both theorems can be extended to the discrete-time setting as follows:
> - **Theorem 2:** Following the proof of Theorem D.4, but invoking Theorem E.1 (for SDA) instead of Theorem D.1 (for LMF), the empirical frequency $$
> \mu_T(U_\delta) = \frac{1}{T} \sum_{t=1}^T \mathbb{1}\\{ x_t \in U_\delta \\}
> $$of observing $x_t$ in a $\delta$-neighborhood $U_\delta$ of $\arg\min f$ is bounded as $$
> \mathbb{E}[\mu_T(U_\delta)] \geq 1 - \frac{2(f(x_1) - \min f)}{\mu T \delta^2} - \frac{2H}{\eta \mu \delta^2 T} - \frac{\delta_c^2}{\delta^2}
> $$where $$\delta_c^2 = \frac{2^p \|\mathcal{X}\|^{2-p} \eta^{p-1} \sigma^p}{\mu p}.$$
>
> - **Theorem 3:** This extension requires finer techniques, but otherwise follows the same proof structure as Theorem 3. Specifically, by adapting the arguments from the proof of Theorem D.5 (we will be happy to provide the relevant details in a follow-up post if needed), we can bound the $\delta$-hitting time $$\tau_\delta = \inf\\{ t \in \mathbb{N} : \| x_t - x^* \| \leq \delta  \\}$$ as $$\mathbb{E}[\tau_\delta] = \mathcal{O}\left( \frac{f(x_1) - \min f}{\delta^2} + \left(\frac{\sigma}{\delta^2} \right)^{p/(p-1)} \right)$$
>
> Importantly, in both cases, the derived bounds consist of two components: a part that is stemming from the continuous-time analysis, and another coming from the "discretization" of the process. We find this particularly appealing, as it provides an additional layer of interpretation of our results.
>
>
> > How to choose $c$ and how does it serve as the threshold between the square-integrable martingale and the martingale with finite $p$-th moments?
>
> Another excellent question, thanks!
>
> A Lévy process $L(t)$ has infinite variance iff it exhibits unbounded jumps $\Delta L(t)$ with infinite variance, and the constant $c>0$—*which can be chosen arbitrarily*—defines the threshold between "small" and "large" jumps. Specifically, for any $c>0$, we can decompose $L(t)$ as
> $$
> L(t) := M_c(t) + N_c(t),
> $$
> where, as per Eq. (8), $M_c(t)$ has jumps bounded by $c$ (so it is square-integrable), while $N_c(t)$ may exhibit large, unbounded jumps and may have infinite variance but finite $p$-th moments (if $L(t)$ does).
>
> Now, although $c>0$ can be chosen arbitrarily, our bounds can be optimized further by a judicious choice thereof. We can provide the following guidelines:
>
> 1. **Bounded jumps:** If the jumps of $L(t)$ are bounded by $M > 0$, simply set $c = M + 1$. In this case, $\sigma_{heavy} = 0$.
> 2. **Finite variance ($p=2$):** If $L(t)$ is square-integrable with unbounded jumps, the tightest version of our bounds is obtained in the limit $c \to \infty$ (i.e., the behavior of $L(t)$ is governed by its square-integrable component).
> 3. **Heavy  noise ($p<2$):** In general, $\sigma_{heavy}$ decreases with $c$, and $\sigma_{tame}$ increases with $c$. Thus, if the Lévy measure $\nu$ of $L(t)$ is known (e.g., if $L(t)$ is an $\alpha$-stable process), the value for $c$ could be optimized to yield the tightest possible bounds. In general however, if no information is available on $L(t)$, the standard convention is to take $c=1$: this may not be optimal, but it remains a robust and practical default choice that avoids the need for complex parameter estimation.
>
> > Could the authors provide several simple examples of Levy processes with infinite variance?
>
> Of course! A standard example is compound Poisson processes of the form $L(t) = \sum_{n=1}^{N(t)} Z_n$, where $N(t)$ is Poisson and $Z_n$ are iid heavy-tailed with $\mathbb{E}[Z_n^2] = \infty$. Another is $\alpha$-stable processes defined by their characteristic function $\mathbb{E}[e^{i u^T L(t)}] = e^{-t \| u \|^\alpha}$ for $\alpha \in (1,2)$, interpolating between Brownian motion ($\alpha=2$) and Cauchy processes ($\alpha=1$). Both examples have been studied extensively in the literature.
>
> ---
> Thank you again for your time and positive evaluation. We will of course be happy to include a version of the above at the first possible revision opportunity—in the meantime, please let us know if you have any further questions.
>
> Kind regards,
>
> The authors

---

> > ### Author Rebuttal · Reviewer_zc7A · 2026-04-03
> >
> > Thanks for the detailed responses and I keep my score unchanged. Results like Theorems 2 and 3 are relatively rare in optimization research, so I think it is better to provide more explanation for them and add the corresponding discrete-time versions.

---

> > > ### Author Response · Authors · 2026-04-05
> > >
> > > Dear Reviewer zc7A,
> > >
> > > Thank you for your reply, your kind words and your continued support—we were very glad to see that our response addressed your questions.
> > >
> > > Kind regards,
> > >
> > > The authors

---

### Decision · Program_Chairs · 2026-04-30

**Decision:**

Accept (regular)

**Comment:**

The paper studies the convergence properties of Stochastic Mirror Descent (SMD) and its variants in both continuous and discrete time under heavy-tailed noise, assuming only a bounded $p$-th moment for $p \in (1,2]$. The authors introduce a Lévy mirror flow framework and establish new convergence results in the continuous-time setting under convexity and (relative) strong convexity. They further derive corresponding guarantees in discrete time under convexity and relative smoothness.

Overall, the reviewers find the paper well-organized, clearly written, and technically sound, with results that are relevant and potentially impactful for the community. One reviewer (31hV) raised concerns regarding the significance and novelty of the discrete-time results, noting their similarity to those presented in v2 of [52]. Given that v2 of [52] appeared on December 30, 2025, and considering that the primary contribution of the present paper lies in its continuous-time analysis, it is reasonable to view these discrete-time results as concurrent with [52] (notwithstanding that v1 of [52] already alludes to extensions to general norms in Remark 1).

Taking all these considerations into account, I recommend acceptance of the paper.